# CROSS-DOMAIN LOSSY COMPRESSION VIA RATE- AND CLASSIFICATION-CONSTRAINED OPTIMAL TRANSPORT

**Nam Nguyen, Thinh Nguyen, Bella Bose**
School of Electrical Engineering and Computer Science
Oregon State University
Corvallis, OR 97331, USA
{nguynam4,thinhq,bella.bose}@oregonstate.edu

## ABSTRACT

We study cross-domain lossy compression, where the encoder observes a degraded source while the decoder reconstructs samples from a distinct target distribution. The problem is formulated as constrained optimal transport with two constraints on compression rate and classification loss. With shared common randomness, the one-shot setting reduces to a deterministic transport plan, and we derive closed-form distortion-rate-classification (DRC) and rate-distortion-classification (RDC) tradeoffs for Bernoulli sources under Hamming distortion. In the asymptotic regime, we establish analytic DRC/RDC expressions for Gaussian models under mean-squared error. The framework is further extended to incorporate perception divergences (Kullback-Leibler and squared Wasserstein), yielding closed-form distortion-rate-perception-classification (DRPC) functions. To validate the theory, we develop deep end-to-end compression models for super-resolution (MNIST), denoising (SVHN, CIFAR-10, ImageNet, KODAK), and inpainting (SVHN) problems, demonstrating the consistency between the theoretical results and empirical performance.

## 1 INTRODUCTION

Classical rate-distortion (RD) theory provides a single-letter characterization of the minimal distortion achievable when reproducing a source under a rate constraint (Cover & Thomas, 1999). This foundation has guided decades of research in lossy compression and inspired the design of modern learned codecs. However, standard RD formulations assume that reconstructions should remain close to the *observed* input distribution. In many setting, this assumption is misaligned: the encoder observes a *degraded* sample $X$ (e.g., noisy or low-resolution), while the desired output is a *restored* sample $Y$ that lies in a different, *target* distribution $p_Y$ (e.g., clean or high-resolution). Moreover, beyond fidelity, the compressed representation must remain informative for downstream tasks such as classification, introducing additional constraints that are not captured by classical RD.

Perception-aware RD extends the RD framework by incorporating a divergence between the source and reconstruction distributions, highlighting an intrinsic rate-distortion-perception (RDP) tradeoff and motivating generative compression approaches (Blau & Michaeli, 2018; 2019; Theis & Wagner, 2021). In restoration tasks, the target is not the degraded input $p_X$ but the clean domain $p_Y$, so enforcing perceptual closeness between $p_Y$ and $p_X$ is conceptually mismatched. Task-aware extensions such as rate-distortion-classification (RDC) (Zhang, 2023; Nguyen et al., 2025; 2026) or rate-distortion-perception-classification (RDPC) explicitly account for classification performance (Wang et al., 2025), but typically assume that reconstructions remain in a single domain rather than supporting cross-domain mappings with distinct marginals. In parallel, compression has also been studied as a denoising mechanism. Weissman et al. (Weissman & Ordentlich, 2005) showed that optimal lossy compression followed by post-processing can asymptotically achieve the fundamental denoising limit, while more recent work (Zafari et al., 2025b) introduced neural compression-based denoising, including a zero-shot framework with theoretical guarantees and algorithmic instantiations. These results highlight the value of compression as a denoising prior but do not address cross-

domain alignment or task-aware constraints. Finally, optimal transport (OT) provides a principled tool for coupling distributions (Villani, 2009), and has been leveraged in unsupervised restoration (Wang et al., 2023a). However, OT by itself ignores coding constraints and does not account for downstream requirements. Related to our work, Liu et al. (2022) formulated cross-domain lossy compression as *entropy-constrained OT* and showed that shared randomness can decouple coding from transport, but classification constraints were not included. Unsupervised image restoration has also been studied in (Zhang et al., 2017; Menon et al., 2020; Pan et al., 2021) with a fixed reconstruction distribution; however, these approaches neither investigate compression constraints nor incorporate classification-awareness for downstream tasks.

To that end, we formulate cross-domain restoration as compression, through a *constrained lossy optimal transport* framework. Given degraded samples from $p_X$ and desired reconstructions from $p_Y$, with distortion function, we optimize couplings $p_{X,Y}$ that simultaneously (i) minimize expected distortion, (ii) satisfy a rate constraint, and (iii) preserve task utility by constraining the uncertainty of the downstream label $S$ given the reconstruction $Y$. Following Liu et al. (2022); Theis & Agustsson (2021); Theis & Wagner (2021), we exploit shared common randomness between the encoder and decoder to show that, in the one-shot regime, the system reduces to selecting a deterministic transport plan with effective rate and classification constraints. In the asymptotic regime, this leads to a mutual-information-constrained transport problem with an additional classification constraint, yielding a Shannon-style single-letter characterization. Our work makes the following contributions.

• We introduce constrained lossy optimal transport, generalizing OT by incorporating both coding and classification constraints. With common randomness, transport (reconstructing $Y$ to match $p_Y$) and compression (coding $Y$) structurally decouple, extending Liu et al. (2022) to task-aware settings. Closed-form characterizations are provided for (i) one-shot Bernoulli sources under Hamming distortion, where DRC and RDC functions admit piecewise-linear forms, and (ii) asymptotic Gaussian sources under mean-squared error (MSE), where analytic expressions are derived.

• The framework is further extended to the distortion-rate-perception-classification (DRPC) setting with two perception divergences: Kullback-Leibler (KL) and squared Wasserstein. For Gaussian sources, extremality results yield closed-form DRPC characterizations that, to our knowledge, are the first to explicitly incorporate classification.

• We implement deep end-to-end compression frameworks, incorporating (i) universal quantization for shared randomness, (ii) an entropy model for rate estimation, (iii) a WGAN discriminator for aligning reconstructions with $p_Y$, and (iv) a classifier head for controlling classification loss. Experiments on super-resolution (MNIST), denoising (SVHN, CIFAR-10, ImageNet, KODAK), and inpainting (SVHN), demonstrate strong agreement with the theoretical results.

## 2 SYSTEM MODEL AND ONE-SHOT SETTING RESULTS

We study a scenario where the encoder observes an input $X \sim p_X$, which represents a degraded version (e.g., corrupted by noise, reduced resolution) of an underlying clean source. Associated with each sample is a classification label $S \sim p_S$, with a prescribed covariance between $X$ and $S$ as $\mathrm{Cov}(X,S) = \mathbb{E}[(X - \mu_X)(S - \mu_S)]$. Following the approach in Liu et al. (2022), we utilize shared randomness between the encoder and decoder to enhance performance. Specifically, we introduce a common random variable $U$, accessible at both sides, which is independent of the input

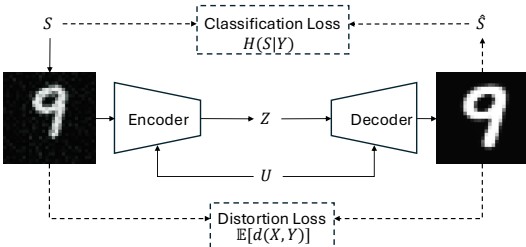

Figure 1: System model: a noisy input $X \sim p_X$ is restored as $Y \sim p_Y$, supporting classification with label $S$.

$X$, i.e., $I(X;U) = 0$. This assumption ensures that the decoder has no prior knowledge of $X$ beyond the transmitted representation. In practice, $U$ can be realized by pre-agreeing on a pseudo-random number generator with a shared seed, enabling both parties to generate identical randomness. The encoder must map $X$ into a compressed representation $Z$ under a rate constraint, namely $H(Z|U) \leq R$. The decoder, given $Z$, produces a reconstruction $Y$ that should follow a target distribution $p_Y$.

We consider $p_X$ and $p_Y$ as probability distributions over $\mathcal{X}, \mathcal{Y} \subseteq \mathbb{R}^n$. Formally, we enforce similarity between $X$ and $Y$ through a fidelity criterion defined by a distortion function $d : \mathcal{X} \times \mathcal{Y} \to \mathbb{R}$. We assume $d(X,Y) = 0$ if and only if $X = Y$. For instance, $d(X,Y)$ can be the Hamming distance or the MSE distortion. In addition, we require the reconstruction to remain useful for downstream classification, specifically, the uncertainty of $S$ conditioned on $Y$ is constrained as $H(S|Y) \leq C$ for some $C > 0$ (Wang et al., 2025).

The main objective of source restoration (e.g., denoising or super-resolution) in our setting is three-fold: (i) *Degradation removal*: mitigate artifacts and imperfections present in the degraded input $X$; (ii) *Information preservation*: retain as much information as possible about the underlying clean source $X'$ contained in $X$; (iii) *Classification utility*: ensure that the reconstructed $Y$ yields high classification performance. As an example, consider $X$ as a noisy image and $Y$ as its clean reconstruction. Figure 1 provides a schematic of the full system, where compression and restoration jointly yield a sample $Y \sim p_Y$ that serves both fidelity and classification purposes. We interpret this formulation through the lens of optimal transport: the problem reduces to identifying a joint distribution $p_{X,Y}$ consistent with given marginals $p_X$ and $p_Y$, subject to a distortion cost $d(\cdot,\cdot)$, rate constraint $R$, and classification constraint $C$. We next connect this framework to optimal transport and describe how our formulation of classification-aware lossy compression naturally extends it.

## 2.1 ONE-SHOT CONSTRAINED OPTIMAL TRANSPORT

**Definition 1** (Optimal Transport). *Let $\Gamma(p_X, p_Y)$ denote the set of all joint distributions $p_{X,Y}$ with marginals $p_X$ and $p_Y$. The classical optimal transport problem identifies a coupling in this set that minimizes the expected transportation cost:*

$$D(p_X, p_Y) = \inf_{p_{X,Y} \in \Gamma(p_X, p_Y)} \mathbb{E}[d(X,Y)], \tag{1}$$

*where $d(\cdot, \cdot)$ is a prescribed distortion (or cost) function and $p_{X,Y} \in \Gamma(p_X, p_Y)$ is a transport plan.*

The transport plan in Definition 1 minimizes the average distortion between input and output, subject only to the marginal distributions $p_X$ and $p_Y$. Our goal is to extend this framework by requiring the transport plan to additionally satisfy a rate constraint and a classification constraint, as formalized in the following definition.

**Definition 2** (Constrained Optimal Transport). *Let $X \sim p_X$ denote the degraded source, $Y \sim p_Y$ be the reconstruction, and $S \sim p_S$ be the associated classification variable with covariance $\mathrm{Cov}(X,S)$ with $S \leftrightarrow X \leftrightarrow Y$. Define $M(p_X, p_Y)$ as the set of joint distributions $p_{U,X,Z,Y}$ with marginals $p_X, p_Y$ that factorize as $p_{U,X,Z,Y} = p_U\, p_X\, p_{Z|X,U}\, p_{Y|Z,U}$, where $U$ is the shared common randomness. The constrained optimal transport problem with rate constraint $R$, classification loss $C$, and shared randomness is given by*

$$D(R, C, p_X, p_Y) = \inf_{p_{U,X,Z,Y} \in M(p_X, p_Y)} \mathbb{E}[d(X,Y)] \tag{2}$$

$$s.t. \quad H(Z|U) \leq R, \quad H(S|Y) \leq C,$$

Given an input $X$ and shared randomness $U$, the encoder produces a compressed representation $Z \sim p_{Z|X,U}$. Leveraging $U$, the representation can be further losslessly encoded at an average rate not exceeding $R$. By standard coding theorems, any discrete $Z$ admits a variable-length code with expected length at most $H(Z|U) + 1$ bits. The decoder, with access to $(Z,U)$, reconstructs $Y$ via $p_{Y|Z,U}$, where the reconstruction is required to satisfy $H(S|Y) \leq C$. The optimization in (2) is carried out jointly over the distribution of shared randomness $p_U$ and the stochastic mappings $p_{Z|X,U}$ (encoder) and $p_{Y|Z,U}$ (decoder). Furthermore, the constrained cost satisfies $D(R, C, p_X, p_Y) \geq D(p_X, p_Y)$, where $D(p_X, p_Y)$ is the classical optimal transport cost from Definition 1. The next result provides a simplification of this architecture.

**Theorem 1.** *Define $Q(p_X, p_Y)$ as the set of joint distributions $p_{U,X,Y}$ with marginals $p_X, p_Y$ that factorize as $p_{U,X,Y} = p_U\, p_X\, p_{Y|X,U}$. Then, the constrained optimal transport cost in Definition 2 admits the representation*

$$D(R, C, p_X, p_Y) = \inf_{p_{U,X,Y} \in Q(p_X, p_Y)} \mathbb{E}[d(X,Y)] \tag{3}$$

$$s.t. \quad H(Y|X,U) = 0, \quad I(X;U) = 0,$$

$$H(Y|U) \leq R, \quad H(S|Y) \leq C.$$

*Proof.* The result follows by adapting Theorem 3 in Liu et al. (2022). For completeness, a detailed proof is provided in Appendix A.1.1. □

Following (Liu et al., 2022), the problem can be equivalently expressed using only the conditional distribution $p_{Y|X,U}$, which generates the reconstruction $Y$ directly without an intermediate representation $Z$, similar to the classical optimal transport formulation in Definition 1. The condition $H(Y|X,U) = 0$ ensures that the transport plan is deterministic once the shared randomness $U$ is fixed, with $U$ providing the sole source of stochasticity. In this architecture, the encoder maps $(X,U)$ to $Y$ (transport), then compresses $Y$ losslessly at an average rate approaching $H(Y|U)$ (compression), while enforcing $H(S|Y) \leq C$ to preserve classification accuracy. The decoder simply decompresses and outputs $Y$.

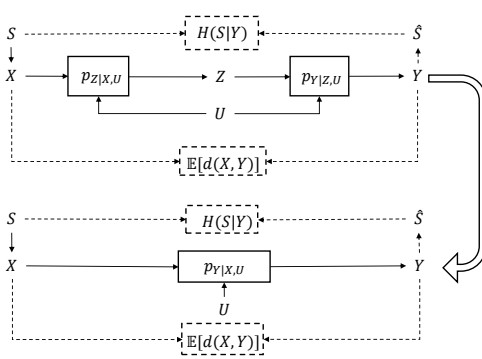

Figure 2: System architecture of Theorem 1.

## 2.2 BERNOULLI CASE EXPRESSIONS

We now investigate the constrained optimal transport framework for Bernoulli sources. Let $X \sim \text{Bern}(q_X)$ and $Y \sim \text{Bern}(q_Y)$ with $0 \leq q_X, q_Y \leq \frac{1}{2}$. Using $\oplus$ for modulo-2 addition, note that $X \oplus Y = 1$ iff $X \neq Y$. The classification variable $S$ is modeled as $S = X \oplus S_1$, where $S_1 \sim \text{Bern}(q_{S_1})$ with $0 \leq q_{S_1} \leq \frac{1}{2}$. This yields the marginal distribution $q_S = P(S = 1) = q_X + q_{S_1} - 2q_X q_{S_1}$.

We adopt the Hamming distortion $d_H(X,Y) = \mathbf{1}\{X \neq Y\}$. For any coupling of $X$ and $Y$ with these marginals, let $p_{xy} = P(X = x, Y = y)$. The expected distortion is

$$\mathbb{E}[d_H(X,Y)] = P(X \neq Y) = p_{01} + p_{10} = q_X + q_Y - 2p_{11}.$$

Thus, minimizing (resp. maximizing) $\Pr(X \neq Y)$ is equivalent to maximizing (resp. minimizing) $p_{11}$ subject to the Fréchet-Hoeffding bounds (Nelsen, 2006, Sec. 2.5): $\max\{0, q_X + q_Y - 1\} \leq p_{11} \leq \min\{q_X, q_Y\}$. The minimum distortion is attained at $p_{11} = \min\{q_X, q_Y\}$, realized by the monotone coupling. Specifically, let $U \sim \text{Unif}[0,1]$, with $X = \mathbf{1}\{U \leq q_X\}$ and $Y = \mathbf{1}\{U \leq q_Y\}$. Then, $D_{\min}^{(B)} = \min_{\text{couplings}} P(X \neq Y) = |q_X - q_Y|$. The maximum distortion is attained at $p_{11} = \max\{0, q_X + q_Y - 1\}$, realized by the antimonotone coupling: $X = \mathbf{1}\{U \leq q_X\}$ and $Y = \mathbf{1}\{U \geq 1 - q_Y\}$. Since $q_X, q_Y \leq \frac{1}{2}$, we obtain $D_{\max}^{(B)} = \max_{\text{couplings}} P(X \neq Y) = q_X + q_Y$.

For the independent coupling, $p_{11} = q_X q_Y$, yielding $D_{\text{ind}}^{(B)} = P(X \neq Y) = q_X(1 - q_Y) + (1 - q_X)q_Y = q_X + q_Y - 2q_X q_Y$. Building on these extremal couplings and benchmark distortions, we next derive the DRC tradeoff under common randomness.

**Theorem 2.** *Consider a Bernoulli source $X \sim \text{Bern}(q_X)$, $Y \sim \text{Bern}(q_Y)$, and a classification variable $S$ with the binary symmetric joint distribution given by $S = X \oplus S_1$ where $S \sim \text{Bern}(q_S)$ and $S_1 \sim \text{Bern}(q_{S_1})$ $(0 \leq q_X, q_S, q_{S_1} \leq \frac{1}{2})$. The problem (3) is feasible if $C \geq H_b(q_{S_1})$. Assume the Hamming distortion measure. Under common randomness, we have*

$$D^{(B)}(R,C,q_X,q_Y) = \begin{cases} \dfrac{-2(1-q_X)q_X(H_b(m)-C)}{H_b(m)-H_b(q_{S_1})} + D_{\text{ind}}^{(B)}, \\ \qquad H_b(q_{s_1}) \leq C \leq \frac{R(H_b(q_{S_1})-H_b(m))}{H_b(q_X)} + H_b(m) \\ \dfrac{-2(1-q_X)q_X R}{H_b(q_X)} + D_{\text{ind}}^{(B)}, \quad C > \frac{R(H_b(q_{S_1})-H_b(m))}{H_b(q_X)} + H_b(m) \\ D_{\min}^{(B)}, \qquad C > H_b(q_S) \text{ and } R > H_b(q_X). \end{cases}$$

*where $m = (1-q_X)(1-q_S_1) + q_X q_{S_1}$, $q_S = q_X + q_{S_1} - 2q_X q_{S_1}$ and $H_b(.)$ denotes the binary entropy function.*

*Proof.* The proof is provided in Appendix A.1.2. □

Theorem 2 reveals three regimes. With a loose classification constraint $C$, distortion decreases linearly with rate $R$, approaching the independent coding distortion $D_{\text{ind}}^{(B)}$. In the intermediate regime, distortion depends jointly on $R$ and $C$, capturing the tradeoff between compression efficiency and task fidelity. When $R$ exceeds the source entropy and $C$ is sufficiently large, the minimal distortion $D_{\text{min}}^{(B)}$ becomes achievable. Similarly, the definition and closed-form expression of $R^{(B)}(D, C, q_X, q_Y)$ are provided in Appendix A.2.

## 3 SYSTEM MODEL AND ASYMPTOTIC SETTING RESULTS

### 3.1 ASYMPTOTIC CONSTRAINED OPTIMAL TRANSPORT

Classical rate-distortion theory is usually considered in the asymptotic block-length regime, where arbitrarily long i.i.d. sequences are compressed and coding theorems yield single-letter characterizations. Motivated by this, we now extend our one-shot constrained optimal transport formulation to the asymptotic case, where large block lengths are allowed. This generalization allows us to connect to Shannon's original setting and to establish information-theoretic characterizations that hold in the limit. Let $\{X_i\}_{i=1}^{\infty}$, $\{Y_i\}_{i=1}^{\infty}$, and $\{S_i\}_{i=1}^{\infty}$ be i.i.d. processes with marginals $p_X$, $p_Y$, and $p_S$, respectively.

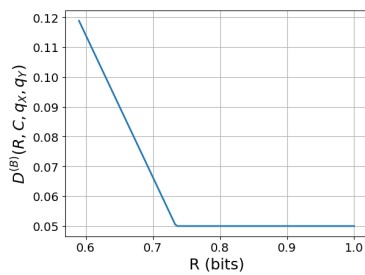

Figure 3: $D^{(B)}(R, C, q_X, q_Y)$ versus $R$ with $C = 0.8$, $q_X = 0.3$, $q_Y = 0.25$, $q_{S_1} = 0.2$.

**Definition 3** (Asymptotic Constrained Optimal Transport).
*Consider i.i.d. random variables $X_i \sim p_X$, $Y_i \sim p_Y$, and $S_i \sim p_S$. The asymptotic constrained optimal transport problem with rate constraint $R$, classification loss $C$, and shared randomness $U$ in the asymptotic regime ($n \to \infty$) is defined as*

$$D^{(\infty)}(R, C, p_X, p_Y) = \inf_{p_{U,X^n,Z,Y^n} \in M(\otimes_{i=1}^n p_X, \otimes_{i=1}^n p_Y)} \frac{1}{n} \sum_{i=1}^n \mathbb{E}[d(X_i, Y_i)]$$

$$s.t. \quad \frac{1}{n} H(Z|U) \le R, \quad \frac{1}{n} \sum_{i=1}^n H(S_i|Y_i) \le C.$$

**Theorem 3.** *In the asymptotic regime, the DRC function admits the single-letter characterization*

$$D^{(\infty)}(R, C, p_X, p_Y) = \inf_{p_{X,Y} \in \Gamma(p_X, p_Y)} \mathbb{E}[d(X, Y)] \tag{4}$$

$$s.t. \quad I(X;Y) \le R, \quad H(S|Y) \le C.$$

*Proof.* The result follows from Saldi et al. (2015b, Theorem 7), combined with the arguments in Wang et al. (2025, Appendix F). □

**Remark 1.** *As in the one-shot formulation, shared common randomness $U$ can be leveraged in the asymptotic regime for the constrained optimal transport problem. However, since coding theorems in the block-length limit already allow randomized mappings without rate penalty, the asymptotic characterization in Theorem 3 coincides with the one-shot formulation, and the role of $U$ does not further tighten the bound.*

### 3.2 GAUSSIAN CASE EXPRESSIONS

We now investigate the constrained optimal transport framework for Gaussian sources under MSE distortion. Let $X \sim \mathcal{N}(\mu_X, \sigma_X^2)$ and $Y \sim \mathcal{N}(\mu_Y, \sigma_Y^2)$ be Gaussian random variables, and let $S \sim \mathcal{N}(\mu_S, \sigma_S^2)$ denote the associated classification variable with $\theta_1 \triangleq \text{Cov}(X, S)$. In the Gaussian case, we derive a single-letter characterization of the asymptotic DRC tradeoffs with shared randomness.

**Theorem 4.** *Consider $X \sim \mathcal{N}(\mu_X, \sigma_X^2)$ and $Y \sim \mathcal{N}(\mu_Y, \sigma_Y^2)$ with MSE distortion, and let $S \sim \mathcal{N}(\mu_S, \sigma_S^2)$ be a classification variable with $\text{Cov}(X, S) = \theta_1$. The problem (4) is feasible if $C \ge$*

$\frac{1}{2}\log\left(1 - \frac{\theta_1^2}{\sigma_S^2 \sigma_X^2}\right) + h(S)$. *Under shared randomness, the asymptotic DRC tradeoff is*

$$D^{(G)}(R, C, q_X, q_Y) = \begin{cases} (\mu_X - \mu_Y)^2 + \sigma_X^2 + \sigma_Y^2 - \frac{2\sigma_S \sigma_X^2 \sigma_Y}{\theta_1}\sqrt{1 - e^{-2h(S)+2C}}, \\ \quad \frac{1}{2}\log\left(1 - \frac{\theta_1^2}{\sigma_S^2 \sigma_X^2}\right) + h(S) \leq C \leq \frac{1}{2}\log\left(1 - \frac{\theta_1^2(1-2^{-2R})}{\sigma_S^2 \sigma_X^2}\right) + h(S), \\ (\mu_X - \mu_Y)^2 + \sigma_X^2 + \sigma_Y^2 - 2\sigma_X \sigma_Y \sqrt{1 - 2^{-2R}}, \\ \quad C > \frac{1}{2}\log\left(1 - \frac{\theta_1^2(1-2^{-2R})}{\sigma_S^2 \sigma_X^2}\right) + h(S), \\ 0, \quad C > h(S) \ \text{and} \ R > h(X). \end{cases}$$

*Proof.* A detailed proof is given in Appendix A.1.3. □

We provide the definition and closed-form expression of $R^{(G)}(D, C, q_X, q_Y)$ in Appendix A.3.

### 3.3 ASYMPTOTIC DRPC FUNCTION FOR GAUSSIAN SOURCES

Classical asymptotic RD analysis yields single-letter characterizations in the block-length limit. We extend this perspective to the DRPC setting, where reconstructions are required not only to satisfy fidelity and rate constraints but also to preserve classification performance (Wang et al., 2025) and align with a perceptual target distribution (Blau & Michaeli, 2019; Theis & Wagner, 2021).

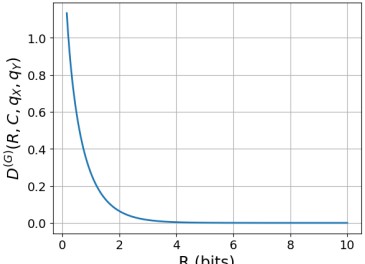

Figure 4: $D^{(G)}(R, C, q_X, q_Y)$ versus $R$ with $C = 2$, $X, Y, S \sim \mathcal{N}(0, 1)$, $\theta_1 = 0.6$.

**Definition 4** (Asymptotic DRPC Function). *For i.i.d. random variables $X_i \sim p_X$, $Y_i \sim p_Y$, and $S_i \sim p_S$, the DRPC function with common randomness in the asymptotic regime is defined as*

$$D^{(\infty)}(R, P, C) = \inf_{p_{U, X^n, Z, Y^n}} \frac{1}{n}\sum_{i=1}^{n} \mathbb{E}[d(X_i, Y_i)]$$

$$s.t. \quad \frac{1}{n}H(Z|U) \leq R, \quad \frac{1}{n}\sum_{i=1}^{n} H(S_i|Y_i) \leq C, \quad \frac{1}{n}\sum_{i=1}^{n} \phi(p_{X_i}, p_{Y_i}) \leq P.$$

*where $\phi(\cdot, \cdot)$ is a nonnegative divergence capturing perceptual quality.*

**Theorem 5.** *In the asymptotic regime, the DRPC function admits the single-letter characterization*

$$D^{(\infty)}(R, P, C) = \inf_{p_{Y|X}} \mathbb{E}[d(X, Y)]$$

$$s.t. \quad I(X; Y) \leq R, \quad H(S|Y) \leq C, \quad \phi(p_X, p_Y) \leq P.$$

*Proof.* The result follows the asymptotic analysis in Theis & Wagner (2021); Saldi et al. (2015a) and adapting the arguments in Wang et al. (2025, Appendix F). □

We investigate two perception divergences of particular interest. The first is the Kullback-Leibler divergence, defined as $\phi(p_X, p_Y) = \phi_{KL}(p_Y \| p_X) = \mathbb{E}\left[\log \frac{p_Y(Y)}{p_X(Y)}\right]$. The second divergence we consider is the squared quadratic Wasserstein distance, defined as $W_2^2(p_X, p_Y) = \inf_{p_{XY} \in \Gamma(p_X, p_Y)} \mathbb{E}[(X - Y)^2]$.

Since the source variables $\{X_i\}$ are i.i.d. and so are the reconstructions $\{Y_i\}$, the divergence term $\phi(p_{X_i}, p_{Y_i})$ is independent of $i$. Thus, DRPC coding can be viewed as output-constrained source coding, where the reconstruction distribution is restricted to the set $\{p_Y : \phi(p_X, p_Y) \leq P\}$. Accordingly, the DRPC function is given by

$$D^{(\infty)}(R, P, C) = \inf_{p_Y : \phi(p_X, p_Y) \leq P} D^{(\infty)}(R, C, p_X, p_Y).$$

Unlike $D^{(\infty)}(R, C, p_X, p_Y)$, the reconstruction distribution in $D^{(\infty)}(R, P, C)$ is not fixed but only required to satisfy the perceptual constraint $\phi(p_X, p_Y) \leq P$. Leveraging the jointly Gaussian structure, we obtain closed-form characterizations of the DRPC tradeoff under MSE distortion, subject to both classification and perception constraints. In particular, explicit expressions are derived when the perception measure is chosen as either the KL divergence or the quadratic Wasserstein distance.

**Theorem 6.** *Let $X \sim \mathcal{N}(\mu_X, \sigma_X^2)$ be a Gaussian source and $S \sim \mathcal{N}(\mu_S, \sigma_S^2)$ a classification variable jointly Gaussian with $X$, such that $\mathrm{Cov}(X, S) = \theta_1$. Consider $Y$ with mean $\mathbb{E}[Y] = \mu_Y$, variance $\mathrm{Var}(Y) = \sigma_Y^2$, and covariance $\mathrm{Cov}(X, Y) = \theta_2$. Define $Y_G$ as a Gaussian random variable such that $(X, Y_G)$ is jointly Gaussian with the same first and second moments as $(X, Y)$: $\mathbb{E}[Y_G] = \mu_Y$, $\mathrm{Var}(Y_G) = \sigma_Y^2$, and $\mathrm{Cov}(X, Y_G) = \theta_2$. Under the MSE distortion with constraints $I(X; Y) \leq R$, $h(S|Y) \leq C$, and $\phi(q_X, q_Y) \leq P$, the function $D^{(\infty)}(R, P, C)$ is attained by such a jointly Gaussian $Y_G$ when the perception measure is either $W_2^2(q_X, q_Y)$ or $\phi_{KL}(q_Y \| q_X)$.*

*Proof.* The proof is provided in Appendix A.1.4. $\qquad\square$

**Theorem 7.** *Let $X \sim \mathcal{N}(\mu_X, \sigma_X^2)$ and $Y \sim \mathcal{N}(\mu_Y, \sigma_Y^2)$ be two Gaussian random variables. Let $S \sim \mathcal{N}(\mu_S, \sigma_S^2)$ be an associated classification variable with a covariance of $Cov(X, S) = \theta_1$ and be jointly Gaussian. For the case $d(X, Y) = (X - Y)^2$ and $\phi(p_X, p_Y) = \phi_{KL}(p_Y \| p_X)$, we have*

$$D_{KL}^{(G)}(R, P, C)$$

$$= \begin{cases} \sigma_X^2 - \sigma_X^2(1 - 2^{-2R}), \sigma(P) \leq \sigma_X \sqrt{1 - 2^{-2R}} \text{ and } C > \dfrac{1}{2}\log\left(1 - \dfrac{\theta_1^2(1 - 2^{-2R})}{\sigma_S^2 \sigma_X^2}\right) + h(S) \\[2mm] \sigma_X^2 + \sigma^2(P) - 2\sigma_X \sigma(P)\sqrt{1 - 2^{-2R}}, \\[1mm] \qquad \sigma(P) > \sigma_X \sqrt{1 - 2^{-2R}} \text{ and } C > \frac{1}{2}\log\left(1 - \frac{\theta_1^2(1 - 2^{-2R})}{\sigma_S^2 \sigma_X^2}\right) + h(S) \\[2mm] \sigma_X^2 - \dfrac{\sigma_S^2 \sigma_X^4}{\theta_1^2}(1 - 2^{-2h(S)+2C}), \sigma(P) \leq \dfrac{\sigma_S \sigma_X^2}{\theta_1}\sqrt{1 - 2^{-2h(S)+2C}} \\[1mm] \qquad \text{and } \dfrac{1}{2}\log\left(1 - \dfrac{\theta_1^2}{\sigma_S^2 \sigma_X^2}\right) + h(S) \leq C \leq \dfrac{1}{2}\log\left(1 - \dfrac{\theta_1^2(\sigma_X^2 - \sigma_X^2 2^{-2R})}{\sigma_S^2 \sigma_X^4}\right) + h(S) \\[2mm] \sigma_X^2 + \sigma^2(P) - \dfrac{2\sigma_S \sigma_X^2 \sigma(P)}{\theta_1}\sqrt{1 - 2^{-2h(S)+2C}}, \sigma(P) > \dfrac{\sigma_S \sigma_X^2}{\theta_1}\sqrt{1 - 2^{-2h(S)+2C}} \\[1mm] \qquad \text{and } \dfrac{1}{2}\log\left(1 - \dfrac{\theta_1^2}{\sigma_S^2 \sigma_X^2}\right) + h(S) \leq C \leq \dfrac{1}{2}\log\left(1 - \dfrac{\theta_1^2(\sigma_X^2 - \sigma_X^2 2^{-2R})}{\sigma_S^2 \sigma_X^4}\right) + h(S) \\[2mm] 0, \quad C > h(S) \text{ and } R > h(X). \end{cases}$$

*where $\sigma(P)$ is the unique number $\sigma \in [0, \sigma_X]$ satisfying $\psi(\sigma) = P$ and $\psi(\sigma_Y) = \log\frac{\sigma_X}{\sigma_Y} + \frac{\sigma_Y^2 - \sigma_X^2}{2\sigma_X^2}$.*

*Proof.* A complete proof is given in Appendix A.1.5. $\qquad\square$

A detailed derivation of the closed-form expression for $D^{(\infty)}(R, P, C)$ under the quadratic Wasserstein distance can be found in Appendix A.4.

## 4 RELATED WORKS

Classical rate-distortion theory characterizes the fundamental limits of lossy compression with the best achievable distortion at a given rate (Cover & Thomas, 1999), while the information bottleneck links compression with task relevance (Chechik et al., 2003). Task-aware extensions, CDP (Liu et al., 2019a;b) and the RDC/RDPC formulations (Zhang, 2023; Wang et al., 2025; Nguyen et al., 2025; 2026), make explicit how accuracy constraints reshape the RD function, yet they typically operate within a single domain. Perception-aware RD augments RD tradeoff with a divergence between source and reconstructions (Blau & Michaeli, 2018; 2019; Theis & Wagner, 2021), inspiring generative codecs based on adversarial learning and distribution-preserving objectives (Goodfellow et al., 2014; Arjovsky et al., 2017; Gulrajani et al., 2017; Tschannen et al., 2018; Agustsson et al., 2019; Mentzer et al., 2020). Modern learned codecs pair analysis-synthesis transforms with entropy models and tighter rate estimation (Ballé et al., 2017; 2018; Minnen et al., 2018; Theis et al., 2017; Williams et al., 2020; Johnston et al., 2018; Agustsson et al., 2017; Mentzer et al., 2018; Wu et al.,

2020; Alemi et al., 2018; Brekelmans et al., 2019; Huang et al., 2020; Park et al., 2020). Beyond fidelity, compression has been used as a denoising prior from asymptotic limits (Weissman & Ordentlich, 2005) to recent neural and zero-shot frameworks with theoretical guarantees (Zafari et al., 2025a;b). Optimal transport provides a principled way to couple marginals (Villani, 2009) and has informed unsupervised restoration (Wang et al., 2023a). Most relevant to our work, cross-domain lossy compression has been cast as entropy-constrained OT with shared randomness that decouples coding and transport (Liu et al., 2022). Unsupervised image restoration has also been explored in (Zhang et al., 2017; Menon et al., 2020; Pan et al., 2021) with a fixed reconstruction distribution, but these approaches neither impose compression constraints nor incorporate classification-awareness for downstream tasks. Common randomness and stochastic encoders, realized via universal/dithered quantization, enable output constraints and "free" synthesis randomness (Saldi et al., 2015a; 2013; Schuchman, 1964; Gray & Stockham, 1993; Ziv, 1985; Li & El Gamal, 2018a; Theis & Agustsson, 2021). Our work extends this line by introducing rate- and classification-constrained optimal transport, providing one-shot and asymptotic characterizations with closed-form tradeoffs, and validating the theory through deep generative compression models.

## 5 EXPERIMENTAL RESULTS

### 5.1 TRAINING SETUP

We consider the setting where the encoder observes degraded samples $X \sim p_X$ (e.g., noisy or low-resolution) and the goal is to reconstruct outputs from a distinct target distribution $Y \sim p_Y$ (e.g., clean or high-resolution). The objective is to compress $X$ while ensuring that reconstructions preserve semantic content, align with $p_Y$, and remain predictive of the downstream label $S$. Note that $Y$ is drawn from the clean dataset distribution, but does not correspond to the exact clean counterpart of $X$. In this unsupervised setting, only unpaired noisy and clean samples are available.

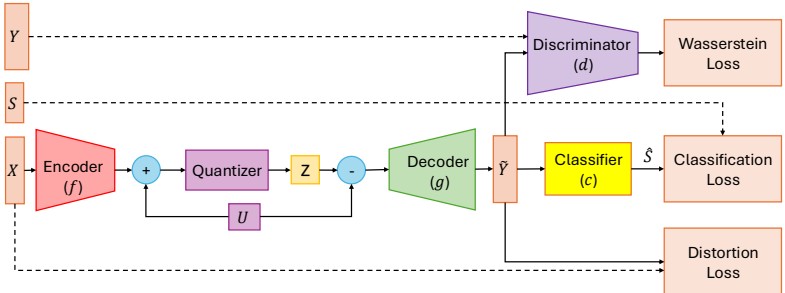

Figure 5: Experimental architecture: a stochastic autoencoder with classifier and WGAN discriminator, conditioned on shared randomness $U$.

Following Liu et al. (2022); Wang et al. (2025), we adopt a stochastic autoencoder consisting of an encoder $f$, quantizer $Q$, decoder $g$, classifier $c$, and WGAN discriminator $d$. Distortion is measured by MSE, while classification is enforced via cross-entropy loss $\mathrm{CE}(S, \hat{S})$, which upper bounds the conditional entropy $H(S|Y)$ (Boudiaf et al., 2021; Wang et al., 2025). The rate is upper bounded by $h \log_2 L$, where $h$ is the encoder output dimension and $L$ the quantization level. Formally, for a target rate $R$ and shared randomness $U$, the system solves

$$\min_{f,g,Q} \quad \mathbb{E}\big[\|X - g(Q(f(X,U)))\|_2^2\big]$$

$$\text{s.t.} \quad p_{g(Q(f(X,U)))} = p_Y, \quad H(Q(f(X,U))) \leq R, \quad H(S|g(Q(f(X,U)))) \leq C.$$

Letting $\tilde{Y} = g(Q(f(X,U)))$, the WGAN discriminator aligns $p_{\tilde{Y}}$ with $p_Y$ via a Wasserstein-1 penalty (Arjovsky et al., 2017). Shared randomness is implemented through universal quantization (Ziv, 1985; Theis & Agustsson, 2021). With trained encoder $f$ and decoder $g$, restoration is obtained as $\tilde{Y} = g(Q(f(X) + U) - U)$, where $U$ is common randomness available to both encoder and decoder. In practice, we optimize the relaxed loss

$$\mathcal{L} = \mathbb{E}[\|X - \tilde{Y}\|^2] + \lambda_p W_1(p_Y, p_{\tilde{Y}}) + \lambda_c \mathrm{CE}(S, \hat{S}),$$

which balances fidelity, distributional alignment, and classification.

## 5.2 RESULTS

Figure 6 presents the tradeoffs between rate, distortion, and accuracy. As expected, higher rates yield lower MSE and improved classification performance. We report both quantitative curves and qualitative results for super-resolution (MNIST) and denoising (SVHN). At low rates (e.g., $R = 4$ in Figure 6(c)), reconstructions capture coarse structure but remain blurry, stylized, or even ambiguous. At higher rates (e.g., $R = 32$), both distortion and perceptual quality improve substantially, producing reconstructions that closely match the high-resolution targets. A similar trend holds for denoising, where noisy digits progressively sharpen and become recognizable as the rate increases.

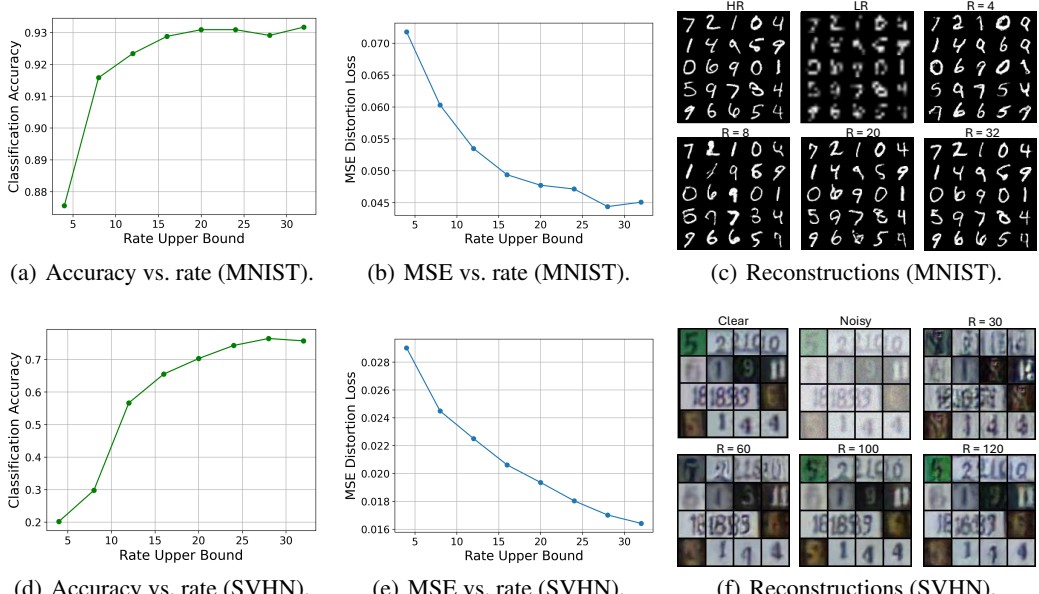

(a) Accuracy vs. rate (MNIST).  (b) MSE vs. rate (MNIST).  (c) Reconstructions (MNIST).

(d) Accuracy vs. rate (SVHN).  (e) MSE vs. rate (SVHN).  (f) Reconstructions (SVHN).

Figure 6: Experimental results: $4\times$ super-resolution on MNIST and denoising on SVHN with Gaussian noise with $\sigma = 20$. Higher rates yield clearer reconstructions and improved classification performance.

Additional results on entropy model-based rate estimation for super-resolution (MNIST) and denoising (SVHN, CIFAR-10, ImageNet, KODAK) are provided in Appendix B.1. We further examine the inpainting problem on SVHN under both supervised and unsupervised settings in Appendix B.2. Together, these experiments enrich the empirical study of the RDC tradeoff and provide additional evidence that the observed behaviors align closely with the theoretical predictions of our framework.

## 6 CONCLUSION

We studied cross-domain lossy compression, where the decoder reconstructs samples from a target distribution distinct from the degraded source observed by the encoder. By casting the problem as constrained optimal transport, compression rate and classification loss were unified into a single information-theoretic framework. In the one-shot regime with shared randomness, the problem reduces to a deterministic transport plan, and we derived closed-form DRC/RDC expressions for Bernoulli sources under Hamming distortion. In the asymptotic regime, analytic DRC/RDC tradeoffs were obtained for Gaussian sources under MSE. The framework was further extended to include perception divergences, such as KL and quadratic Wasserstein, leading to closed-form DRPC functions. To bridge theory and practice, we implemented deep end-to-end compression frameworks incorporating universal quantization for shared common randomness, entropy modeling for rate estimation, adversarial distribution alignment, and a task-specific classifier. Experiments on super-resolution (MNIST), denoising (SVHN, CIFAR-10, ImageNet, KODAK), and inpainting (SVHN) confirmed that empirical performance closely matches the theoretical predictions.

ACKNOWLEDGMENT

This work was supported by the National Science Foundation under Grant No. CCF:SHF:2417898.

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

# Supplementary Material of "Cross-Domain Lossy Compression via Rate- and Classification-Constrained Optimal Transport"

## A  THEORETICAL RESULTS

### A.1  PROOFS OF THEORETICAL RESULTS IN THE MAIN PAPER

#### A.1.1  PROOF OF THEOREM 1

**Theorem 1.** *Define $Q(p_X, p_Y)$ as the set of joint distributions $p_{U,X,Y}$ with marginals $p_X, p_Y$ that factorize as $p_{U,X,Y} = p_U \, p_X \, p_{Y|X,U}$. Then, the constrained optimal transport cost in Definition 2 admits the representation*

$$D(R, C, p_X, p_Y) = \inf_{p_{U,X,Y} \in Q(p_X, p_Y)} \mathbb{E}[d(X, Y)]$$
$$\text{s.t.} \quad H(Y|X, U) = 0, \quad I(X; U) = 0,$$
$$H(Y|U) \leq R, \quad H(S|Y) \leq C.$$

*Proof.* Recall the formulation from Definition 2:

$$D(R, C, p_X, p_Y) = \inf_{p_{U,X,Z,Y} \in M(p_X, p_Y)} \mathbb{E}[d(X, Y)]$$
$$\text{s.t.} \quad H(Z|U) \leq R,$$
$$H(S|Y) \leq C.$$

where $M(p_X, p_Y) = \{p_{U,X,Z,Y} : p_{U,X,Z,Y} = p_U \, p_X \, p_{Z|X,U} \, p_{Y|Z,U}\}$. Let $Q(p_X, p_Y) = \{p_{U,X,Y} : p_{U,X,Y} = p_U \, p_X \, p_{Y|U,X}, \; H(Y|U,X) = 0\}$.

**Upper bound.** Fix any $p_{U,X,Y} \in Q(p_X, p_Y)$ that satisfies $H(Y|U) \leq R$ and $H(S|Y) \leq C$, and set $Z \triangleq Y$. Then $p_{U,X,Z,Y} = p_U \, p_X \, \delta_{Z=Y(X,U)} \, \delta_{Y=Z} \in M(p_X, p_Y)$, with $H(Z|U) = H(Y|U) \leq R$ and the same $\mathbb{E}[d(X,Y)]$ and $H(S|Y)$. Hence

$$D(R, C, p_X, p_Y) \leq \inf_{p_{U,X,Y} \in Q(p_X, p_Y)} \mathbb{E}[d(X, Y)]$$
$$\text{s.t.} \quad H(Y|U, X) = 0, \quad I(X; U) = 0,$$
$$H(Y|U) \leq R, \quad H(S|Y) \leq C.$$

**Tightness.** Take any feasible $p_{U,X,Z,Y} \in M(p_X, p_Y)$ with $H(Z|U) \leq R$ and $H(S|Y) \leq C$. By the functional representation lemma (El Gamal & Kim, 2011; Li & El Gamal, 2018b), there exist: (i) a random seed $V_1$ independent of $(U, X)$ and a measurable $\phi_1$ such that $Z = \phi_1(U, X, V_1)$ in distribution (for $p_{Z|X,U}$); (ii) a random seed $V_2$ independent of $(U, X, V_1)$ and a measurable $\phi_2$ such that $Y = \phi_2(U, Z, V_2)$ in distribution (for $p_{Y|Z,U}$).

Let $U' \triangleq (U, V_1, V_2)$. Then, $Y = \phi_2(U, \phi_1(U, X, V_1), V_2)$ is deterministic given $(U', X)$, so $H(Y|U', X) = 0$ and $(U', X, Y) \in Q(p_X, p_Y)$. The marginal $(X, Y)$ is preserved, hence $\mathbb{E}[d(X, Y)]$ and $H(S|Y)$ are unchanged.

For the rate term, conditioning reduces entropy, and determinism gives

$$H(Z|U) \geq H(Z|U, V_1, V_2) = H(Z|U') \geq H(Y|U'),$$

and $H(Y|U') \leq R$. Therefore,

$$D(R, C, p_X, p_Y) \geq \inf_{p_{U',X,Y} \in Q(p_X, p_Y)} \mathbb{E}[d(X, Y)]$$
$$\text{s.t.} \quad H(Y|U', X) = 0, \quad I(X; U') = 0,$$
$$H(Y|U') \leq R, \quad H(S|Y) \leq C.$$

Since the auxiliary alphabet is unrestricted, we can relabel $U'$ as $U$ inside the infimum. Combining the two bounds, the proof is completed. $\square$

### A.1.2 PROOF OF THEOREM 2

**Theorem 2.** *Consider a Bernoulli source $X \sim Bern(q_X)$, $Y \sim Bern(q_Y)$, and a classification variable $S$ with the binary symmetric joint distribution given by $S = X \oplus S_1$ where $S \sim Bern(q_S)$ and $S_1 \sim Bern(q_{S_1})$ ($0 \leq q_X, q_S, q_{S_1} \leq \frac{1}{2}$). The problem (3) is feasible if $C \geq H_b(q_{S_1})$. Assume the Hamming distortion measure. Under common randomness, we have*

$$
D^{(B)}(R,C,q_X,q_Y) = \begin{cases}
\dfrac{-2(1-q_X)q_X(H_b(m)-C)}{H_b(m)-H_b(q_{S_1})} + D^{(B)}_{\text{ind}}, \\
\qquad H_b(q_{s_1}) \leq C \leq \frac{R(H_b(q_{S_1})-H_b(m))}{H_b(q_X)} + H_b(m) \\
\dfrac{-2(1-q_X)q_X R}{H_b(q_X)} + D^{(B)}_{\text{ind}}, \quad C > \frac{R(H_b(q_{S_1})-H_b(m))}{H_b(q_X)} + H_b(m) \\
D^{(B)}_{\text{min}}, \qquad C > H_b(q_S) \text{ and } R > H_b(q_X).
\end{cases}
$$

*where $m = (1-q_X)(1-q_{S_1}) + q_X q_{S_1}$, $q_S = q_X + q_{S_1} - 2q_X q_{S_1}$, $H_b(q_S) = H_b(m)$, and $H_b(.)$ denotes the binary entropy function.*

*Proof.* Since $H(Y|U) = I(X;Y|U) + H(Y|U,X) = I(X;Y|U)$ and $H(Y|U,X) = 0$, we can equivalently view $Y$ as a deterministic function of $(X,U)$, i.e., $Y = f(X,U)$. The problem (3), under the Hamming distortion measure, reduces to finding a distribution $p_U$ such that

$$D^{(B)}(R,C,q_X,q_Y) = \inf_{p_U} \quad P(X \neq Y)$$

$$\text{s.t.} \quad H(Y|U,X) = 0, \quad I(X;U) = 0,$$
$$I(X;Y|U) \leq R, \quad H(S|Y) \leq C.$$

Because Shannon entropy is defined only for discrete random variables, the auxiliary variable $U$ must be chosen such that $Y|U = u$ is discrete for each $u$, even in continuous $(X,Y)$ settings (Liu et al., 2022). Here, $p_U$ is supported on $\mathcal{U} \triangleq \{1, 2, \ldots, |\mathcal{Y}|^{|\mathcal{X}|}\}$ and $\{f_u : u \in \mathcal{U}\}$ denotes the set of all distinct mappings $f_u : \mathcal{X} \to \mathcal{Y}$. By the support lemma (Appendix C, p. 631 of El Gamal & Kim, 2011), it suffices to assign positive probability to at most $|\mathcal{Y}| + 1$ such functions.

The optimization can be expressed as follows

$$D^{(B)}(R,C,q_X,q_Y) = \min_{p_U} \sum_{u \in \mathcal{U}} p_U(u) P(X \neq Y|U = u)$$

$$\text{s.t.} \quad \sum_{u \in \mathcal{U}} p_U(u) I(X;Y|U = u) \leq R,$$

$$\sum_{u \in \mathcal{U}} p_U(u) P(f_u(X) = y) = q_Y, \quad \forall y \in \mathcal{Y},$$

$$\sum_{u \in \mathcal{U}} p_U(u) H(S|f_u(X)) \leq C.$$

Without loss of optimality, the size of the alphabet of $U$ can be restricted to at most four. There are exactly four distinct mappings from $\{0,1\}$ to $\{0,1\}$: $f_1(x) = x$, $f_2(x) = 1 - x$, $f_3(x) = 0$, and $f_4(x) = 1$ with $x \in \{0,1\}$. Hence, $Y = X$ if $U = 1$; $Y = 1 - X$ if $U = 2$; $Y = 0$ if $U = 3$; and $Y = 1$ if $U = 4$. Therefore, we obtain

$$P(X \neq Y) = \sum_{u \in \mathcal{U}} p_U(u) P(X \neq Y|U = u) = p_U(2) + q_X p_U(3) + (1 - q_X) p_U(4),$$

$$I(X;Y|U) = \sum_{u \in \mathcal{U}} p_U(u) I(X;Y|U = u) = \sum_{u \in \mathcal{U}} p_U(u) H(f_u(X)) = H_b(q_X)(p_U(1) + p_U(2)),$$

$$P(Y = 1) = q_Y = \sum_{u \in \mathcal{U}} p_U(u) P(f_u(X) = y) = p_U(1) q_X + (1 - q_X) p_U(2) + p_U(4).$$

By the data-processing inequality (Cover & Thomas, 1999) and the Markov relation $S \leftrightarrow X \leftrightarrow Y$,

$$H(S|Y) \geq H(S|X) = H(X \oplus S_1|X) = H(S_1) = H_b(q_{S_1}).$$

The classification constraint is feasible only if $C \geq H(S_1)$.

The evaluation of $H(S|Y)$ for each mapping is derived as follows

- For $U = 1$: $H(S|Y, U = 1) = H(S|X) = H_b(q_{S_1})$.

- For $U = 2$: $H(S|Y, U = 2) = H(S|X) = H_b(q_{S_1})$.

- For $U = 3$: $P(S = 0) = (1 - q_X)(1 - q_{S_1}) + q_X q_{S_1}$,
$$\Rightarrow H(S|Y, U = 3) = H(S|U = 3) = H_b((1 - q_X)(1 - q_{S_1}) + q_X q_{S_1}).$$

- For $U = 4$: $H(S|Y, U = 4) = H(S|U = 4) = H_b((1 - q_X)(1 - q_{S_1}) + q_X q_{S_1})$.

Hence,
$$H(S|Y) = \sum_{u \in \mathcal{U}} p_U(u) \, H(S|f_u(X))$$
$$= (p_U(1) + p_U(2))H_b(q_{S_1}) + (p_U(3) + p_U(4))H_b((1 - q_X)(1 - q_{S_1}) + q_X q_{S_1}).$$

Let $m = (1 - q_X)(1 - q_{S_1}) + q_X q_{S_1}$, we have
$$H(S|Y) = (p_U(1) + p_U(2))H_b(q_{S_1}) + (p_U(3) + p_U(4))H_b(m).$$

The final optimization problem is represented as

$$D^{(B)}(R, C, q_X, q_Y) = \min_{p_U(1),\, p_U(2),\, p_U(3),\, p_U(4)} p_U(2) + q_X p_U(3) + (1 - q_X)p_U(4) \tag{5}$$

$$\text{s.t.} \quad H_b(q_X)\,(p_U(1) + p_U(2)) \leq R, \tag{6}$$
$$q_X p_U(1) + (1 - q_X)p_U(2) + p_U(4) = q_Y, \tag{7}$$
$$(p_U(1) + p_U(2))H_b(q_{S_1}) + (p_U(3) + p_U(4))H_b(m) \leq C, \tag{8}$$
$$p_U(1) + p_U(2) + p_U(3) + p_U(4) = 1, \tag{9}$$
$$p_U(1),\, p_U(2),\, p_U(3),\, p_U(4) \geq 0. \tag{10}$$

**The activity of the nonnegative constraints.** From (9) and (7), we have
$$p_U(3) = 1 - [p_U(1) + p_U(2)] - p_U(4) \text{ and } p_U(4) = q_Y - q_X p_U(1) - (1 - q_X)p_U(2).$$

The objective is: $\mathbb{E}[d_H(X, Y)] = D_{\text{ind}}^{(B)} - 2(1 - q_X)q_X p_U(1) + 2(1 - q_X)q_X\, p_U(2)$.

For $0 \leq q_X \leq \frac{1}{2}$, the distortion $\mathbb{E}[d_H(X, Y)]$ is strictly increasing in $p_U(2)$ and decreasing in $p_U(1)$; hence it is optimal to take $p_U(2) = 0$ and $p_U(1) > 0$. We can also write
$$p_U(4) = q_Y - q_X[p_U(1) + p_U(2)] + (2q_X - 1)\, p_U(2),$$
$$p_U(3) = (1 - q_Y) - (1 - q_X)[p_U(1) + p_U(2)] - (2q_X - 1)\, p_U(2).$$

With $p_U(2) = 0$, it follows that
$$p_U(4) = q_Y - q_X p_U(1) \geq 0 \Rightarrow p_U(1) \leq \frac{q_Y}{q_X},$$
$$p_U(3) = 1 - q_Y - (1 - q_X)p_U(1) \geq 0 \Rightarrow 0 < p_U(1) \leq \frac{1 - q_y}{1 - q_X}.$$

We state a supporting fact that will be used repeatedly in the arguments as follows.

**Lemma A.1.** *We have,*
$$H_b(m) \geq H_b(q_{S_1}),$$
*with equality if only if $q_X \in \{0, 1\}$ or $q_{S_1} = \frac{1}{2}$.*

*Proof of Lemma A.1.* The identity for $m$ is immediate:
$$m = (1 - q_X)(1 - q_{S_1}) + q_X q_{S_1} = \tfrac{1}{2} + (q_X - \tfrac{1}{2})(2q_{S_1} - 1),$$
$$m - \tfrac{1}{2} = (q_X - \tfrac{1}{2})(2q_{S_1} - 1) \Rightarrow |m - \tfrac{1}{2}| = 2|q_X - \tfrac{1}{2}||q_{S_1} - \tfrac{1}{2}| \leq |q_{S_1} - \tfrac{1}{2}|$$
since $|q_X - \tfrac{1}{2}| \leq \tfrac{1}{2}$ for $q_X \in [0, 1]$. The binary entropy is maximized at $\frac{1}{2}$ and strictly decreases with $|p - \tfrac{1}{2}|$, hence $H_b(m) \geq H_b(q_{S_1})$, with equality if only if $|q_X - \tfrac{1}{2}| = \tfrac{1}{2}$ (i.e., $q_X \in \{0, 1\}$) or $|q_{S_1} - \tfrac{1}{2}| = 0$ (i.e., $q_{S_1} = \frac{1}{2}$). $\qquad \square$

We will invoke Lemma A.1 in the analysis below to justify the sign of denominators of the form $H_b(m) - H_b(q_{S_1})$. Note that,

$$p_U(2) + q_X p_U(3) + (1 - q_X)p_U(4) = -(1 - q_X)p_U(1) + q_X p_U(2) + (2q_X - 1)p_U(3) + (1 - q_X).$$

Combining constraints (7) and (9), we obtain $p_U(3) = -(1 - q_X)p_U(1) - q_X p_U(2) + (1 - q_Y)$.

Since problem (5) is a linear program, it can be solved efficiently using standard convex optimization tools such as CVX in MATLAB (CVX Research, Inc., 2012; Grant & Boyd, 2008) or CVXPY in PYTHON (Diamond & Boyd, 2016; Agrawal et al., 2018). Alternatively, an analytical solution can be derived via the Karush-Kuhn-Tucker (KKT) conditions. In practice, our approach systematically explores all possible combinations of active and inactive rate and classification constraints to fully characterize the optimal solution.

**Case 1.** Constraint (6) is active and constraint (8) is inactive.

Using $p_U(2) \geq 0$ and the fact that (6) is active, we obtain

$$R = H_b(q_X)(p_U(1) + p_U(2)) \geq H_b(q_X)\, p_U(1) \Rightarrow p_U(1) \leq \frac{R}{H_b(q_X)}.$$

Moreover,

$$p_U(2) + q_X p_U(3) + (1 - q_X)p_U(4) = -2(1 - q_X)q_X(p_U(1) + p_U(2)) + 4(1 - q_X)q_X p_U(2) + D_{\text{ind}}^{(B)}$$
$$\geq -2(1 - q_X)q_X(p_U(1) + p_U(2)) + D_{\text{ind}}^{(B)}$$
$$\geq \frac{-2(1 - q_X)q_X R}{H_b(q_X)} + D_{\text{ind}}^{(B)}.$$

Thus, $D^{(B)}(R, C, q_X, q_Y) = \frac{-2(1 - q_X)q_X R}{H_b(q_X)} + D_{\text{ind}}^{(B)}$.

This lower bound is tight, achieved by

$$p_U^\star(1) = \frac{R}{H_b(q_X)}, \qquad\qquad p_U^\star(2) = 0,$$
$$p_U^\star(3) = \frac{-(1 - q_X)R}{H_b(q_X)} + 1 - q_Y, \qquad\qquad p_U^\star(4) = \frac{-q_X R}{H_b(q_X)} + q_Y.$$

Constraint (8) is inactive if

$$(p_U(1) + p_U(2))H_b(q_{S_1}) + (p_U(3) + p_U(4))H_b(m) < C,$$
$$C > \frac{R(H_b(q_{S_1}) - H_b(m))}{H_b(q_X)} + H_b(m).$$

**Case 2.** Constraint (8) is active and constraint (6) is inactive.

The constraint (8) is active if

$$(p_U(1) + p_U(2))H_b(q_{S_1}) + (p_U(3) + p_U(4))H_b(m) = C$$

From the constraint (9), we have

$$(p_U(1) + p_U(2))H_b(q_{S_1}) + (1 - p_U(1) - p_U(2))H_b(m) = C,$$
$$p_U(1) + p_U(2) = \frac{C - H_b(m)}{H_b(q_{S_1}) - H_b(m)}.$$

Since $p_U(2) \geq 0$, which implies

$$\mathbb{E}[d_H(X, Y)] = -2(1 - q_X)q_X(p_U(1) + p_U(2)) + 4(1 - q_X)q_X p_U(2) + D_{\text{ind}}^{(B)}$$
$$\geq \frac{-2(1 - q_X)q_X(C - H_b(m))}{H_b(q_{S_1}) - H_b(m)} + D_{\text{ind}}^{(B)}.$$

This lower bound is tight, achieved by

$$p_U^\star(1) = \frac{C - H_b(m)}{H_b(q_{S_1}) - H_b(m)}, \qquad\qquad p_U^\star(2) = 0,$$
$$p_U^\star(3) = \frac{-(1 - q_X)(C - H_b(m))}{H_b(q_{S_1}) - H_b(m)} + 1 - q_Y, \qquad\qquad p_U^\star(4) = \frac{-q_X(C - H_b(m))}{H_b(q_{S_1}) - H_b(m)} + q_Y.$$

The constraint (6) is inactive if

$$H_b(q_X)(p_U(1) + p_U(2)) < R \Rightarrow C < \frac{R(H_b(q_{S_1}) - H_b(m))}{H_b(q_X)} + H_b(m).$$

**Case 3.** Both constraints (6) and (8) are active.

From case 2, the classification constraint (8) is active if $p_U(1) + p_U(2) = \frac{C - H_b(m)}{H_b(q_{S_1}) - H_b(m)}$ and $D^{(B)}(R, C, q_X, q_Y) = \frac{-2(1 - q_X)q_X(C - H_b(m))}{H_b(q_{S_1}) - H_b(m)} + D^{(B)}_{\text{ind}}$.

The rate constraint (6) is active if $C = \frac{R(H_b(q_{S_1}) - H_b(m))}{H_b(q_X)} + H_b(m)$.

**Case 4.** Both constraints (6) and (8) are inactive. When $C > H_b(q_S)$, implying that the classification constraint (8) is inactive, and the rate $R$ is sufficiently large such that $R > H_b(q_X)$, meaning the rate constraint (6) is also inactive, the minimum achievable distortion $D^{(B)}(R, C, q_X, q_Y)$ reaches its theoretical lower bound, i.e., $D^{(B)}(R, C, q_X, q_Y) = D^{(B)}_{\text{min}}$.

In summary, combining all of the cases, the closed-form expression for $D^{(B)}(R, C, q_X, q_Y)$ under Hamming distortion is given by Theorem 2. □

### A.1.3 PROOF OF THEOREM 4

**Theorem 4.** *Consider $X \sim \mathcal{N}(\mu_X, \sigma_X^2)$ and $Y \sim \mathcal{N}(\mu_Y, \sigma_Y^2)$ with MSE distortion, and let $S \sim \mathcal{N}(\mu_S, \sigma_S^2)$ be a classification variable with $\text{Cov}(X, S) = \theta_1$. The problem (4) is feasible if $C \geq \frac{1}{2} \log\left(1 - \frac{\theta_1^2}{\sigma_S^2 \sigma_X^2}\right) + h(S)$. Under shared randomness, the asymptotic DRC tradeoff is*

$$D^{(G)}(R, C, q_X, q_Y) = \begin{cases} (\mu_X - \mu_Y)^2 + \sigma_X^2 + \sigma_Y^2 - \frac{2\sigma_S \sigma_X^2 \sigma_Y}{\theta_1} \sqrt{1 - e^{-2h(S) + 2C}}, \\ \quad \frac{1}{2} \log\left(1 - \frac{\theta_1^2}{\sigma_S^2 \sigma_X^2}\right) + h(S) \leq C \leq \frac{1}{2} \log\left(1 - \frac{\theta_1^2(1 - 2^{-2R})}{\sigma_S^2 \sigma_X^2}\right) + h(S), \\ (\mu_X - \mu_Y)^2 + \sigma_X^2 + \sigma_Y^2 - 2\sigma_X \sigma_Y \sqrt{1 - 2^{-2R}}, \\ \quad C > \frac{1}{2} \log\left(1 - \frac{\theta_1^2(1 - 2^{-2R})}{\sigma_S^2 \sigma_X^2}\right) + h(S), \\ 0, \quad C > h(S) \text{ and } R > h(X). \end{cases}$$

*Proof.* We now consider problem (4) with the MSE distortion criterion. Formally,

$$D^{(G)}(R, C, q_X, q_Y) = \inf_{q_{X,Y} \in \Gamma(q_X, q_Y)} \mathbb{E}[(X - Y)^2]$$
$$\text{s.t.} \quad I(X; Y) \leq R,$$
$$h(S|Y) \leq C.$$

Using the closed-form expression for the mutual information of Gaussian variables (Cover & Thomas, 1999), we obtain

$$I(X; Y) \geq -\frac{1}{2} \log\left(1 - \frac{\theta_2^2}{\sigma_X^2 \sigma_Y^2}\right), \tag{11}$$

where $\theta_2 = \text{Cov}(X, Y) = \mathbb{E}[(X - \mu_X)(Y - \mu_Y)]$. The equality holds if and only if $X$ and $Y$ are jointly Gaussian random variables.

Since $(X, Y, S)$ are jointly Gaussian and satisfy the Markov chain $S \rightarrow X \rightarrow Y$, we have $\text{Cov}(S, Y) = \mathbb{E}\left[\mathbb{E}[S - \mu_S|X]\,\mathbb{E}[Y - \mu_Y|X]\right] = \frac{\theta_1 \theta_2}{\sigma_X^2}$. For the classification constraint,

$$h(S|Y) = h(S) - I(S; Y) \leq C \Rightarrow -\frac{1}{2} \log\left(1 - \frac{\theta_1^2}{\sigma_S^2 \sigma_X^4} \cdot \frac{\theta_2^2}{\sigma_Y^2}\right) \geq h(S) - C.$$

The MSE between $X$ and $Y$ admits the decomposition (Zhang et al., 2025):

$$\mathbb{E}[(X - Y)^2] = (\mu_X - \mu_Y)^2 + \sigma_X^2 + \sigma_Y^2 - 2\theta_2. \tag{12}$$

Hence, the optimization problem can be formulated as

$$D^{(G)}(R, C, q_X, q_Y) = \min_{\theta_2} \quad (\mu_X - \mu_Y)^2 + \sigma_X^2 + \sigma_Y^2 - 2\theta_2 \tag{13a}$$

$$\text{s.t.} \quad -\frac{1}{2}\log\left(1 - \frac{\theta_2^2}{\sigma_X^2 \sigma_Y^2}\right) \leq R, \tag{13b}$$

$$-\frac{1}{2}\log\left(1 - \frac{\theta_1^2}{\sigma_S^2 \sigma_X^4}\frac{\theta_2^2}{\sigma_Y^2}\right) \geq h(S) - C. \tag{13c}$$

To ensure (13b) is well-defined, it is necessary that $1 - \frac{\theta_2^2}{\sigma_X^2 \sigma_Y^2} > 0 \Rightarrow \frac{\theta_2^2}{\sigma_Y^2} < \sigma_X^2$. Under this condition, the mutual information between $S$ and $Y$ satisfies

$$I(S;Y) = -\frac{1}{2}\log\left(1 - \frac{\theta_1^2}{\sigma_S^2 \sigma_X^4} \times \frac{\theta_2^2}{\sigma_Y^2}\right) \leq -\frac{1}{2}\log\left(1 - \frac{\theta_1^2}{\sigma_S^2 \sigma_X^2}\right).$$

Thus, constraint (13c) is infeasible whenever $C < \frac{1}{2}\log\left(1 - \frac{\theta_1^2}{\sigma_S^2 \sigma_X^2}\right) + h(S)$. To guarantee feasibility, we assume throughout that $C \geq \frac{1}{2}\log\left(1 - \frac{\theta_1^2}{\sigma_S^2 \sigma_X^2}\right) + h(S)$.

The optimization problem (13) can then be analyzed using the KKT conditions. By systematically considering all possible combinations of active and inactive constraints, we can fully characterize the optimal solution.

**Case 1.** Constraint (13b) is active, while constraint (13c) is inactive.

From the entropy inequality and the fact that (13b) holds with equality, we obtain

$$R = I(X;Y) = h(X) + h(Y) - h(X,Y) = \frac{1}{2}\log(2\pi e\sigma_X^2) + \frac{1}{2}\log(2\pi e\sigma_Y^2) - h(X,Y)$$

$$\geq \frac{1}{2}\log(2\pi e\sigma_X^2) + \frac{1}{2}\log(2\pi e\sigma_Y^2) - \frac{1}{2}\log((2\pi e)^2(\sigma_X^2\sigma_Y^2 - \theta_2^2)) = \frac{1}{2}\log\left(\frac{\sigma_X^2\sigma_Y^2}{\sigma_X^2\sigma_Y^2 - \theta_2^2}\right).$$

The equality holds if only if $X$ and $Y$ are two jointly Gaussian random variables. This implies that

$$\theta_2 \leq \sigma_X\sigma_Y\sqrt{1 - 2^{-2R}}. \tag{14}$$

Substituting (14) into (12) yields

$$D^{(G)}(R, C, q_X, q_Y) \geq (\mu_X - \mu_Y)^2 + \sigma_X^2 + \sigma_Y^2 - 2\sigma_X\sigma_Y\sqrt{1 - 2^{-2R}}.$$

The classification constraint is inactive if $-\frac{1}{2}\log\left(1 - \frac{\theta_1^2}{\sigma_S^2 \sigma_X^4}\frac{\theta_2^2}{\sigma_Y^2}\right) > h(S) - C$, which reduces to $C > \frac{1}{2}\log\left(1 - \frac{\theta_1^2(1 - 2^{-2R})}{\sigma_S^2 \sigma_X^2}\right) + h(S)$.

**Case 2.** Constraint (13b) is inactive, while constraint (13c) is active.

When (13c) is tight, we have

$$-\frac{1}{2}\log\left(1 - \frac{\theta_1^2}{\sigma_S^2 \sigma_X^4}\frac{\theta_2^2}{\sigma_Y^2}\right) = h(S) - C \Rightarrow \theta_2 = \frac{\sigma_S\sigma_X^2\sigma_Y}{\theta_1}\sqrt{1 - 2^{-2h(S)+2C}}. \tag{15}$$

Substituting (15) into the distortion expression gives

$$\mathbb{E}[(X - Y)^2] = (\mu_X - \mu_Y)^2 + \sigma_X^2 + \sigma_Y^2 - \frac{2\sigma_S\sigma_X^2\sigma_Y}{\theta_1}\sqrt{1 - 2^{-2h(S)+2C}}.$$

The corresponding mutual information is $I(X;Y) = -\frac{1}{2}\log\left(1 - \frac{\sigma_S^2\sigma_X^2}{\theta_1^2}(1 - 2^{-2h(S)+2C})\right)$.

Thus, the rate constraint is inactive whenever

$$-\frac{1}{2}\log\left(1 - \frac{\sigma_S^2\sigma_X^2}{\theta_1^2}(1 - 2^{-2h(S)+2C})\right) < R \Rightarrow C < \frac{1}{2}\log\left(1 - \frac{\theta_1^2(1 - 2^{-2R})}{\sigma_S^2 \sigma_X^2}\right) + h(S).$$

**Case 3.** Both constraints (13b) and (13c) are active.

From Case 2, when the classification constraint is tight, we have $\theta_2 = \frac{\sigma_S \sigma_X^2 \sigma_Y}{\theta_1} \sqrt{1 - 2^{-2h(S)+2C}}$. This yields the distortion

$$\mathbb{E}[(X - Y)^2] = (\mu_X - \mu_Y)^2 + \sigma_X^2 + \sigma_Y^2 - \frac{2\sigma_S \sigma_X^2 \sigma_Y}{\theta_1} \sqrt{1 - 2^{-2h(S)+2C}}.$$

The rate constraint holds with equality: $I(X;Y) = -\frac{1}{2} \log \left( 1 - \frac{\sigma_S^2 \sigma_X^2}{\theta_1^2} \left( 1 - 2^{-2h(S)+2C} \right) \right) = R$, which implies $C = \frac{1}{2} \log \left( 1 - \frac{\theta_1^2 (1 - 2^{-2R})}{\sigma_S^2 \sigma_X^2} \right) + h(S)$.

**Case 4.** Both constraints (13b) and (13c) are inactive.

When $C > h(S)$, the classification constraint (13c) is inactive, and if the rate satisfies $R > h(X)$, the rate constraint (13b) is also inactive. In this regime, the minimum distortion $D^{(G)}(R, C, q_X, q_Y)$ achieves its theoretical bound of zero, realized by setting $Y = X$, which yields $\mathbb{E}[(X - Y)^2] = 0$. All constraints are satisfied since $I(X;Y) = h(X) < R$ and $h(S|Y) = h(S|X) \leq h(S) < C$.

In summary, combining all of the cases, the closed-form expression for $D^{(G)}(R, C, q_X, q_Y)$ under MSE distortion is given by Theorem 4. $\qquad \square$

### A.1.4 PROOF OF THEOREM 6

**Theorem 6.** *Let $X \sim \mathcal{N}(\mu_X, \sigma_X^2)$ be a Gaussian source and $S \sim \mathcal{N}(\mu_S, \sigma_S^2)$ a classification variable jointly Gaussian with $X$, such that $\mathrm{Cov}(X, S) = \theta_1$. Consider $Y$ with mean $\mathbb{E}[Y] = \mu_Y$, variance $\mathrm{Var}(Y) = \sigma_Y^2$, and covariance $\mathrm{Cov}(X, Y) = \theta_2$. Define $Y_G$ as a Gaussian random variable such that $(X, Y_G)$ is jointly Gaussian with the same first and second moments as $(X, Y)$: $\mathbb{E}[Y_G] = \mu_Y$, $\mathrm{Var}(Y_G) = \sigma_Y^2$, and $\mathrm{Cov}(X, Y_G) = \theta_2$. Under the MSE distortion with constraints $I(X;Y) \leq R$, $h(S|Y) \leq C$, and $\phi(q_X, q_Y) \leq P$, the function $D^{(\infty)}(R, P, C)$ is attained by such a jointly Gaussian $Y_G$ when the perception measure is either $W_2^2(q_X, q_Y)$ or $\phi_{KL}(q_Y \| q_X)$.*

*Proof.* Consider the DRPC problem in Theorem 5 with MSE distortion as follows.

$$D^{(\infty)}(R, P, C) = \inf_{q_{Y|X}} \mathbb{E}[(X - Y)^2]$$
$$\text{s.t.} \quad I(X;Y) \leq R, \quad h(S|Y) \leq C,$$
$$\phi(q_Y, q_X) \leq P.$$

**Distortion objective equality.** For any $Y$, $\mathbb{E}[(X - Y)^2] = (\mu_X - \mu_Y)^2 + \sigma_X^2 + \sigma_Y^2 - 2\theta_2$, which depends only on first and second moments. Since $Y$ and $Y_G$ share $(\mu_Y, \sigma_Y^2, \theta_2)$, then $\mathbb{E}[(X - Y)^2] = \mathbb{E}[(X - Y_G)^2]$.

**Rate constraint under Gaussian.** We begin with a lemma from estimation theory that compares the performance of Gaussian and non-Gaussian estimators with matched second-order statistics.

**Lemma A.2.** *(Willsky & Wornell, 2005) Let $Y$ be a random variable with mean $\mathbb{E}[Y] = \mu_Y$, variance $\mathrm{Var}(Y) = \sigma_Y^2$, and covariance $\mathrm{Cov}(X, Y) = \theta_2$. Let $Y_G$ be jointly Gaussian with $X$ and share the same mean, variance, and covariance as $Y$. Then,*

$$\mathbb{E}[(X - \mathbb{E}[X|Y_G])^2] \geq \mathbb{E}[(X - \mathbb{E}[X|Y])^2].$$

This result implies that the MMSE of a general (possibly non-Gaussian) estimator $Y$ is always less than or equal to that of a Gaussian estimator with the same first and second-order moments.

Following the derivation approach in (Zhang et al., 2025), we show that the mutual information $I(X;Y)$ is minimized when $Y$ is constrained to be jointly Gaussian with $X$. Specifically, we have:

$$
\begin{aligned}
I(X;Y) &= h(X) - h(X|Y) \\
&\geq h(X) - h(X - \mathbb{E}[X|Y]) \\
&\overset{(a)}{\geq} h(X) - \tfrac{1}{2}\log\big(2\pi e\, \mathbb{E}[(X - \mathbb{E}[X|Y])^2]\big) \\
&\overset{(b)}{\geq} h(X) - \tfrac{1}{2}\log\big(2\pi e\, \mathbb{E}[(X - \mathbb{E}[X|Y_G])^2]\big) \\
&= h(X) - h(X - \mathbb{E}[X|Y_G]) \\
&\overset{(c)}{=} h(X) - h(X|Y_G) \\
&= I(X;Y_G).
\end{aligned}
$$

where inequality (a) follows from the fact that the Gaussian distribution maximizes differential entropy for a given variance; inequality (b) follows from Lemma A.2; and equality (c) holds because the estimation error is independent of $Y_G$. Hence, if $I(X;Y) \leq R$ then $I(X;Y_G) \leq R$.

**Perception divergence under Gaussian.** From Xie et al. (2025), we have the following proposition

**Proposition 1.** *Xie et al. (2025) For $X \sim \mathcal{N}(\mu_X, \sigma_X^2)$ and any distribution $q_Y$ with $\mathbb{E}[Y^2] < \infty$, we obtain*

$$
\phi_{KL}(q_Y \| q_X) \geq \phi_{KL}\big(\mathcal{N}(\mu_Y, \sigma_Y^2) \,\|\, \mathcal{N}(\mu_X, \sigma_X^2)\big) = \log\frac{\sigma_X}{\sigma_Y} + \frac{(\mu_X - \mu_Y)^2 + \sigma_Y^2 - \sigma_X^2}{2\sigma_X^2}.
$$

Therefore, $\phi_{KL}(q_Y \| q_X) \geq \phi_{KL}(\mathcal{N}(\mu_Y, \sigma_Y^2) \| \mathcal{N}(\mu_X, \sigma_X^2)) = \phi_{KL}(q_{Y_G}, q_X)$.

Similarly, for the case of $W_2(q_X, q_Y)$, we have

**Proposition 2.** *Givens & Shortt (1984) proved that for distributions $p_X$ and $p_Y$ with $\mathbb{E}[X^2] < \infty$ and $\mathbb{E}[Y^2] < \infty$,*

$$
W_2^2(p_X, p_Y) \geq W_2^2\big(\mathcal{N}(\mu_X, \sigma_X^2), \mathcal{N}(\mu_Y, \sigma_Y^2)\big) = (\mu_X - \mu_Y)^2 + (\sigma_X - \sigma_Y)^2.
$$

Note that by expanding out $W_2(q_X, q_Y)$, one can see that the optimal coupling is identified only through the cross-term between $X$ and $Y$; since every coupling of $q_X$ and $q_Y$ induces a Gaussian coupling of $q_X$ and $q_{Y_G}$ with the same covariance, it follows that $W_2^2(q_X, q_Y) \geq W_2^2(q_X, q_{Y_G})$. Hence, if $\phi(q_Y, q_X) \leq P$, then $\phi(q_{Y_G}, q_X) \leq P$.

**Classification constraint under Gaussian.** We now demonstrate that the jointly Gaussian estimator $Y_G$ for $X$ is optimal. To formalize this result and establish the optimality of $Y_G$, we first propose the following lemma.

**Lemma A.3.** *Given $S \to X \to Y$, we have $h(S|Y) \geq h(S|Y_G)$.*

*Proof of Lemma A.3.* Let $\rho_{XS} = \frac{\text{Cov}(X,S)}{\sigma_X \sigma_S} = \frac{\theta_1}{\sigma_X \sigma_S}$ denote the correlation coefficient between $X$ and $S$. Since $X$ and $S$ are jointly Gaussian, $S$ can be expressed as $S = aX + N$, where $a = \rho_{XS}\frac{\sigma_S}{\sigma_X}$ and $N$ is a zero-mean Gaussian random variable with variance $(1 - \rho_{XS}^2)\sigma_S^2$, independent of $X$.

Applying the conditional entropy-power inequality (Berger & Zamir, 1999), we obtain:

$$
h(S|Y) = h(aX + N|Y) \geq \frac{1}{2}\log\left(2^{2h(aX|Y)} + 2^{2h(N)}\right).
$$

with equality if and only if $(X, Y)$ are jointly Gaussian. It follows that $h(S|Y) \geq h(S|Y_G) = \frac{1}{2}\log\left(2^{2h(aX|Y_G)} + 2^{2h(N)}\right)$, which completes the proof. $\qquad\square$

The covariance of $(S, Y)$ is fixed by $(\sigma_S^2, \sigma_Y^2, \text{Cov}(S, Y) = a\theta_2)$. Among all $(S, Y)$ with a given covariance matrix, the Gaussian joint maximizes $I(S;Y)$; equivalently, it minimizes $h(S|Y)$. Thus, if $h(S|Y) \leq C$ then also $h(S|Y_G) \leq C$.

Overall, it suffices to solve the following optimization problem:

$$D^{(\infty)}(R,P,C) = \inf_{p_{Y_G|X}} \mathbb{E}[(X - Y_G)^2]$$
$$\text{s.t.} \quad I(X;Y_G) \leq R, \quad h(S|Y_G) \leq C,$$
$$\phi(q_{Y_G}, q_X) \leq P.$$

Therefore, the DRPC function, $D^{(\infty)}(R,P,C)$, is achieved by $Y_G$ with the cases of $\phi(q_X, q_Y) = W_2^2(q_X, q_Y)$ or $\phi(q_X, q_Y) = \phi_{KL}(q_Y, q_X)$. $\square$

### A.1.5 PROOF OF THEOREM 7

**Theorem 7.** *Let* $X \sim \mathcal{N}(\mu_X, \sigma_X^2)$ *and* $Y \sim \mathcal{N}(\mu_Y, \sigma_Y^2)$ *be two Gaussian random variables. Let* $S \sim \mathcal{N}(\mu_S, \sigma_S^2)$ *be an associated classification variable with a covariance of* $Cov(X, S) = \theta_1$ *and be jointly Gaussian. For the case* $d(X, Y) = (X - Y)^2$ *and* $\phi(p_X, p_Y) = \phi_{KL}(p_Y \| p_X)$, *we have*

$D_{KL}^{(G)}(R,P,C)$

$$= \begin{cases} \sigma_X^2 - \sigma_X^2(1 - 2^{-2R}), \sigma(P) \leq \sigma_X\sqrt{1 - 2^{-2R}} \text{ and } C > \frac{1}{2}\log\left(1 - \frac{\theta_1^2(1 - 2^{-2R})}{\sigma_S^2\sigma_X^2}\right) + h(S) \\ \sigma_X^2 + \sigma^2(P) - 2\sigma_X\sigma(P)\sqrt{1 - 2^{-2R}}, \\ \quad \sigma(P) > \sigma_X\sqrt{1 - 2^{-2R}} \text{ and } C > \frac{1}{2}\log\left(1 - \frac{\theta_1^2(1 - 2^{-2R})}{\sigma_S^2\sigma_X^2}\right) + h(S) \\ \sigma_X^2 - \frac{\sigma_S^2\sigma_X^4}{\theta_1^2}(1 - 2^{-2h(S)+2C}), \sigma(P) \leq \frac{\sigma_S\sigma_X^2}{\theta_1}\sqrt{1 - 2^{-2h(S)+2C}} \\ \quad \text{and} \frac{1}{2}\log\left(1 - \frac{\theta_1^2}{\sigma_S^2\sigma_X^2}\right) + h(S) \leq C \leq \frac{1}{2}\log\left(1 - \frac{\theta_1^2(\sigma_X^2 - \sigma_X^2 2^{-2R})}{\sigma_S^2\sigma_X^4}\right) + h(S) \\ \sigma_X^2 + \sigma^2(P) - \frac{2\sigma_S\sigma_X^2\sigma(P)}{\theta_1}\sqrt{1 - 2^{-2h(S)+2C}}, \sigma(P) > \frac{\sigma_S\sigma_X^2}{\theta_1}\sqrt{1 - 2^{-2h(S)+2C}} \\ \quad \text{and} \frac{1}{2}\log\left(1 - \frac{\theta_1^2}{\sigma_S^2\sigma_X^2}\right) + h(S) \leq C \leq \frac{1}{2}\log\left(1 - \frac{\theta_1^2(\sigma_X^2 - \sigma_X^2 2^{-2R})}{\sigma_S^2\sigma_X^4}\right) + h(S) \\ 0, \quad C > h(S) \text{ and } R > h(X). \end{cases}$$

*where* $\sigma(P)$ *being the unique number* $\sigma \in [0, \sigma_X]$ *satisfying* $\psi(\sigma) = P$ *and* $\psi(\sigma_Y) = \log\frac{\sigma_X}{\sigma_Y} + \frac{\sigma_Y^2 - \sigma_X^2}{2\sigma_X^2}$.

*Proof.* Extending from the result of Xie et al. (2025), we establish the following lemma.

**Lemma A.4.** *Consider* $X \sim \mathcal{N}(\mu_X, \sigma_X^2)$ *with MSE distortion* $d(X, Y) = (X - Y)^2$ *and perception measure* $\phi(p_X, p_Y) = \phi_{KL}(p_Y \| p_X)$. *Then*

$$D_{KL}^{(G)}(R,P,C) = \inf_{p_Y} D^{(\infty)}(R, C, q_X, q_Y)$$
$$\text{s.t.} \quad \mu_Y = \mu_X, \quad \sigma_Y \leq \sigma_X,$$
$$h(S|Y) \leq C, \quad \phi_{KL}(q_Y \| q_X) \leq P.$$

*Proof of Lemma A.4.* We argue that restricting to $p_Y$ with $\sigma_Y \leq \sigma_X$ incurs no loss of optimality. Suppose $\sigma_Y > \sigma_X$ and define $Y' := \frac{\sigma_X}{\sigma_Y}(Y - \mu_Y) + \mu_X$.

**Distortion objective.** A direct calculation gives

$$\mathbb{E}[(X - Y')^2] = 2\sigma_X^2 - \frac{2\sigma_X}{\sigma_Y}\mathbb{E}[(X - \mu_X)(Y - \mu_Y)]$$
$$\leq \sigma_X^2 + \sigma_Y^2 - 2\mathbb{E}[(X - \mu_X)(Y - \mu_Y)] = \mathbb{E}[(X - Y)^2],$$

where the inequality holds since $k^2\sigma_Y^2 - 2k\,\mathbb{E}[(X - \mu_X)(Y - \mu_Y)]$ is increasing in $k \in [\frac{\sigma_X}{\sigma_Y}, 1]$.

**Rate constraint.** Since $X \leftrightarrow Z \leftrightarrow Y$ is a Markov chain, then so is $X \leftrightarrow Z \leftrightarrow Y'$, and we have $I(Y'; Z) = I(Y; Z)$.

**Classification constraint.** Let $\theta_2' = \mathrm{Cov}(X, Y')$ and $\theta_2 = \mathrm{Cov}(X, Y)$. Then

$$h(S|Y') = h(S) + \tfrac{1}{2}\log\left(1 - \frac{\theta_1^2}{\sigma_S^2\sigma_X^4}\frac{\theta_2'^2}{\sigma_{Y'}^2}\right),$$

$$h(S|Y) = h(S) + \tfrac{1}{2}\log\left(1 - \frac{\theta_1^2}{\sigma_S^2\sigma_X^4}\frac{\theta_2^2}{\sigma_Y^2}\right).$$

Since $Y' := \frac{\sigma_X}{\sigma_Y}(Y - \mu_Y) + \mu_X$, then $\mu_{Y'} = \mathbb{E}\left[\frac{\sigma_X}{\sigma_Y}(Y - \mu_Y) + \mu_X\right] = \frac{\sigma_X}{\sigma_Y}(\mu_Y - \mu_Y) + \mu_X = \mu_X$.
We also have $\sigma_{Y'} = \left(\frac{\sigma_X}{\sigma_Y}\right)^2\sigma_Y = \frac{\sigma_X^2}{\sigma_Y}$ and $\theta_2' = \frac{\sigma_X}{\sigma_Y}\theta_2$. Therefore,

$$h(S|Y') = h(S) - I(S;Y')$$

$$= h(S) + \frac{1}{2}\log\left(1 - \frac{\theta_1^2}{\sigma_S^2\sigma_X^4}\frac{\theta_2'^2}{\sigma_{Y'}^2}\right)$$

$$\stackrel{(a)}{\leq} h(S) + \frac{1}{2}\log\left(1 - \frac{\theta_1^2}{\sigma_S^2\sigma_X^4}\frac{\theta_2^2}{\sigma_Y^2}\right)$$

$$= h(S) - I(S;Y) = h(S|Y).$$

where (a) is due to $\sigma_Y > \sigma_X$, infer that $\frac{\theta_2'^2}{\sigma_{Y'}^2} \geq \frac{\theta_2^2}{\sigma_Y^2}$.

**Perception constraint.** Finally,

$$\phi_{KL}(q_{Y'}\|q_X) = -h(Y') + \frac{1}{2}\log(2\pi\sigma_X^2) + \frac{(\mu_X - \mu_{Y'})^2 + \sigma_{Y'}^2}{2\sigma_X^2}$$

$$\leq -h(Y) + \frac{1}{2}\log(2\pi\sigma_X^2) + \frac{(\mu_X - \mu_Y)^2 + \sigma_Y^2}{2\sigma_X^2} = \phi_{KL}(q_Y\|q_X) = \phi_{KL}(q_Y\|q_X),$$

where the inequality follows from the convexity of $\psi(\sigma_Y) := \log\frac{\sigma_X}{\sigma_Y} + \frac{\sigma_Y^2 - \sigma_X^2}{2\sigma_X^2}$, which is nonnegative for $\sigma_Y \geq \sigma_X$.

Together, these arguments establish that replacing $Y$ by $Y'$ cannot increase distortion, rate, and perception divergence, nor violate the classification constraint. This proves the lemma. $\square$

From Proposition 1, together with the constraints $\mu_Y = \mu_X$, $\sigma_Y \leq \sigma_X$, and $\phi_{KL}(p_Y\|p_X) \leq P$, it follows that $\sigma_Y \in [\sigma(P), \sigma_X]$, where $\sigma(P)$ is uniquely defined as the value $\sigma \in [0, \sigma_X]$ satisfying $\psi(\sigma) = P$.

In view of Lemma A.4, it suffices to restrict to distributions $p_Y$ with $\mu_Y = \mu_X$ and $\sigma_Y \leq \sigma_X$ when evaluating $D^{(G)}(R, P, C)$. Moreover, Theorem 6 allows us to further assume that $Y$ is Gaussian, yielding

$$D_{KL}^{(G)}(R, P, C) = \min_{\sigma_Y \in [\sigma(P), \sigma_X]} D^{(G)}(R, C, q_X, q_Y). \tag{16}$$

**Case 1.** If $C > \frac{1}{2}\log\left(1 - \frac{\theta_1^2(1 - 2^{-2R})}{\sigma_S^2\sigma_X^2}\right) + h(S)$, then

$$D^{(G)}(R, C, q_X, q_Y) = (\mu_X - \mu_Y)^2 + \sigma_X^2 + \sigma_Y^2 - 2\sigma_X\sigma_Y\sqrt{1 - 2^{-2R}}.$$

The term $\sigma_Y^2 - 2\sigma_X\sigma_Y\sqrt{1 - 2^{-2R}}$ decreases monotonically over $\sigma_Y \in [0, \sigma_X\sqrt{1 - 2^{-2R}}]$ and increases thereafter. Thus, the minimizing $\sigma_Y$ in (16) is

$$\sigma_Y = \begin{cases} \sigma_X\sqrt{1 - 2^{-2R}}, & \sigma(P) \leq \sigma_X\sqrt{1 - 2^{-2R}}, \\ \sigma(P), & \sigma(P) > \sigma_X\sqrt{1 - 2^{-2R}}. \end{cases}$$

Hence,

$$D_{KL}^{(G)}(R, P, C) = \begin{cases} \sigma_X^2 - \sigma_X^2(1 - 2^{-2R}), & \sigma(P) \leq \sigma_X\sqrt{1 - 2^{-2R}}, \\ \sigma_X^2 + \sigma(P)^2 - 2\sigma_X\sigma(P)\sqrt{1 - 2^{-2R}}, & \sigma(P) > \sigma_X\sqrt{1 - 2^{-2R}}. \end{cases}$$

**Case 2.** If $\frac{1}{2}\log\left(1 - \frac{\theta_1^2}{\sigma_S^2\sigma_X^2}\right) + h(S) \leq C \leq \frac{1}{2}\log\left(1 - \frac{\theta_1^2(\sigma_X^2 - \sigma_X^2 2^{-2R})}{\sigma_S^2\sigma_X^4}\right) + h(S)$, then

$$D^{(G)}(R, C, q_X, q_Y) = (\mu_X - \mu_Y)^2 + \sigma_X^2 + \sigma_Y^2 - \frac{2\sigma_S\sigma_X^2\sigma_Y}{\theta_1}\sqrt{1 - 2^{-2h(S)+2C}}.$$

Here, $\sigma_Y^2 - \frac{2\sigma_S\sigma_X^2\sigma_Y}{\theta_1}\sqrt{1 - 2^{-2h(S)+2C}}$ decreases over $\sigma_Y \in \left[0, \frac{\sigma_S\sigma_X^2}{\theta_1}\sqrt{1 - 2^{-2h(S)+2C}}\right]$ and increases thereafter. Thus,

$$\sigma_Y = \begin{cases} \frac{\sigma_S\sigma_X^2}{\theta_1}\sqrt{1 - 2^{-2h(S)+2C}}, & \sigma(P) \leq \frac{\sigma_S\sigma_X^2}{\theta_1}\sqrt{1 - 2^{-2h(S)+2C}}, \\ \sigma(P), & \sigma(P) > \frac{\sigma_S\sigma_X^2}{\theta_1}\sqrt{1 - 2^{-2h(S)+2C}}. \end{cases}$$

Therefore,

$$D_{KL}^{(G)}(R, P, C) = \begin{cases} \sigma_X^2 - \frac{\sigma_S^2\sigma_X^4}{\theta_1^2}\left(1 - 2^{-2h(S)+2C}\right), & \sigma(P) \leq \frac{\sigma_S\sigma_X^2}{\theta_1}\sqrt{1 - 2^{-2h(S)+2C}}, \\ \sigma_X^2 + \sigma(P)^2 - \frac{2\sigma_S\sigma_X^2\sigma(P)}{\theta_1}\sqrt{1 - 2^{-2h(S)+2C}}, & \sigma(P) > \frac{\sigma_S\sigma_X^2}{\theta_1}\sqrt{1 - 2^{-2h(S)+2C}}. \end{cases}$$

**Case 3.** If $C > h(S)$ and $R > h(X)$, then $D_{KL}^{(G)}(R, P, C) = 0$.

In summary, by combining the above cases, we obtain the closed-form expression for $D_{KL}^{(G)}(R, P, C)$, as stated in Theorem 7. $\qquad\square$

## A.2 RDC Expression for Bernoulli Case in One-shot Setting

In addition to Theorem 1, we propose the following definition of the rate-distortion-classification function based on the constrained optimal transport in the one-shot setting.

**Definition 5.** *Let $X \sim p_X$ be the degraded source, $Y \sim p_Y$ the reconstruction, and $S \sim p_S$ the associated classification variable with covariance $\text{Cov}(X, S)$. Define $Q(p_X, p_Y)$ as the set of joint distributions $p_{U,X,Y}$ with marginals $p_X, p_Y$ that factorize as $p_{U,X,Y} = p_U\, p_X\, p_{Y|X,U}$. The rate-distortion-classification function is based on the constrained optimal transport with distortion loss $D$, classification loss $C$, and shared randomness as follows*

$$R(D, C, p_X, p_Y) = \inf_{p_{U,X,Y} \in Q(p_X, p_Y)} H(Y|U) \tag{17}$$

$$\text{s.t.} \quad H(Y|X, U) = 0, \quad I(X; U) = 0,$$
$$\mathbb{E}[d(X, Y)] \leq D, \quad H(S|Y) \leq C.$$

The closed-form solution of $R^{(B)}(D, C, q_X, q_Y)$ for the Bernoulli case is derived by Theorem A.2.

**Theorem A.2.** *Consider a Bernoulli source $X \sim \text{Bern}(q_X)$, $Y \sim \text{Bern}(q_Y)$, and a classification variable $S$ with the binary symmetric joint distribution given by $S = X \oplus S_1$ where $S \sim \text{Bern}(q_S)$ and $S_1 \sim \text{Bern}(q_{S_1})$ $(0 \leq q_X, q_S, q_{S_1} \leq \frac{1}{2})$. The problem 17 is feasible if $C \geq H_b(q_{S_1})$. Assume the Hamming distortion measure. Under common randomness, we have*

$$R^{(B)}(D, C, q_X, q_Y) = \begin{cases} \dfrac{H_b(q_X)(D_{\text{ind}}^{(B)} - D)}{2(1 - q_X)q_X}, & D_{\min}^{(B)} \leq D < \dfrac{2(1 - q_X)q_X[C - H_b(m)]}{H_b(m) - H_b(q_{S_1})} + D_{\text{ind}}^{(B)} \\ \dfrac{H_b(q_X)[H_b(m) - C]}{H_b(m) - H_b(q_{S_1})}, & \dfrac{2(1 - q_X)q_X[C - H_b(m)]}{H_b(m) - H_b(q_{S_1})} + D_{\text{ind}}^{(B)} \leq D \leq D_{\max}^{(B)} \\ 0, & C \geq H_b(m) \text{ and } D_{\text{ind}}^{(B)} \leq D \leq D_{\max}^{(B)}. \end{cases}$$

*Proof.* Following the proof of Theorem 2, we can formulate the problem (17) as

$$R^{(B)}(D, C, q_X, q_Y) = \min_{p_U(1),\, p_U(2),\, p_U(3),\, p_U(4)} H_b(q_X)(p_U(1) + p_U(2)) \tag{18}$$

$$\text{s.t.} \quad p_U(2) + q_X p_U(3) + (1 - q_X)p_U(4) \leq D, \tag{19}$$
$$q_X p_U(1) + (1 - q_X)p_U(2) + p_U(4) = q_Y, \tag{20}$$
$$(p_U(1) + p_U(2))H_b(q_{S_1}) + (p_U(3) + p_U(4))H_b(m) \leq C, \tag{21}$$
$$p_U(1) + p_U(2) + p_U(3) + p_U(4) = 1, \tag{22}$$
$$p_U(1), p_U(2), p_U(3), p_U(4) \geq 0. \tag{23}$$

**The activity of the nonnegative constraints.** We now analyze the activity of the non-negativity constraints in (23). Since the objective function in (18) depends only on $p_U(1)$ and $p_U(2)$, minimization requires reducing their values whenever possible. Hence, we examine the following cases:

• For $p_U(1) = p_U(2) = 0$. Substituting into (20) and (22) yields $p_U(3) = 1 - q_Y$ and $p_U(4) = q_Y$. In this situation, the feasibility conditions $p_U(3) \geq 0$ and $p_U(4) \geq 0$ are automatically satisfied and hence inactive.

• For $p_U(1) = 0$, $p_U(2) \neq 0$. Constraint (22) becomes $p_U(2) + p_U(3) + p_U(4) = 1$. Minimizing the objective requires allocating as much probability mass as possible to $p_U(3)$ and $p_U(4)$, rendering the non-negativity constraints $p_U(3) \geq 0$ and $p_U(4) \geq 0$ inactive.

• For $p_U(2) = 0$, $p_U(1) \neq 0$. This case is symmetric to Case 2 and leads to the same conclusion.

In all of the cases, the optimal solution enforces inactivity of the non-negative constraints on $p_U(3)$ and $p_U(4)$. On the other hand, eliminating $p_U(3), p_U(4)$ via (20)–(22) gives

$$\mathbb{E}[d_H(X,Y)] = D_{\text{ind}}^{(B)} - 2(1-q_X)q_X p_U(1) + 2(1-q_X)q_X\, p_U(2).$$

For $0 \leq q_X \leq \frac{1}{2}$, the distortion $\mathbb{E}[d_H(X,Y)]$ is strictly increasing in $p_U(2)$ and decreasing in $p_U(1)$; hence it is optimal to take $p_U(2) = 0$ and $p_U(1) > 0$.

Similar to the proof of Theorem 2, our approach explores all possible combinations of active and inactive rate and classification constraints to characterize the optimal solution.

**Case 1.** Constraint (19) is active, while constraint (21) is inactive.

Since $p_U(2) \geq 0$ and the rate constraint holds with equality, we have

$$D = -2(1-q_X)q_X(p_U(1) + p_U(2)) + 4(1-q_X)q_X p_U(2) + D_{\text{ind}}^{(B)}$$
$$\geq -2(1-q_X)q_X(p_U(1) + p_U(2)) + D_{\text{ind}}^{(B)}.$$

Combining with (19) gives $p_U(1) + p_U(2) \geq \frac{D_{\text{ind}}^{(B)} - D}{2(1-q_X)q_X}$. Hence,

$$R^{(B)}(D, C, q_X, q_Y) \geq \frac{H_b(q_X)\,(D_{\text{ind}}^{(B)} - D)}{2(1-q_X)q_X}.$$

This lower bound is tight, achieved by

$$p_U^\star(1) = \frac{D_{\text{ind}}^{(B)} - D}{2(1-q_X)q_X}, \qquad\qquad p_U^\star(2) = 0,$$
$$p_U^\star(3) = -\frac{D_{\text{ind}}^{(B)} - D}{2q_X} + 1 - q_Y, \qquad\qquad p_U^\star(4) = -\frac{D_{\text{ind}}^{(B)} - D}{2(1-q_X)} + q_Y.$$

Constraint (21) is inactive if

$$(p_U(1) + p_U(2))H_b(q_{S_1}) + (p_U(3) + p_U(4))H_b(m) < C,$$
$$D < \frac{2(1-q_X)q_X\,[C - H_b(m)]}{H_b(m) - H_b(q_{S_1})} + D_{\text{ind}}^{(B)}.$$

**Case 2.** Constraint (21) is active, while constraint (19) is inactive.

If (21) holds with equality, then

$$(p_U(1) + p_U(2))H_b(q_{S_1}) + (p_U(3) + p_U(4))H_b(m) = C.$$

Using (22), this yields

$$(p_U(1) + p_U(2))H_b(q_{S_1}) + (1 - p_U(1) - p_U(2))H_b(m) = C,$$
$$p_U(1) + p_U(2) = \frac{H_b(m) - C}{H_b(m) - H_b(q_{S_1})}.$$

Consequently, $R^{(B)}(D, C, q_X, q_Y) = \frac{H_b(q_X)\,[H_b(m) - C]}{H_b(m) - H_b(q_{S_1})}$.

This bound is tight, attained by

$$p_U^\star(1) = \frac{H_b(m) - C}{H_b(m) - H_b(q_{S_1})}, \qquad\qquad p_U^\star(2) = 0,$$

$$p_U^\star(3) = \frac{-(1-q_X)(H_b(m) - C)}{H_b(m) - H_b(q_{S_1})} + 1 - q_Y, \qquad p_U^\star(4) = \frac{-q_X(H_b(m) - C)}{H_b(m) - H_b(q_{S_1})} + q_Y.$$

The rate constraint (19) is inactive provided

$$p_U(2) + q_X p_U(3) + (1 - q_X)p_U(4) < D \Rightarrow D > \frac{2(1 - q_X)q_X \left[C - H_b(m)\right]}{H_b(m) - H_b(q_{S_1})} + D_{\text{ind}}^{(B)}.$$

**Case 3.** Both constraints (19) and (21) are active.

From Case 2, if the classification constraint is tight,

$$p_U(1) + p_U(2) = \frac{H_b(m) - C}{H_b(m) - H_b(q_{S_1})} \text{ and } R^{(B)}(D, C, q_X, q_Y) = \frac{H_b(q_X)\left[H_b(m) - C\right]}{H_b(m) - H_b(q_{S_1})}.$$

The rate constraint is simultaneously active if $D = \frac{2(1-q_X)q_X\left[C - H_b(m)\right]}{H_b(m) - H_b(q_{S_1})} + D_{\text{ind}}^{(B)}$.

**Case 4.** Neither constraint (19) nor (21) is active.

We observe that the rate achieves its theoretical minimum when $R^{(B)}(D, C, q_X, q_Y) = H_b(q_X)(p_U(1) + p_U(2)) = 0$, which implies $p_U(1) = p_U(2) = 0$. Substituting into the constraints (20) and (22), we obtain $p_U(3) = 1 - q_Y, \quad p_U(4) = q_Y$. Then, using constraints (19) and (21), we find that the feasibility of this configuration requires $D \geq D_{\text{ind}}^{(B)}$ and $C \geq H_b(m)$. Therefore, the minimum achievable rate is zero, i.e., $R^{(B)}(D, C, q_X, q_Y) = 0$, if and only if the distortion and classification loss exceed the respective thresholds: $D_{\text{ind}}^{(B)} \leq D \leq D_{\text{max}}^{(B)}$ and $C \geq H_b(m)$.

In summary, combining all of the cases, the closed-form expression for $R^{(B)}(D, C, q_X, q_Y)$ under Hamming distortion is given by Theorem A.2. □

## A.3 RDC EXPRESSION FOR GAUSSIAN CASE IN ASYMTOPIC SETTING

In addition to Theorem 3, we propose the following definition of the rate-distortion-classification function based on the constrained optimal transport in the asymptotic setting.

**Definition 6.** *We have*

$$R^{(\infty)}(D, C, p_X, p_Y) = \inf_{p_{X,Y} \in \Gamma(p_X, p_Y)} I(X; Y) \tag{24}$$

$$\text{s.t.} \quad \mathbb{E}[d(X, Y)] \leq D,$$
$$H(S|Y) \leq C.$$

The pair of functions $(D^{(\infty)}(R, C, p_X, p_Y), R^{(\infty)}(D, C, p_X, p_Y))$ are natural asymptotic analogues of their one-shot counterparts. They provide a Shannon-style single-letter characterization of cross-domain lossy compression with classification constraints. In the next section, we apply these results for Gaussian sources, deriving closed-form expressions that reveal the explicit tradeoffs between rate, distortion, and classification accuracy.

**Theorem A.3.** *Consider $X \sim \mathcal{N}(\mu_X, \sigma_X^2)$ and $Y \sim \mathcal{N}(\mu_Y, \sigma_Y^2)$ be two Gaussian random variables, and let $d(\cdot, \cdot)$ be the MSE distortion measure. Let $S \sim \mathcal{N}(\mu_S, \sigma_S^2)$ be an associated classification variable, with a covariance of $Cov(X, S) = \theta_1$. The problem (24) is feasible if*

$C \geq \frac{1}{2} \log \left(1 - \frac{\theta_1^2}{\sigma_S^2 \sigma_X^2}\right) + h(S)$. *Under the common randomness, we have*

$$R^{(G)}(D, C, q_X, q_Y) = \begin{cases} -\frac{1}{2} \log \left(1 - \frac{[(\mu_X - \mu_Y)^2 + \sigma_X^2 + \sigma_Y^2 - D]^2}{4\sigma_X^2 \sigma_Y^2}\right), \\ \quad D < \frac{[(\mu_X - \mu_Y)^2 + \sigma_X^2 + \sigma_Y^2]\theta_1 - 2\sqrt{1 - 2^{2(C-h(S))}} \sigma_S \sigma_X^2 \sigma_Y}{\theta_1} \\ -\frac{1}{2} \log \left(1 - \frac{\sigma_S^2 \sigma_X^2 (1 - 2^{-2h(S)+2C})}{\theta_1^2}\right), \\ \quad D \geq \frac{[(\mu_X - \mu_Y)^2 + \sigma_X^2 + \sigma_Y^2]\theta_1 - 2\sqrt{1 - 2^{2(C-h(S))}} \sigma_S \sigma_X^2 \sigma_Y}{\theta_1} \\ 0, \quad C > h(S) \text{ and } D > (\mu_X - \mu_Y)^2 + \sigma_X^2 + \sigma_Y^2. \end{cases}$$

*Proof.* Following the proof of Theorem 4, the problem (24) can be formulated as

$$R^{(G)}(D, C, q_X, q_Y) = \min_{\theta_2} \quad -\frac{1}{2} \log \left(1 - \frac{\theta_2^2}{\sigma_X^2 \sigma_Y^2}\right) \tag{25a}$$

$$\text{s.t.} \quad (\mu_X - \mu_Y)^2 + \sigma_X^2 + \sigma_Y^2 - 2\theta_2 \leq D, \tag{25b}$$

$$-\frac{1}{2} \log \left(1 - \frac{\theta_1^2}{\sigma_S^2 \sigma_X^4} \frac{\theta_2^2}{\sigma_Y^2}\right) \geq h(S) - C. \tag{25c}$$

The optimization problem (25) can be analyzed using the KKT conditions. By systematically examining all possible combinations of active and inactive rate and classification constraints, we obtain the following cases.

**Case 1.** Constraint (25b) is active while constraint (25c) is inactive.

From the distortion constraint, we have

$$(\mu_X - \mu_Y)^2 + \sigma_X^2 + \sigma_Y^2 - 2\theta_2 = D \Rightarrow \theta_2 = \frac{(\mu_X - \mu_Y)^2 + \sigma_X^2 + \sigma_Y^2 - D}{2}.$$

Substituting this into the rate expression gives

$$R^{(G)}(D, C, q_X, q_Y) = -\frac{1}{2} \log \left(1 - \frac{[(\mu_X - \mu_Y)^2 + \sigma_X^2 + \sigma_Y^2 - D]^2}{4\sigma_X^2 \sigma_Y^2}\right).$$

The classification constraint is inactive provided

$$-\frac{1}{2} \log \left(1 - \frac{\theta_1^2}{\sigma_S^2 \sigma_X^4} \frac{\theta_2^2}{\sigma_Y^2}\right) > h(S) - C$$

$$\Rightarrow D < \frac{[(\mu_X - \mu_Y)^2 + \sigma_X^2 + \sigma_Y^2]\theta_1 - 2\sqrt{1 - 2^{2(C-h(S))}} \sigma_S \sigma_X^2 \sigma_Y}{\theta_1}.$$

**Case 2.** Constraint (25b) is inactive while constraint (25c) is active.

From the classification constraint, we obtain

$$-\frac{1}{2} \log \left(1 - \frac{\theta_1^2}{\sigma_S^2 \sigma_X^4} \frac{\theta_2^2}{\sigma_Y^2}\right) = h(S) - C \Rightarrow \theta_2 = \frac{\sigma_S \sigma_X^2 \sigma_Y}{\theta_1} \sqrt{1 - 2^{-2h(S)+2C}}.$$

Substituting into the rate expression yields

$$R^{(G)}(D, C, q_X, q_Y) = -\frac{1}{2} \log \left(1 - \frac{\sigma_S^2 \sigma_X^2 (1 - 2^{-2h(S)+2C})}{\theta_1^2}\right).$$

The distortion constraint is inactive if

$$(\mu_X - \mu_Y)^2 + \sigma_X^2 + \sigma_Y^2 - 2\theta_2 < D,$$

$$\Rightarrow D > \frac{[(\mu_X - \mu_Y)^2 + \sigma_X^2 + \sigma_Y^2]\theta_1 - 2\sqrt{1 - 2^{2(C-h(S))}} \sigma_S \sigma_X^2 \sigma_Y}{\theta_1}.$$

**Case 3.** Both constraints (25b) and (25c) are active.

From Case 2, if (25c) is active then $\theta_2 = \frac{\sigma_S \sigma_X^2 \sigma_Y}{\theta_1}\sqrt{1 - 2^{-2h(S)+2C}}$. Meanwhile, the distortion constraint requires $(\mu_X - \mu_Y)^2 + \sigma_X^2 + \sigma_Y^2 - 2\theta_2 = D$, which implies

$$D = \frac{[(\mu_X - \mu_Y)^2 + \sigma_X^2 + \sigma_Y^2]\theta_1 - 2\sqrt{1 - 2^{2(C-h(S))}}\sigma_S \sigma_X^2 \sigma_Y}{\theta_1}.$$

In this case, the rate is

$$R^{(G)}(D, C, q_X, q_Y) = -\frac{1}{2}\log\left(1 - \frac{[(\mu_X - \mu_Y)^2 + \sigma_X^2 + \sigma_Y^2 - D]^2}{4\sigma_X^2 \sigma_Y^2}\right).$$

**Case 4.** Both constraints (25b) and (25c) are inactive.

When $C > h(S)$, the classification constraint (25c) becomes inactive, and if the distortion threshold satisfies $D > (\mu_X - \mu_Y)^2 + \sigma_X^2 + \sigma_Y^2$, the distortion constraint (25b) is also inactive. In this regime, the minimum rate reduces to its theoretical bound, i.e., $R^{(G)}(D, C, q_X, q_Y) = 0$. This occurs when $X$ and $Y$ are independent, yielding $I(X; Y) = 0$. All constraints are satisfied since $\mathbb{E}[(X - Y)^2] = (\mu_X - \mu_Y)^2 + \sigma_X^2 + \sigma_Y^2 < D$ and $h(S|Y) = h(S|X) \leq h(S) < C$.

In summary, combining all of the cases, the closed-form expression for $R^{(G)}(D, C, q_X, q_Y)$ under MSE distortion is given by Theorem A.3. $\qquad \square$

### A.4 DRPC EXPRESSION FOR GAUSSIAN CASE WITH WASSERSTEIN DIVERGENCE

**Theorem A.4.** *Let $X \sim \mathcal{N}(\mu_X, \sigma_X^2)$ and $Y \sim \mathcal{N}(\mu_Y, \sigma_Y^2)$ be two Gaussian random variables. Let $S \sim \mathcal{N}(\mu_S, \sigma_S^2)$ be an associated classification variable with a covariance of $Cov(X, S) = \theta_1$ and be jointly Gaussian. For the case $d(X, Y) = (X-Y)^2$ and $\phi(p_X, p_Y) = W_2^2(p_X, p_{\hat{X}})$, we have*

$D_W^{(G)}(R, P, C)$

$$= \begin{cases} \sigma_X^2 - \sigma_X^2(1 - 2^{-2R}), \\ \quad \sigma_X - \sqrt{P} \leq \sigma_X\sqrt{1 - 2^{-2R}} \text{ and } C > \frac{1}{2}\log\left(1 - \frac{\theta_1^2(1 - 2^{-2R})}{\sigma_S^2 \sigma_X^2}\right) + h(S) \\ \sigma_X^2 + (\sigma_X - \sqrt{P})^2 - 2\sigma_X(\sigma_X - \sqrt{P})\sqrt{1 - 2^{-2R}}, \\ \quad \sigma_X - \sqrt{P} > \sigma_X\sqrt{1 - 2^{-2R}} \text{ and } C > \frac{1}{2}\log\left(1 - \frac{\theta_1^2(1 - 2^{-2R})}{\sigma_S^2 \sigma_X^2}\right) + h(S) \\ \sigma_X^2 - \frac{\sigma_S^2 \sigma_X^4}{\theta_1^2}(1 - 2^{-2h(S)+2C}), \sigma_X - \sqrt{P} \leq \frac{\sigma_S \sigma_X^2}{\theta_1}\sqrt{1 - 2^{-2h(S)+2C}} \\ \quad \text{and } \frac{1}{2}\log\left(1 - \frac{\theta_1^2}{\sigma_S^2 \sigma_X^2}\right) + h(S) \leq C \leq \frac{1}{2}\log\left(1 - \frac{\theta_1^2(\sigma_X^2 - \sigma_X^2 2^{-2R})}{\sigma_S^2 \sigma_X^4}\right) + h(S) \\ \sigma_X^2 + (\sigma_X - \sqrt{P})^2 - \frac{2\sigma_S \sigma_X^2(\sigma_X - \sqrt{P})}{\theta_1}\sqrt{1 - 2^{-2h(S)+2C}}, \\ \quad \sigma_X - \sqrt{P} > \frac{\sigma_S \sigma_X^2}{\theta_1}\sqrt{1 - 2^{-2h(S)+2C}} \text{ and } \\ \quad \frac{1}{2}\log\left(1 - \frac{\theta_1^2}{\sigma_S^2 \sigma_X^2}\right) + h(S) \leq C \leq \frac{1}{2}\log\left(1 - \frac{\theta_1^2(\sigma_X^2 - \sigma_X^2 2^{-2R})}{\sigma_S^2 \sigma_X^4}\right) + h(S) \\ 0, \quad C > h(S) \text{ and } R > h(X). \end{cases}$$

*Proof.* Based on the proof of Theorem 7, we can show the following Lemma.

**Lemma A.5.** *For the case $X \sim \mathcal{N}(\mu_X, \sigma_X^2)$ with distortion $d(X, Y) = (X - Y)^2$ and perception divergence $\phi(p_X, p_Y) = W_2^2(p_X, p_Y)$, we have*

$$D_W^{(G)}(R, P, C) = \inf_{p_Y} D^{(\infty)}(R, C, q_X, q_Y)$$

$$s.t. \quad \mu_Y = \mu_X, \quad \sigma_Y \leq \sigma_X,$$
$$h(S|Y) \leq C, \quad W_2^2(p_X, p_Y) \leq P.$$

*Proof of Lemma A.5.* The argument parallels that of Lemma A.4. It suffices to show

$$W_2^2(p_X, p_{Y'}) \leq W_2^2(p_X, p_Y),$$

which follows directly from $\mathbb{E}[(X - Y')^2] \leq \mathbb{E}[(X - Y)^2]$. □

By Lemma A.5, it suffices to restrict to $p_Y$ with $\mu_Y = \mu_X$ and $\sigma_Y \leq \sigma_X$ when computing $D_W^{(G)}(R, P, C)$. Further restricting $p_Y$ to Gaussian distributions (Theorem 6) gives

$$D_W^{(G)}(R, P, C) = \min_{\sigma_Y \in [(\sigma_X - \sqrt{P})_+, \sigma_X]} D^{(G)}(R, C, q_X, q_Y). \tag{26}$$

**Case 1.** If $C > \frac{1}{2} \log\left(1 - \frac{\theta_1^2(1 - 2^{-2R})}{\sigma_S^2 \sigma_X^2}\right) + h(S)$, then

$$D^{(G)}(R, C, q_X, q_Y) = (\mu_X - \mu_Y)^2 + \sigma_X^2 + \sigma_Y^2 - 2\sigma_X \sigma_Y \sqrt{1 - 2^{-2R}}.$$

The term $\sigma_Y^2 - 2\sigma_X \sigma_Y \sqrt{1 - 2^{-2R}}$ is decreasing for $\sigma_Y \in [0, \sigma_X \sqrt{1 - 2^{-2R}}]$ and increasing for $\sigma_Y \in [\sigma_X \sqrt{1 - 2^{-2R}}, \infty)$. Thus, the minimizing $\sigma_Y$ in (26) is

$$\sigma_Y = \begin{cases} \sigma_X \sqrt{1 - 2^{-2R}}, & \sigma_X - \sqrt{P} \leq \sigma_X \sqrt{1 - 2^{-2R}}, \\ \sigma_X - \sqrt{P}, & \sigma_X - \sqrt{P} > \sigma_X \sqrt{1 - 2^{-2R}}. \end{cases}$$

Therefore,

$$D_W^{(G)}(R, P, C) = \begin{cases} \sigma_X^2 - \sigma_X^2(1 - 2^{-2R}), \\ \qquad \sigma_X - \sqrt{P} \leq \sigma_X \sqrt{1 - 2^{-2R}} \\ \sigma_X^2 + (\sigma_X - \sqrt{P})^2 - 2\sigma_X(\sigma_X - \sqrt{P})\sqrt{1 - 2^{-2R}}, \\ \qquad \sigma_X - \sqrt{P} > \sigma_X \sqrt{1 - 2^{-2R}}. \end{cases}$$

**Case 2.** If $\frac{1}{2} \log\left(1 - \frac{\theta_1^2}{\sigma_S^2 \sigma_X^2}\right) + h(S) \leq C \leq \frac{1}{2} \log\left(1 - \frac{\theta_1^2(\sigma_X^2 - \sigma_X^2 2^{-2R})}{\sigma_S^2 \sigma_X^4}\right) + h(S)$, then

$$D^{(G)}(R, C, q_X, q_Y) = (\mu_X - \mu_Y)^2 + \sigma_X^2 + \sigma_Y^2 - \frac{2\sigma_S \sigma_X^2 \sigma_Y}{\theta_1} \sqrt{1 - 2^{-2h(S) + 2C}}.$$

Here, $\sigma_Y^2 - \frac{2\sigma_S \sigma_X^2 \sigma_Y}{\theta_1} \sqrt{1 - 2^{-2h(S) + 2C}}$ decreases for $\sigma_Y \in \left[0, \frac{\sigma_S \sigma_X^2}{\theta_1} \sqrt{1 - 2^{-2h(S) + 2C}}\right]$ and increases thereafter. Thus, the minimizing $\sigma_Y$ in (26) is

$$\sigma_Y = \begin{cases} \frac{\sigma_S \sigma_X^2}{\theta_1} \sqrt{1 - 2^{-2h(S) + 2C}}, & \sigma_X - \sqrt{P} \leq \frac{\sigma_S \sigma_X^2}{\theta_1} \sqrt{1 - 2^{-2h(S) + 2C}}, \\ \sigma_X - \sqrt{P}, & \sigma_X - \sqrt{P} > \frac{\sigma_S \sigma_X^2}{\theta_1} \sqrt{1 - 2^{-2h(S) + 2C}}. \end{cases}$$

Therefore,

$$D_W^{(G)}(R, P, C) = \begin{cases} \sigma_X^2 - \frac{\sigma_S^2 \sigma_X^4}{\theta_1^2}(1 - 2^{-2h(S) + 2C}), \\ \qquad \sigma_X - \sqrt{P} \leq \frac{\sigma_S \sigma_X^2}{\theta_1} \sqrt{1 - 2^{-2h(S) + 2C}} \\ \sigma_X^2 + (\sigma_X - \sqrt{P})^2 - \frac{2\sigma_S \sigma_X^2(\sigma_X - \sqrt{P})}{\theta_1} \sqrt{1 - 2^{-2h(S) + 2C}}, \\ \qquad \sigma_X - \sqrt{P} > \frac{\sigma_S \sigma_X^2}{\theta_1} \sqrt{1 - 2^{-2h(S) + 2C}}. \end{cases}$$

**Case 3.** If $C > h(S)$ and $R > h(X)$, then $D_W^{(G)}(R, P, C) = 0$.

In summary, by combining the three cases, the closed-form expression for $D_W^{(G)}(R, P, C)$ is established in Theorem A.4.

□

# B EXPERIMENTAL RESULTS

## B.1 ENTROPY MODEL-BASED RATE ESTIMATION

This experiment directly estimates the compression rate $R$ by computing the entropy of the latent representation rather than relying on an upper bound. The rate is defined as the expected code length under efficient entropy coding: $R = \mathbb{E}_{X \sim p_X}[-\log \mathbb{P}(Q(f(X, U)))]$, where $\mathbb{P}(Q(f(X, U)))$ is the learned entropy model of the encoder outputs. Following Ballé et al. (2018), we parameterize $\mathbb{P}$ as a factorized, non-parametric distribution. Since the achieved rates of entropy coding are typically close to the true entropy (Rissanen & Langdon, 1981; Ballé et al., 2017), we define the training loss directly in terms of entropy:

$$\mathcal{L}_{\text{rate}} = \mathbb{E}\Big[\|X - \tilde{Y}\|^2\Big] - \lambda \log \mathbb{P}(Q(f(X, U))) + \lambda_p W_1(p_Y, p_{\tilde{Y}}) + \lambda_c \, \text{CE}(S, \hat{S}),$$

which jointly balances fidelity, rate regularization, distribution alignment, and classification accuracy.

The rate-accuracy and rate-distortion tradeoff curves for super-resolution (MNIST) and denoising (SVHN) are shown in Figure 7. As expected, increasing the rate yields reconstructions with sharper visual quality and higher classification accuracy. Qualitative samples in Figures 7(c) and 7(f) illustrate this trend: at low rates, MNIST digits appear blurry or ambiguous and SVHN digits remain heavily corrupted by noise, whereas at higher rates, reconstructions become clearer and more faithful to the target distribution.

Additional experiments on CIFAR-10, ImageNet, and KODAK further confirm that the empirical tradeoffs align closely with our theoretical predictions (Figures 8, 9, 10, and 11).

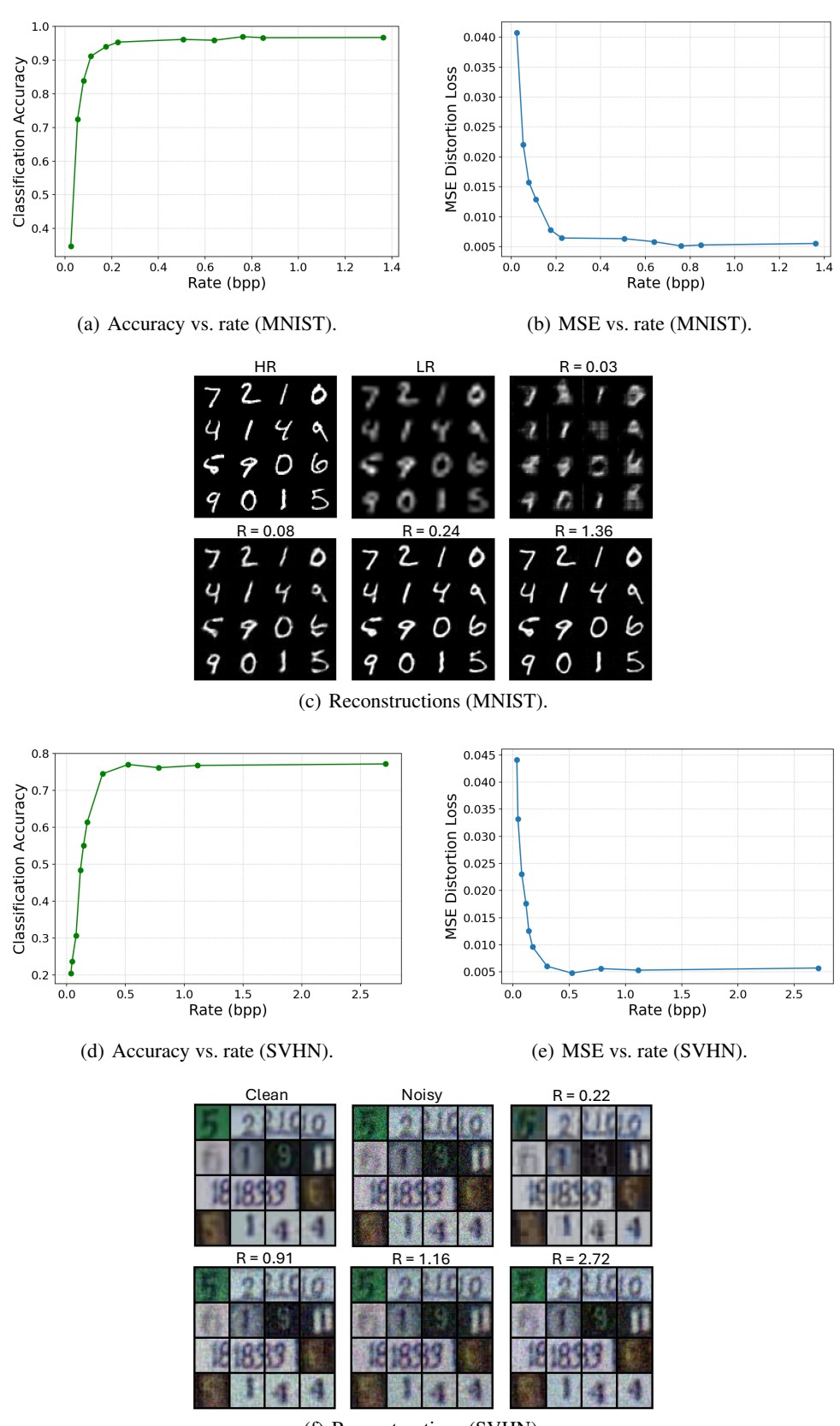

(a) Accuracy vs. rate (MNIST).

(b) MSE vs. rate (MNIST).

(c) Reconstructions (MNIST).

(d) Accuracy vs. rate (SVHN).

(e) MSE vs. rate (SVHN).

(f) Reconstructions (SVHN).

Figure 7: The experimental results of $4\times$ image super-resolution on the MNIST dataset and image denoising on the SVHN dataset corrupted by Gaussian noise, $\mathcal{N}(0, \sigma^2)$ with $\sigma = 25$.

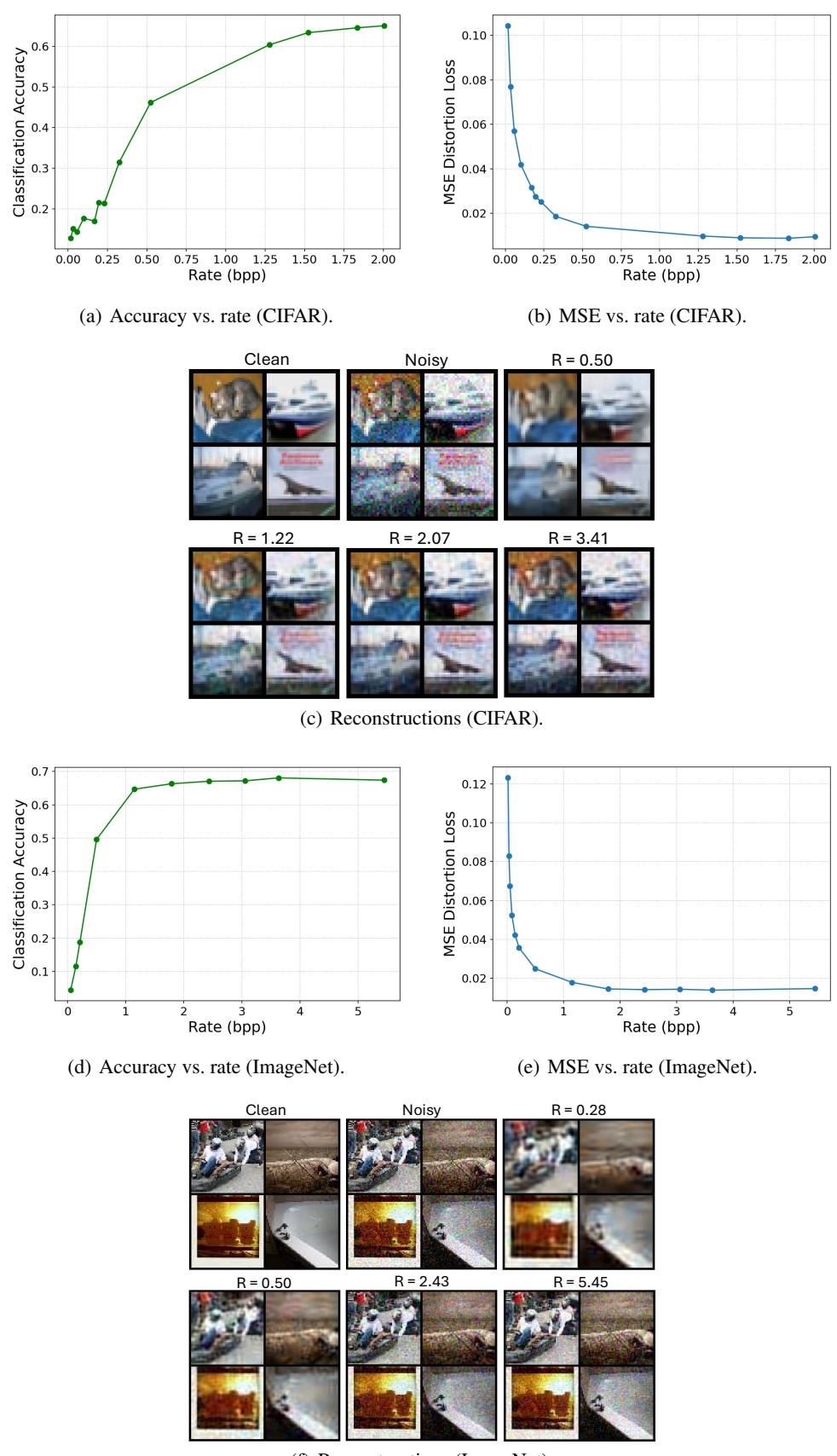

(a) Accuracy vs. rate (CIFAR).

(b) MSE vs. rate (CIFAR).

(c) Reconstructions (CIFAR).

(d) Accuracy vs. rate (ImageNet).

(e) MSE vs. rate (ImageNet).

(f) Reconstructions (ImageNet).

Figure 8: The experimental results of image denoising on the CIFAR-10 and ImageNet datasets corrupted by Gaussian noise, $\mathcal{N}(0, \sigma^2)$ with $\sigma = 25$.

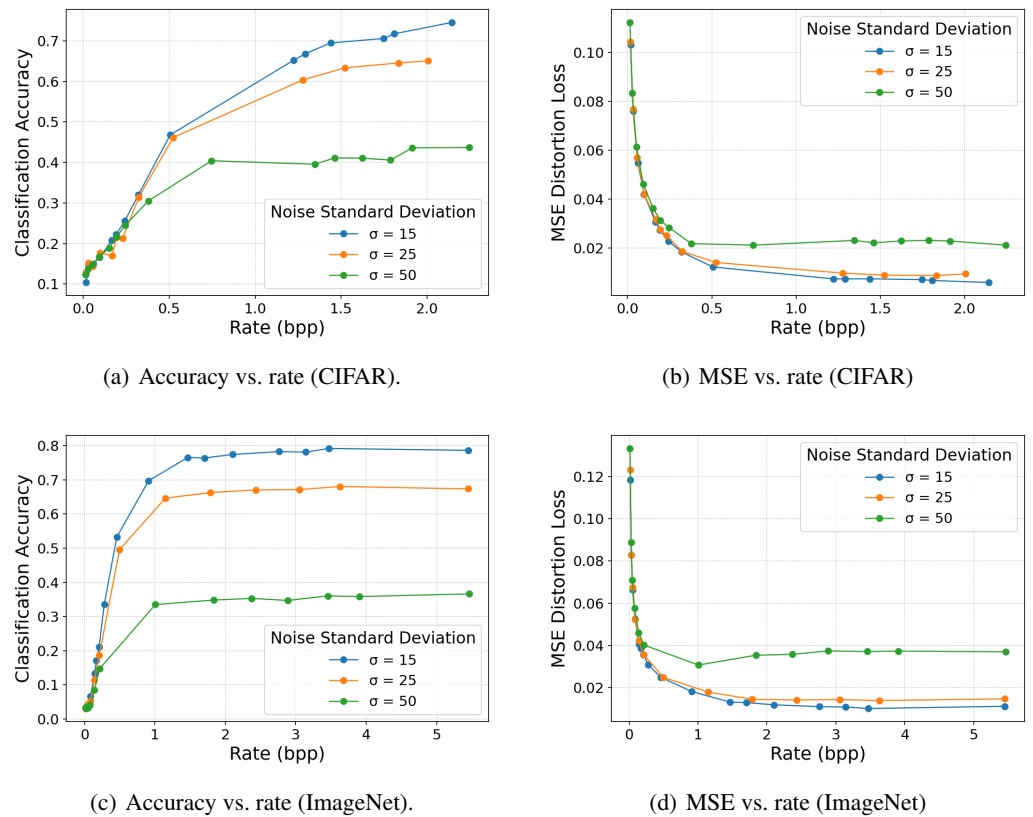

(a) Accuracy vs. rate (CIFAR).

(b) MSE vs. rate (CIFAR)

(c) Accuracy vs. rate (ImageNet).

(d) MSE vs. rate (ImageNet)

Figure 9: The experimental results of image denoising on the CIFAR and ImageNet datasets corrupted by Gaussian noise, $\mathcal{N}(0, \sigma^2)$.

Figure 9 reports denoising results on CIFAR-10 and ImageNet under varying Gaussian noise levels $\sigma \in \{15, 25, 50\}$. Across both datasets, we observe a clear rate-accuracy and rate-distortion tradeoffs.

For CIFAR-10, classification accuracy (Figure 9(a)) improves monotonically with rate, saturating near $0.7$ for $\sigma = 15$ and at lower levels for heavier noise. The corresponding MSE curves (Figure 9(b)) exhibit steep distortion reduction at low rates, followed by a plateau, with higher $\sigma$ consistently yielding larger residual errors.

ImageNet shows a similar pattern (Figures 9(c)–9(d)), though the impact of noise is more pronounced. For $\sigma = 15$, accuracy rises sharply with rate and approaches $\sim 0.8$, while for $\sigma = 50$ it remains below $0.4$ even at the highest rates. MSE again drops quickly at low rates before stabilizing at dataset- and noise-dependent levels.

Overall, stronger noise corruption reduces both achievable accuracy and rate-distortion efficiency. These results align with theoretical predictions: increasing rate enhances reconstruction fidelity and downstream performance, but noise severity imposes fundamental limits.

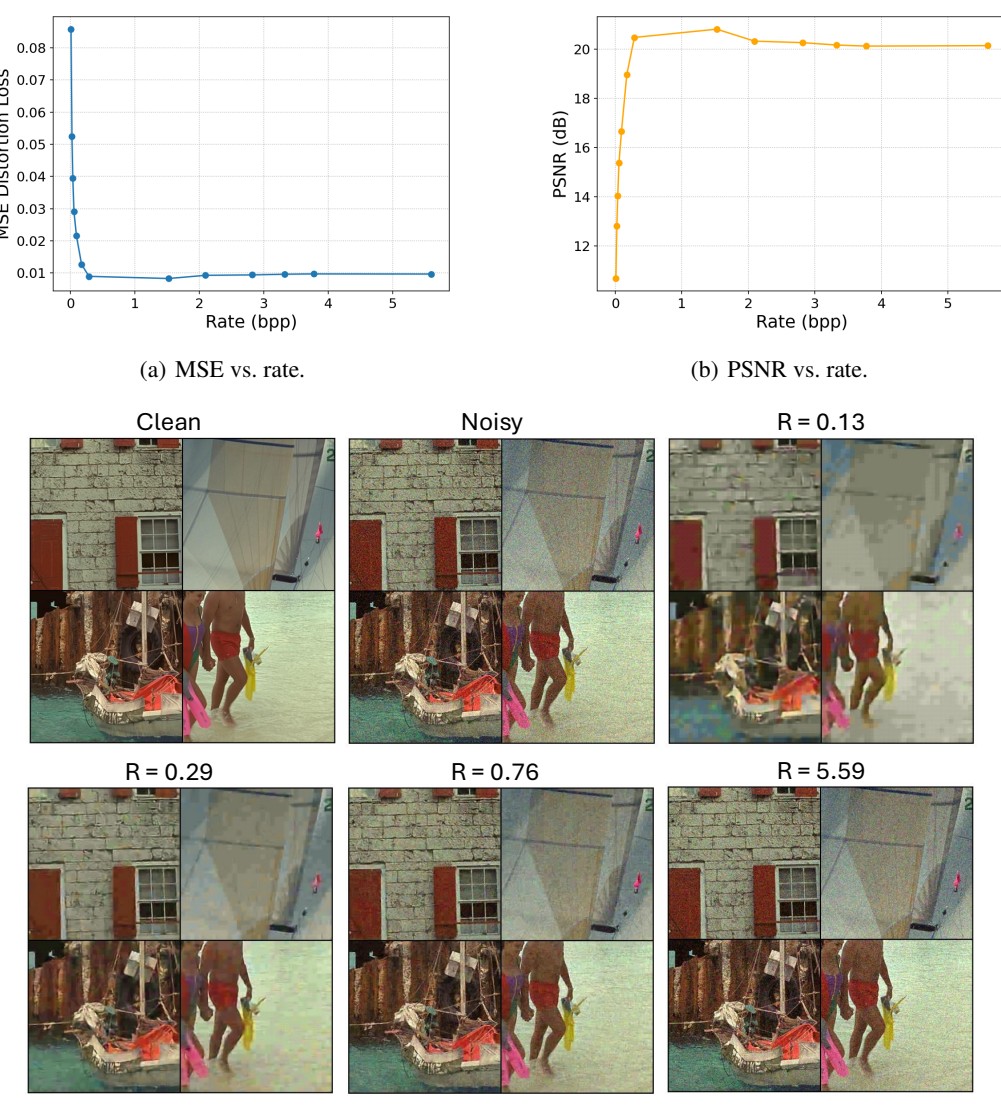

(a) MSE vs. rate.

(b) PSNR vs. rate.

(c) Reconstructions.

Figure 10: The experimental results of image denoising on the KODAK dataset corrupted by Gaussian noise, $\mathcal{N}(0, \sigma^2)$ with $\sigma = 25$.

Figure 10 reports denoising results on the KODAK dataset with Gaussian noise at $\sigma = 25$. The rate-distortion and rate-PSNR curves (Figures 10(a)–10(b)) show sharp distortion reduction as the rate increases from very low values, with PSNR rising rapidly and stabilizing near 20 dB.

Qualitative reconstructions (Figure 10(c)) follow the same progression. At very low rates ($R = 0.13$), outputs remain dominated by residual noise and lose fine details. Moderate rates ($R = 0.29$-$0.76$) yield substantially cleaner images, with sharper edges and restored textures. At high rates ($R = 5.59$), reconstructions closely approximate the ground truth, effectively suppressing noise while preserving perceptual details such as wall textures and color consistency. Additional close-up comparisons in Figure 11 further highlight these improvements.

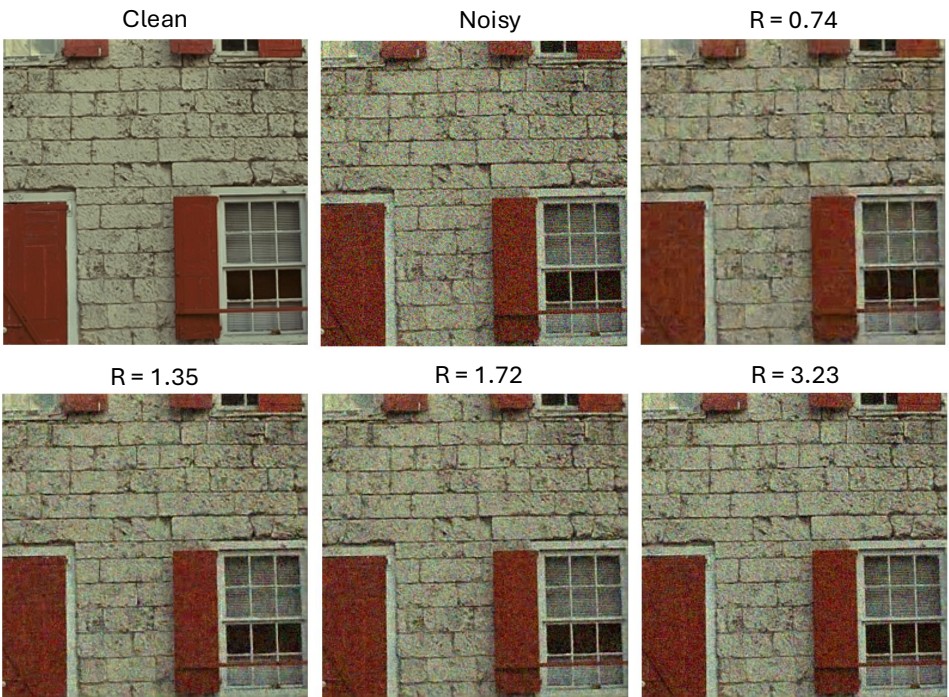

Figure 11: Examples for reconstructions with different rates of image denoising of Kodim01 on the KODAK dataset corrupted by Gaussian noise, $\mathcal{N}(0, \sigma^2)$ with $\sigma = 25$.

## B.2 INPAINTING PROBLEM

We study the inpainting setting, where the goal is to recover missing or occluded regions of an image from partially observed inputs. Let $X \in [0,1]^{3 \times H \times W}$ denote a clean image and $M \in \{0,1\}^{1 \times H \times W}$ a binary mask indicating missing pixels ($M = 1$ for missing, $M = 0$ for observed). The observed input is then given by

$$X_{\text{obs}} = (1 - M) \odot X + Mc,$$

where $c \in [0,1]$ is a fixed fill value (e.g., zero) and $\odot$ denotes element-wise multiplication. The model reconstructs $Y$ such that missing regions are restored while consistency is preserved in observed areas.

### B.2.1 SUPERVISED INPAINTING (WITH CLEAN SOURCES)

When paired clean images are available, training is based on two complementary mean-squared error losses. The *primary* loss focuses on fidelity in the masked region, while the *context* loss ensures coherence in the visible region:

$$\text{MSE}_{\text{miss}} = \frac{\|M \odot (Y - X)\|_2^2}{\max(1, \sum M)},$$

$$\text{MSE}_{\text{ctx}} = \frac{\|(1 - M) \odot (Y - X)\|_2^2}{\max(1, \sum(1 - M))},$$

where $\sum M = \sum_{i=1}^{H} \sum_{j=1}^{W} M_{ij}$ counts the masked pixels, while $\sum(1 - M) = HW - \sum M$ counts unmasked ones. This normalization ensures averaging over the respective region sizes:

$$\text{MSE}_{\text{miss}} = \frac{1}{\max(1, \sum M)} \sum_{i=1}^{H} \sum_{j=1}^{W} M_{ij}(Y_{ij} - X_{ij})^2,$$

$$\text{MSE}_{\text{ctx}} = \frac{1}{\max(1, \sum(1 - M))} \sum_{i=1}^{H} \sum_{j=1}^{W} (1 - M_{ij})(Y_{ij} - X_{ij})^2.$$

The full supervised objective function augments these terms with additional constraints:

$$\mathcal{L}_{\text{super}} = \text{MSE}_{\text{miss}} + \alpha \, \text{MSE}_{\text{ctx}} - \lambda \log \mathbb{P}(Q(f(X, U))) + \lambda_p W_1(p_Y, p_{\tilde{Y}}) + \lambda_c \, \text{CE}(S, \hat{S}), \quad (27)$$

where the entropy term controls rate, the Wasserstein penalty enforces distributional alignment with $p_Y$, and the classification term encourages task-aware reconstructions. This parallels classical inpainting formulations (Pathak et al., 2016; Iizuka et al., 2017; Yu et al., 2018; 2019), but extends them by integrating compression and classification constraints.

### B.2.2 Unsupervised inpainting (without clean sources)

In the absence of paired clean images, we adopt a self-supervised masking strategy inspired by Noise2Self (Batson & Royer, 2019) and related approaches (Krull et al., 2019; Laine et al., 2019). Specifically, we randomly drop a subset of observed pixels $M_{\text{drop}} \subseteq (1 - M)$ and form the input

$$X_{\text{in}} = (1 - M_{\text{drop}}) \odot X_{\text{obs}} + M_{\text{drop}} c,$$

requiring the model to predict the dropped entries. Losses are defined on both the dropped and retained pixels:

$$\text{MSE}_{\text{drop}} = \frac{\|M_{\text{drop}} \odot (Y - X_{\text{obs}})\|_2^2}{\max(1, \sum M_{\text{drop}})},$$

$$\text{MSE}_{\text{id}} = \frac{\|[(1 - M) \odot (1 - M_{\text{drop}})] \odot (Y - X_{\text{obs}})\|_2^2}{\max(1, \sum (1 - M) \odot (1 - M_{\text{drop}}))},$$

where $\sum M_{\text{drop}}$ counts dropped pixels, while $\sum (1 - M) \odot (1 - M_{\text{drop}})$ counts visible ones not dropped. Explicitly,

$$\text{MSE}_{\text{drop}} = \frac{1}{\max(1, \sum M_{\text{drop}})} \sum_{i=1}^{H} \sum_{j=1}^{W} M_{\text{drop},ij}(Y_{ij} - X_{\text{obs},ij})^2,$$

$$\text{MSE}_{\text{id}} = \frac{1}{\max(1, \sum (1 - M) \odot (1 - M_{\text{drop}}))} \sum_{i=1}^{H} \sum_{j=1}^{W} (1 - M_{ij})(1 - M_{\text{drop},ij})(Y_{ij} - X_{\text{obs},ij})^2.$$

The resulting unsupervised objective function is

$$\mathcal{L}_{\text{unsuper}} = \text{MSE}_{\text{drop}} + \alpha_{\text{id}} \text{MSE}_{\text{id}} - \lambda \log \mathbb{P}(Q(f(X, U))) + \lambda_p W_1(p_Y, p_{\tilde{Y}}) + \lambda_c \, \text{CE}(S, \hat{S}), \quad (28)$$

where $\text{MSE}_{\text{id}}$ discourages trivial copying of visible pixels and promotes robust self-prediction.

Supervised inpainting leverages paired clean targets, while unsupervised inpainting relies on redundancy within noisy observations. Both variants naturally integrate into our compression-restoration framework, ensuring rate efficiency, perceptual alignment, and classification fidelity.

### B.2.3 Inpainting Image Results

Figures 12 and 13 report quantitative and qualitative results on the SVHN dataset for supervised and unsupervised inpainting, respectively.

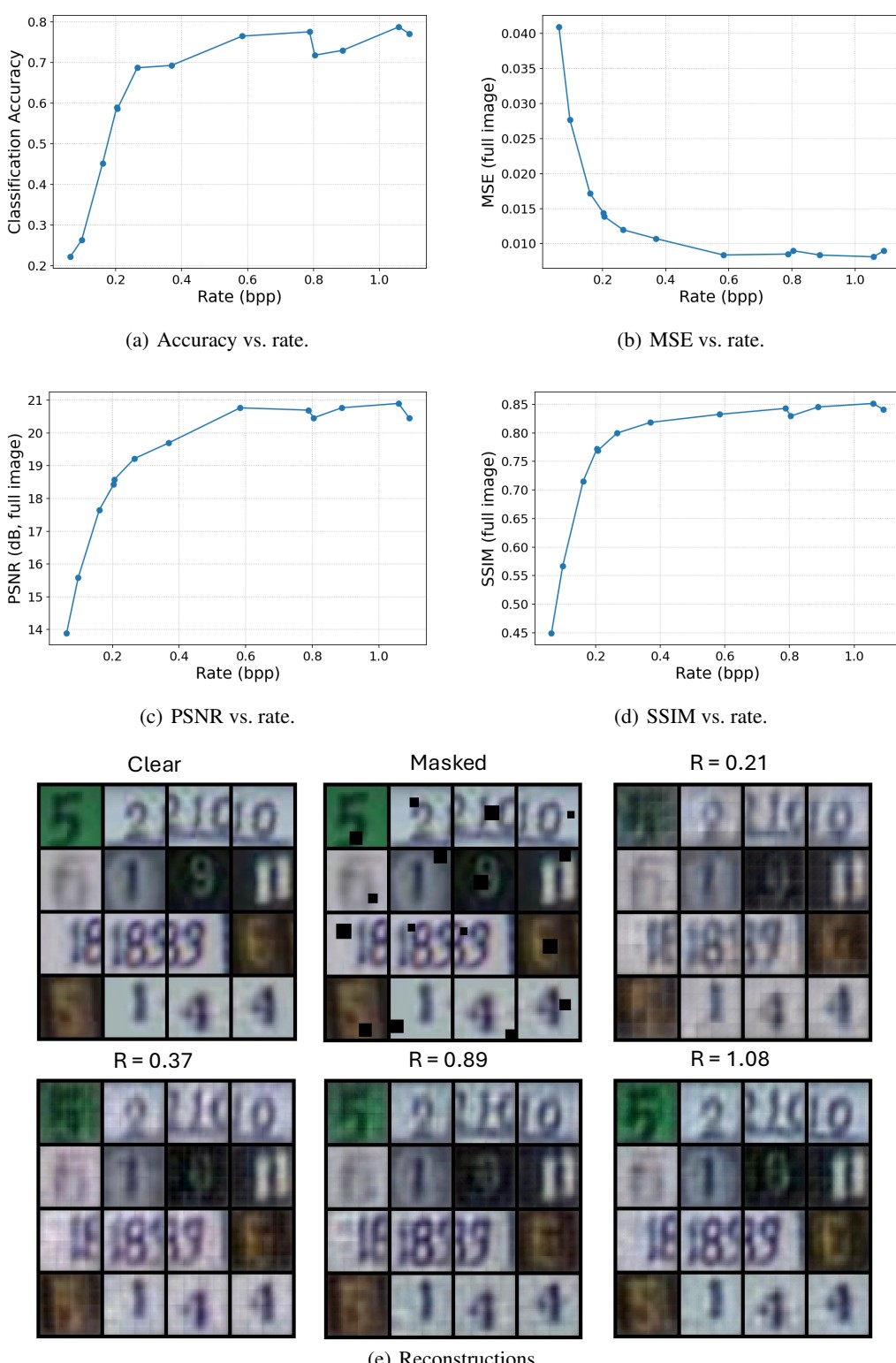

(a) Accuracy vs. rate.

(b) MSE vs. rate.

(c) PSNR vs. rate.

(d) SSIM vs. rate.

(e) Reconstructions.

Figure 12: Supervised inpainting results on SVHN using loss (27). Higher rates consistently improve accuracy, reduce distortion, and yield reconstructions with sharper details and higher perceptual quality.

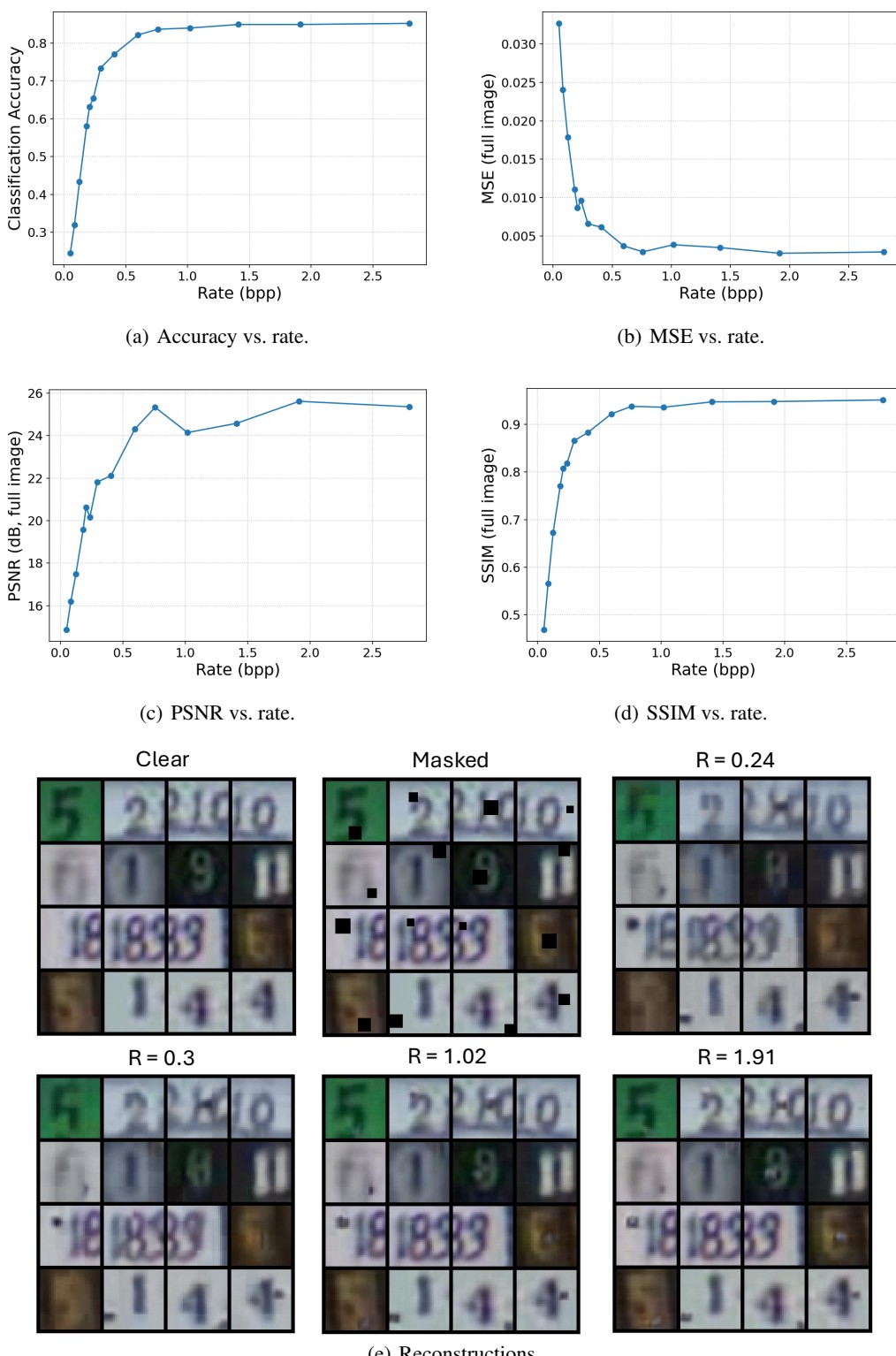

(a) Accuracy vs. rate.

(b) MSE vs. rate.

(c) PSNR vs. rate.

(d) SSIM vs. rate.

(e) Reconstructions.

Figure 13: Unsupervised inpainting results on SVHN using loss (28). Despite lacking clean targets, the self-supervised model exhibits similar trends to the supervised case: accuracy and perceptual quality improve with higher rates, while qualitative examples show more coherent digit structures.

**Supervised inpainting.** With access to paired clean targets, the model is trained using region-specific reconstruction losses ($\text{MSE}_{\text{miss}}$ and $\text{MSE}_{\text{ctx}}$) alongside rate, adversarial, and classification constraints. As shown in Figure 12, this setup produces reconstructions that are sharper and more faithful to ground-truth digits, with missing regions plausibly filled and contextual consistency well preserved.

**Unsupervised inpainting.** Without clean targets, the model adopts a self-supervised masking strategy, minimizing prediction error on randomly dropped subsets ($\text{MSE}_{\text{drop}}$) and regularizing against trivial copying through $\text{MSE}_{\text{id}}$. As illustrated in Figure 13, the reconstructions remain semantically coherent but are generally noisier, with softer digit boundaries than in the supervised setting. This underscores the robustness of self-supervision, albeit with a tradeoff in fine-detail fidelity.

Both supervised and unsupervised settings exhibit the expected RDC tradeoff: at low rates, reconstructions are blurry and task accuracy degrades, while higher rates yield substantial gains in both pixel-level metrics (MSE, PSNR, SSIM) and classification accuracy. Remarkably, the unsupervised variant achieves competitive performance without paired clean data, validating the effectiveness of self-masked training within our compression-restoration framework. Taken together, supervised models capitalize on explicit ground truth to achieve sharper reconstructions, while unsupervised models demonstrate resilience by leveraging internal redundancy, highlighting the flexibility of our framework across supervision regimes.

## B.3 EXPERIMENTAL RESULTS OF SECTION B.1

### B.3.1 DATASET DETAILS

**MNIST.** For $4\times$ super-resolution, we use MNIST (LeCun et al., 1998). Each $28\times28$ digit is downsampled to $7\times7$ via bilinear interpolation and then upsampled back to $28\times28$ to form the low-resolution (LR) input. Reconstructions are evaluated against the original high-resolution (HR) targets.

**SVHN.** We train on SVHN (Netzer et al., 2011) and evaluate on the full $32\times32$ RGB test split. Inputs are corrupted with additive Gaussian noise at $\sigma \in \{15, 25, 50\}$, and reconstructions are compared to the clean targets.

**CIFAR-10.** To study small-scale natural images, we use CIFAR-10 (Krizhevsky et al., 2009), consisting of $32\times32$ images from 10 classes. Inputs are corrupted with Gaussian noise with $\sigma \in \{15, 25, 50\}$, yielding a non-trivial denoising task while remaining computationally efficient.

**ImageNet.** For large-scale evaluation, we adopt ImageNet (Deng et al., 2009), comprising high-resolution images across 1000 categories. Inputs are degraded with Gaussian noise at $\sigma \in \{15, 25, 50\}$.

**KODAK.** Finally, we test on the KODAK dataset (Company, 1991), a benchmark of 24 uncompressed high-resolution images widely used for perceptual quality evaluation due to its fine textures and details.

### B.3.2 TRAINING DETAILS

We adopt a WGAN framework for distributional alignment, jointly training encoder $f$, decoder $g$, and discriminator $d$. By Kantorovich-Rubinstein duality (Villani, 2009), the Wasserstein-1 distance is

$$W_1(p_Y, p_{\tilde{Y}}) = \sup_{\|\nabla d\| \leq 1} \mathbb{E}[d(Y)] - \mathbb{E}[d(\tilde{Y})],$$

where $\tilde{Y} = g(Q(f(X)))$, and the Lipschitz constraint enforced via a gradient penalty (Gulrajani et al., 2017). Unless otherwise specified, WGAN-GP uses $\lambda_{\text{p}} = 0.02$, $\lambda_{\text{GP}} = 10$, $n_{\text{discriminator}} = 5$, and classification weight $\lambda_{\text{c}} = 0.01$. Optimization employs Adam (Kingma & Ba, 2014) with learning rates $5\times10^{-3}$ (autoencoder), $10^{-4}$ (entropy bottleneck), and $10^{-4}$ (discriminator), with $(\beta_1, \beta_2) = (0.5, 0.999)$. We apply gradient clipping (norm 2.0), mixed precision, and compute exact rate bpp during evaluation.

**Super-resolution (MNIST).** Models are trained for 3000 steps with a batch size 512. The classifier is ResNet18-small (grayscale), trained with SGD (learning rate 0.05, cosine schedule with 2-epoch warm-up, momentum 0.9, weight decay $5\times10^{-4}$), batch size 512.

**Denoising (SVHN).** Models are trained for 3000 steps with a batch size 1024. The classifier is ResNet18-small (RGB), trained with SGD (learning rate 0.2, cosine schedule with 5-epoch warm-up, momentum 0.9, weight decay $5\times10^{-4}$), batch size 512.

**Denoising (CIFAR-10).** Models are trained for 3000 steps with a batch size 1024. The classifier is WRN-28-10, trained with SGD (learning rate 0.1, cosine warm-up 5 epochs, momentum 0.9, weight decay $5\times10^{-4}$), batch size 256.

**Denoising (ImageNet).** Models are trained for 10,000 steps with batch size 128, codec learning rate $5\times10^{-4}$, and otherwise identical optimization settings. The classifier is ResNet-18, trained with SGD (learning rate 0.05, cosine warm-up 5 epochs, momentum 0.9, weight decay $10^{-4}$), batch size 256.

**Denoising (BSDS500 → KODAK).** Training is performed on BSDS500 with additive Gaussian noise and evaluation on the 24 KODAK images. We sweep $\lambda$ with batch size 32, using a Conv+GDN autoencoder with an entropy bottleneck (EB). Adam is used for codec and EB parameters (learning rate $10^{-3}$), and the discriminator is trained with Adam (learning rate $10^{-4}$, $(\beta_1, \beta_2) = (0.5, 0.999)$). Here WGAN-GP uses $\lambda_{\mathrm{p}} = 10^{-3}$, $\lambda_{\mathrm{GP}} = 10$, and $n_{\mathrm{discriminator}} = 5$.

### B.3.3 DETAILED RESULTS

Tables 1–5 summarize the effect of the rate-weight parameter $\lambda$ across datasets. For MNIST super-resolution (Table 1), larger $\lambda$ values reduce the rate but incur higher distortion and lower accuracy. SVHN (Table 2) exhibits the same trend, with accuracy dropping sharply under strong rate penalties. CIFAR-10 and ImageNet (Tables 3 and 4) show consistent trade-offs: accuracy remains stable for small $\lambda$ but collapses once rate is heavily constrained. For KODAK (Table 5), PSNR decreases steadily as $\lambda$ increases.

Table 1: Performance across $\lambda$ values for $4\times$ super-resolution on MNIST (Fig. 7(a), Fig. 7(b)).

| $\lambda$ | 0 | 10 | 50 | 100 | 200 | 1000 |
|---|---|---|---|---|---|---|
| Rate (bpp) | 1.3623 | 0.8483 | 0.7609 | 0.6394 | 0.5060 | 0.2262 |
| MSE | 0.0055 | 0.0053 | 0.0051 | 0.0058 | 0.0063 | 0.0064 |
| Accuracy | 0.9670 | 0.9664 | 0.9693 | 0.9587 | 0.9613 | 0.9532 |
| $\lambda$ | 2000 | 5000 | 10000 | 20000 | 50000 | - |
| Rate (bpp) | 0.1755 | 0.1109 | 0.0791 | 0.0537 | 0.0246 | - |
| MSE | 0.0078 | 0.0129 | 0.0158 | 0.0220 | 0.0407 | - |
| Accuracy | 0.9399 | 0.9114 | 0.8389 | 0.7248 | 0.3477 | - |

Table 2: Performance across $\lambda$ values for denoising on SVHN with Gaussian noise, $\mathcal{N}(0, \sigma^2)$ with $\sigma = 25$ (Fig. 7(d), Fig. 7(e)).

| $\lambda$ | 0 | 50 | 100 | 200 | 500 | 1500 |
|---|---|---|---|---|---|---|
| Rate (bpp) | 2.7139 | 1.1143 | 0.7819 | 0.5237 | 0.3048 | 0.1748 |
| MSE | 0.0057 | 0.0053 | 0.0056 | 0.0048 | 0.0060 | 0.0096 |
| Accuracy | 0.7717 | 0.7676 | 0.7615 | 0.7702 | 0.7449 | 0.6144 |
| $\lambda$ | 2000 | 2500 | 5000 | 10000 | 20000 | - |
| Rate (bpp) | 0.1426 | 0.1165 | 0.0792 | 0.0456 | 0.0356 | - |
| MSE | 0.0126 | 0.0176 | 0.0231 | 0.0332 | 0.0441 | - |
| Accuracy | 0.5502 | 0.4841 | 0.3058 | 0.2362 | 0.2044 | - |

Table 3: Performance across $\lambda$ values for denoising on CIFAR-10 with Gaussian noise, $\mathcal{N}(0, \sigma^2)$ with $\sigma = 25$ (Fig. 8(a), Fig. 8(b)).

| $\lambda$ | 10 | 20 | 50 | 100 | 500 |
|---|---|---|---|---|---|
| Rate (bpp) | 2.0058 | 1.8343 | 1.5225 | 1.2778 | 0.5238 |
| MSE | 0.0095 | 0.0088 | 0.0090 | 0.0098 | 0.0141 |
| Accuracy | 0.6507 | 0.6456 | 0.6336 | 0.6038 | 0.4617 |
| $\lambda$ | 1000 | 2000 | 5000 | 50000 | - |
| Rate (bpp) | 0.3248 | 0.1953 | 0.0997 | 0.0168 | - |
| MSE | 0.0187 | 0.0275 | 0.0419 | 0.1043 | - |
| Accuracy | 0.3146 | 0.2155 | 0.1769 | 0.1276 | - |

Table 4: Performance across $\lambda$ values for denoising on ImageNet with Gaussian noise, $\mathcal{N}(0, \sigma^2)$ with $\sigma = 25$ (Fig. 8(d), Fig. 8(e)).

| $\lambda$ | 0 | 5 | 20 | 50 | 100 | 200 | 500 |
|---|---|---|---|---|---|---|---|
| Rate (bpp) | 5.4522 | 3.6295 | 3.0546 | 2.4334 | 1.7890 | 1.1477 | 0.4967 |
| MSE | 0.0147 | 0.0139 | 0.0143 | 0.0141 | 0.0145 | 0.0179 | 0.0249 |
| Accuracy | 0.6737 | 0.6806 | 0.6717 | 0.6705 | 0.6630 | 0.6466 | 0.4959 |
| $\lambda$ | 1500 | 2500 | 5000 | 10000 | 20000 | 50000 | - |
| Rate (bpp) | 0.2103 | 0.1405 | 0.0842 | 0.0503 | 0.0317 | 0.0153 | - |
| MSE | 0.0357 | 0.0423 | 0.0524 | 0.0675 | 0.0829 | 0.1233 | - |
| Accuracy | 0.1876 | 0.1154 | 0.0528 | 0.0438 | 0.0336 | 0.0333 | - |

Table 5: Performance across $\lambda$ values for denoising on KODAK with Gaussian noise, $\mathcal{N}(0, \sigma^2)$ with $\sigma = 25$ (Fig. 10).

| $\lambda$ | 0 | 5 | 10 | 20 | 50 | 100 | 500 |
|---|---|---|---|---|---|---|---|
| Rate (bpp) | 5.5969 | 3.7743 | 3.3253 | 2.8155 | 2.0936 | 1.5241 | 0.2865 |
| MSE | 0.0097 | 0.0097 | 0.0096 | 0.0094 | 0.0093 | 0.0083 | 0.0090 |
| PSNR (dB) | 20.1417 | 20.1227 | 20.1665 | 20.2633 | 20.3248 | 20.8120 | 20.4752 |
| $\lambda$ | 1000 | 2500 | 5000 | 10000 | 20000 | 50000 | - |
| Rate (bpp) | 0.1723 | 0.0929 | 0.0549 | 0.0367 | 0.0232 | 0.0088 | - |
| MSE | 0.0127 | 0.0216 | 0.0291 | 0.0395 | 0.0525 | 0.0858 | - |
| PSNR (dB) | 18.9560 | 16.6618 | 15.3659 | 14.0345 | 12.7972 | 10.6646 | - |

### B.3.4 NEURAL NETWORK ARCHITECTURES

We employ the same set of architectures across all datasets, summarized in Table 6. The codec consists of an encoder $f$ with three convolutional layers followed by a learnable entropy bottleneck, and a decoder $g$ with three deconvolutional layers and GDN activations. The adversarial branch is a WGAN-GP discriminator $d$, implemented as three strided convolutional blocks with LeakyReLU activations, global average pooling, and a linear output. For downstream evaluation, we use ResNet18-small classifiers: a grayscale variant for MNIST and RGB variants for SVHN, CIFAR-10, and ImageNet. Each classifier contains four residual stages (two BasicBlocks per stage) without an initial max-pooling layer, followed by global average pooling and a linear prediction head.

**Computational complexity and scalability of the proposed framework.** Our model uses standard components from learned compression and restoration: a convolutional autoencoder with an entropy model, a WGAN-GP discriminator, and a classifier. Training was performed on two RTX 3090 GPUs under Ubuntu using the PyTorch framework. Both training and inference are dominated by a single forward/backward pass through the autoencoder. The discriminator and classifier reuse the same feature maps and therefore introduce only modest computational overhead, while rate estimation via the entropy model and universal quantization is computationally negligible.

For very high-resolution images, we follow the standard patch/tile strategy used in modern learned codecs, so the overall complexity scales approximately linearly with the number of pixels and remains comparable to existing adversarial denoising and compression methods.

Table 6: Network architectures for autoencoder, discriminator, and classifier of Section B.1.

**Encoder $f$**

| |
|---|
| Conv2D, stride 2, GDN |
| Conv2D, stride 2, GDN |
| Conv2D, stride 2 |
| Entropy Bottleneck (EB) |

**Decoder $g$**

| |
|---|
| Deconv2D, stride 2, GDN |
| Deconv2D, stride 2, GDN |
| Deconv2D, stride 2 |

**Discriminator $d$ (WGAN-GP)**

| |
|---|
| Conv2D, stride 2, LeakyReLU |
| Conv2D, stride 2, LeakyReLU |
| Conv2D, stride 2, LeakyReLU |
| Global AvgPool |

**Classifier (ResNet18-small, gray)**

| |
|---|
| Stem: $3{\times}3$ Conv, BN, ReLU |
| Residual Stage 1: BasicBlock $\times 2$ (stride 1) |
| Residual Stage 2: BasicBlock $\times 2$ (stride 2) |
| Residual Stage 3: BasicBlock $\times 2$ (stride 2) |
| Residual Stage 4: BasicBlock $\times 2$ (stride 2) |
| Global Average Pooling |

## B.4 EXPERIMENTAL RESULTS OF SECTION B.2

### B.4.1 DATASET DETAILS

**SVHN (inpainting).** We evaluate on the SVHN dataset (Netzer et al., 2011) with full $32{\times}32$ RGB images. For each image, a binary mask $M \in \{0, 1\}^{1\times 32\times 32}$ is synthesized ($M{=}1$ indicates missing pixels). Unless otherwise specified, the mask is a random square box with side length sampled uniformly from $[4, 8]$ pixels, and the fill value is set to $c = 0.0$ (black). Reconstructions are evaluated both on the masked region and on the entire image.

### B.4.2 TRAINING DETAILS

**Hyperparameters & schedule.** Unless noted otherwise, training uses $\alpha{=}0.1$, $\lambda_{\mathrm{p}}{=}0.02$, $\lambda_{\mathrm{GP}}{=}10$, and $n_{\mathrm{discriminator}}{=}5$. The codec (encoder-decoder) is optimized with Adam (learning rate $5{\times}10^{-3}$), while the WGAN discriminator is trained with Adam (learning rate $10^{-4}$, $(\beta_1, \beta_2) = (0.5, 0.999)$). Training runs for 3000 steps with batch size 1024, gradient clipping (norm 2.0), and mixed precision. For downstream evaluation, a ResNet18-small classifier (RGB) is trained with SGD (learning rate 0.2, cosine schedule with 5-epoch warm-up, momentum 0.9, weight decay $5{\times}10^{-4}$), batch size 512.

### B.4.3 DETAILED RESULTS

Tables 7 and 8 report the performance of supervised and unsupervised inpainting on the SVHN dataset across varying rate-penalty weights $\lambda$, corresponding to the curves shown in Figures 12 and 13.

Table 7: Supervised inpainting on SVHN across $\lambda$ values (Figure 12).

| $\lambda$ | Rate (bpp) | Accuracy | MSE | PSNR | SSIM |
|---|---|---|---|---|---|
| 1 | 1.0909 | 0.7700 | 0.008996 | 20.4594 | 0.8410 |
| 10 | 0.8038 | 0.7176 | 0.008990 | 20.4626 | 0.8296 |
| 20 | 1.0586 | 0.7871 | 0.008122 | 20.9033 | 0.8517 |
| 50 | 0.8885 | 0.7296 | 0.008372 | 20.7719 | 0.8455 |
| 100 | 0.7879 | 0.7750 | 0.008514 | 20.6986 | 0.8430 |
| 200 | 0.5836 | 0.7644 | 0.008375 | 20.7703 | 0.8326 |
| 500 | 0.3692 | 0.6924 | 0.010719 | 19.6986 | 0.8185 |
| 1000 | 0.2656 | 0.6866 | 0.011975 | 19.2171 | 0.7997 |
| 1500 | 0.2055 | 0.5860 | 0.013879 | 18.5765 | 0.7688 |
| 2000 | 0.2033 | 0.5895 | 0.014372 | 18.4248 | 0.7721 |
| 2500 | 0.1605 | 0.4519 | 0.017193 | 17.6466 | 0.7151 |
| 5000 | 0.0973 | 0.2639 | 0.027659 | 15.5816 | 0.5665 |
| 10000 | 0.0624 | 0.2223 | 0.040927 | 13.8799 | 0.4488 |

Table 8: Unsupervised inpainting on SVHN across $\lambda$ values (Figure 13).

| $\lambda$ | Rate (bpp) | Accuracy | MSE | PSNR | SSIM |
|---|---|---|---|---|---|
| 0 | 2.796 | 0.852 | 0.002910 | 25.362 | 0.951 |
| 5 | 1.914 | 0.849 | 0.002742 | 25.619 | 0.948 |
| 20 | 1.412 | 0.849 | 0.003483 | 24.580 | 0.947 |
| 50 | 1.019 | 0.840 | 0.003849 | 24.147 | 0.936 |
| 100 | 0.760 | 0.836 | 0.002928 | 25.334 | 0.938 |
| 200 | 0.596 | 0.821 | 0.003712 | 24.303 | 0.922 |
| 500 | 0.407 | 0.771 | 0.006137 | 22.120 | 0.883 |
| 1000 | 0.295 | 0.733 | 0.006586 | 21.814 | 0.866 |
| 1500 | 0.237 | 0.654 | 0.009616 | 20.170 | 0.818 |
| 2000 | 0.207 | 0.632 | 0.008653 | 20.628 | 0.808 |
| 2500 | 0.183 | 0.580 | 0.011040 | 19.570 | 0.771 |
| 5000 | 0.125 | 0.433 | 0.017845 | 17.485 | 0.672 |
| 10000 | 0.085 | 0.319 | 0.024026 | 16.193 | 0.566 |
| 20000 | 0.050 | 0.245 | 0.032689 | 14.856 | 0.469 |

### B.4.4 NEURAL NETWORK ARCHITECTURES

The codec is a lightweight convolutional autoencoder with an entropy bottleneck. The encoder consumes the 4-channel tensor $[X_{\text{obs}}; M]$, and the decoder outputs an RGB reconstruction $\hat{Y}$. A small WGAN-GP discriminator $d$ is used for distribution alignment, and a ResNet18-small classifier evaluates downstream task performance.

Table 9: Architectures for SVHN inpainting with **nb** denotes the number of latent bottleneck channels.

**Encoder $f$ (4→nb)**

| |
|---|
| Conv3×3, stride 2, GDN |
| Conv3×3, stride 2, GDN |
| Conv3×3, stride 2 |
| Entropy Bottleneck |

**Decoder $g$ (nb→3)**

| |
|---|
| Deconv3×3, stride 2, GDN |
| Deconv3×3, stride 2, GDN |
| Deconv3×3, stride 2 |

**Discriminator $d$ (WGAN-GP)**

| |
|---|
| Conv, stride 2, LeakyReLU |
| Conv, stride 2, LeakyReLU |
| Conv, stride 2, LeakyReLU |

**Classifier (ResNet18-small)**

| |
|---|
| 3×3 stem (stride 1), BN, ReLU |
| 4 residual stages (BasicBlock ×2) |
| Strides: $(1, 2, 2, 2)$ |
| Global Average Pooling |

## B.5 EXPERIMENTAL RESULTS OF SECTION 5

For these experiments, we follow the setup of Liu et al. (2022) with the following modifications. For super-resolution, models are trained for 100 epochs with penalty weight $\lambda = 0.05$ across all rates. The learning rate is initialized at $10^{-4}$, decayed by a factor of 5 after 30 epochs, and optimized using Adam (Kingma & Ba, 2014). The denoising setup mirrors super-resolution, except with $\lambda = 0.03$ and learning rate decay applied after 40 epochs.

The classifier is a ResNet20 trained jointly with the GAN. It is updated alongside the discriminator using reconstructed images and ground-truth labels, optimized with SGD (learning rate $10^{-2}$, momentum $0.9$, weight decay $10^{-4}$). Training runs for 100 epochs with batch size $64$ using standard cross-entropy loss. The classifier architecture is summarized in Table 10.

Table 10: Classifier architecture.

| **Classifier** |
| --- |
| Input |
| Conv2D (10 filters, kernel=5), ReLU |
| MaxPool2D (kernel=2) |
| Conv2D (10 filters, kernel=5), ReLU |
| MaxPool2D (kernel=2) |
| Flatten |
| Linear, ReLU |
| Linear, Softmax |

### B.5.1 DETAILED RESULTS

Tables 11 and 12 summarize the empirical trade-offs between distortion and accuracy under different rate budgets. For $4\times$ super-resolution on MNIST (Table 11), higher rates consistently reduce MSE while improving classification accuracy. For SVHN denoising with additive Gaussian noise $\mathcal{N}(0, 20)$ (Table 12), accuracy rises steadily with rate and distortion decreases correspondingly, reflecting the transition from rate-limited to task-limited regimes. These observations closely match the theoretical predictions shown in Figures 6(a), 6(b), 6(d), and 6(e).

Table 11: MSE distortion and accuracy at different rates for $4\times$ image super-resolution on MNIST (Fig. 6(a), Fig. 6(b)).

| Rate Upper Bound | 4 | 8 | 12 | 16 | 20 | 24 | 28 | 32 |
| --- | --- | --- | --- | --- | --- | --- | --- | --- |
| MSE | 0.0718 | 0.0603 | 0.0535 | 0.0494 | 0.0477 | 0.0471 | 0.0444 | 0.0451 |
| Accuracy | 0.8756 | 0.9158 | 0.9234 | 0.9288 | 0.9309 | 0.9309 | 0.9291 | 0.9317 |

Table 12: MSE distortion and accuracy at different rates for denoising on SVHN with Gaussian noise, $\mathcal{N}(0, \sigma^2)$ with $\sigma = 20$ (Fig. 6(d), Fig. 6(e)).

| Rate Upper Bound | 4 | 10 | 20 | 30 | 40 | 50 | 60 | - |
| --- | --- | --- | --- | --- | --- | --- | --- | --- |
| MSE | 0.0364 | 0.0300 | 0.0226 | 0.0196 | 0.0189 | 0.0164 | 0.0170 | - |
| Accuracy | 0.1959 | 0.2115 | 0.3585 | 0.4459 | 0.5001 | 0.5389 | 0.5043 | - |
| Rate Upper Bound | 70 | 80 | 90 | 100 | 110 | 120 | - | - |
| MSE | 0.0147 | 0.0142 | 0.0133 | 0.0127 | 0.0133 | 0.0126 | - | - |
| Accuracy | 0.6328 | 0.6442 | 0.6863 | 0.7056 | 0.6965 | 0.7070 | - | - |

## C    ADDITIONAL EXPERIMENTAL RESULTS

In this section, we provide supplementary empirical results that further validate our theoretical analysis and support the claims made in the main paper.

### C.1    HEATMAP VISUALIZATION OF THE EMPIRICAL RDC FUNCTION IN FIGURE 6.

The heatmap visualizations of the empirical RDC/CDR functions for the $4\times$ super-resolution task on MNIST and the denoising task on SVHN with Gaussian noise $\sigma = 20$ are presented in Figure 14 and Figure 15, respectively. These results clearly illustrate the qualitative tradeoffs among distortion, rate, and classification accuracy: enforcing a tighter classification constraint (i.e., requiring higher accuracy) at a fixed rate consistently increases the achievable distortion. This behavior aligns with the theoretical structure of the RDC tradeoff.

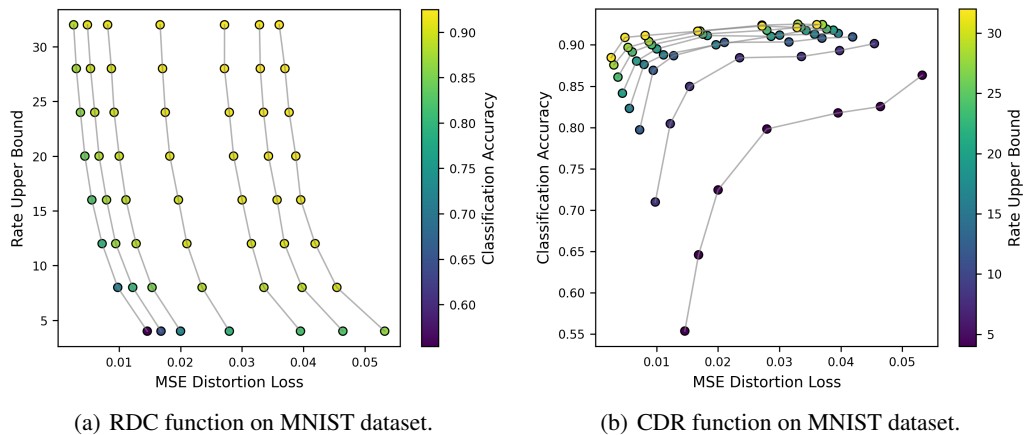

(a) RDC function on MNIST dataset.          (b) CDR function on MNIST dataset.

Figure 14: Experimental results: $4\times$ super-resolution on MNIST. Higher rates yield better reconstructions and improved classification performance.

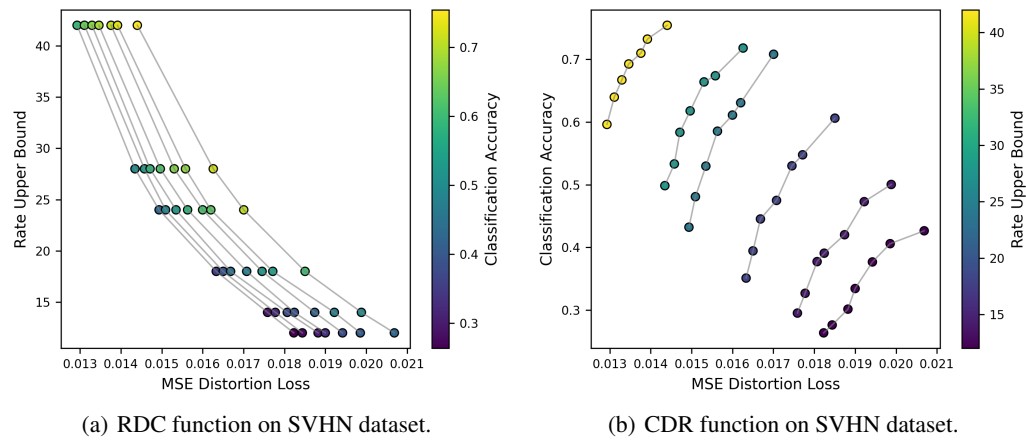

(a) RDC function on SVHN dataset.          (b) CDR function on SVHN dataset.

Figure 15: Experimental results: denoising on SVHN with Gaussian noise with $\sigma = 20$. Higher rates yield better reconstructions and improved classification performance.

### C.2    CONTOUR VISUALIZATION OF THE EMPIRICAL RDC FUNCTION IN FIGURE 6.

The contour visualizations of the generated samples for the MNIST dataset on the super-resolution task and for the SVHN dataset on the denoising task are shown in Figure 16. The equi-rate lines plot-

ted on $R(D, C)$ highlight the inherent tradeoff between distortion and classification performance: for any fixed rate constraint, improving classification accuracy (i.e., achieving lower cross-entropy) requires accepting higher distortion, and conversely, reducing distortion necessitates a weaker classification constraint.

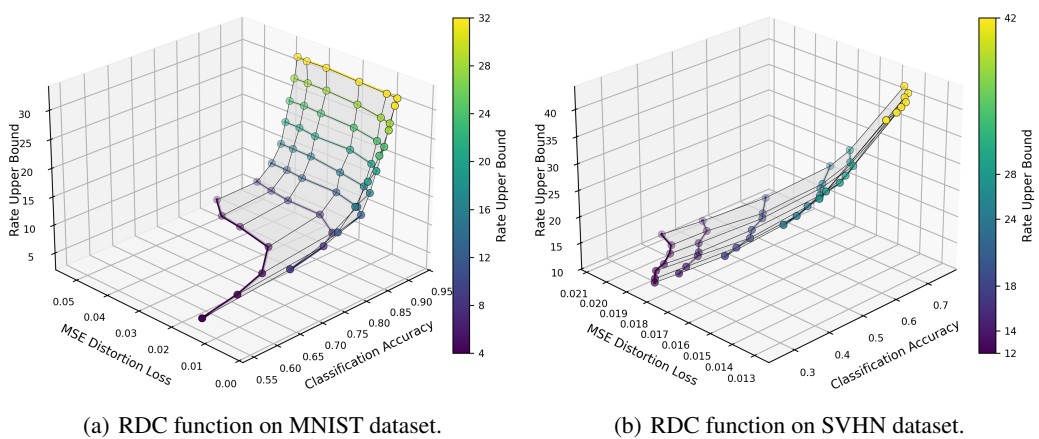

(a) RDC function on MNIST dataset.        (b) RDC function on SVHN dataset.

Figure 16: The RDC functions on the MNIST and SVHN datasets, together with the equi-rate lines plotted on $R(D, C)$, highlight the tradeoff between distortion and classification performance at any fixed rate constraint.

### C.3 QUANTITATIVE COMPARISON BETWEEN THEORETICAL AND EMPIRICAL DRC CURVES IN FIGURES 3 AND 4.

We conducted experiments to quantitatively compare the closed-form DRC curves with empirical estimates on synthetic data. The results are summarized in Figure 17. In both the Bernoulli and Gaussian settings, the theoretically predicted DRC curves closely match the empirical estimates, confirming the correctness of the closed-form expressions.

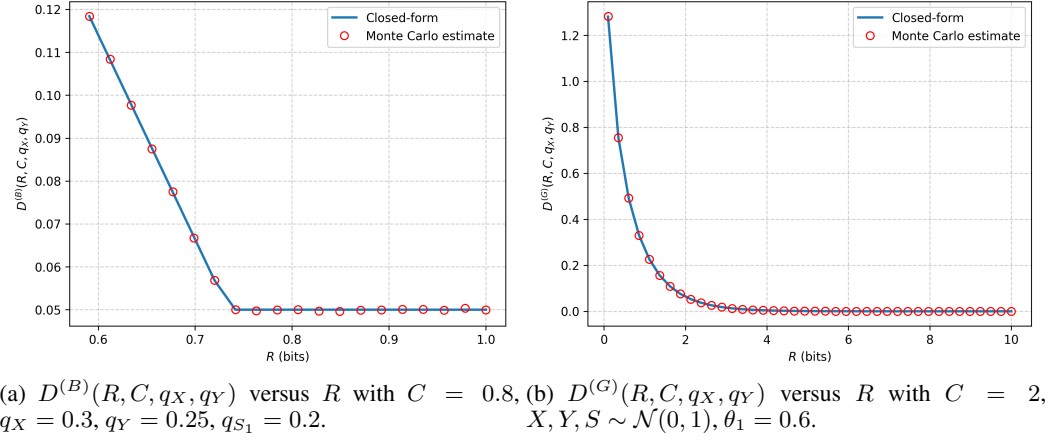

(a) $D^{(B)}(R, C, q_X, q_Y)$ versus $R$ with $C = 0.8$, (b) $D^{(G)}(R, C, q_X, q_Y)$ versus $R$ with $C = 2$, $q_X = 0.3$, $q_Y = 0.25$, $q_{S_1} = 0.2$.      $X, Y, S \sim \mathcal{N}(0, 1), \theta_1 = 0.6$.

Figure 17: Closed-form DRC curves versus empirical estimates for Bernoulli and Gaussian distributions.

**Bernoulli case.** We consider the one-shot setting with $X \sim \mathrm{Bern}(q_X)$, $Y \sim \mathrm{Bern}(q_Y)$, $S = X \oplus S_1$, and $S_1 \sim \mathrm{Bern}(q_{S_1})$, under Hamming distortion and constraints $(R, C)$. For each admissible pair $(R, C)$, Theorem 2 specifies the optimal joint distribution of $(X, S, Y)$. We sample from this

joint distribution, estimate the mutual information $I(X;Y)$ and conditional entropy $H(S|Y)$, and compute the empirical distortion $\hat{D}_{\text{emp}}(R,C)$. As seen in Figure 17(a), the empirical points align closely with the theoretical DRC curve $D^{(B)}(R,C,q_X,q_Y,q_{S_1})$.

**Gaussian case.** We similarly validate Theorem 4 for jointly Gaussian $(X,Y,S)$ with quadratic distortion. For each $(R,C)$, we construct the optimal coupling prescribed by the theorem, draw i.i.d. samples, and estimate $I(X;Y)$, $H(S|Y)$, and the empirical distortion. Figure 17(b) shows that the empirical distortions again match the theoretical curve $D^{(G)}(R,C,q_X,q_Y)$ with high fidelity.

### C.4 EXPERIMENTS USING THE EXACT CONDITIONAL ENTROPY $H(S|Y)$ IN PLACE OF CROSS-ENTROPY

**Training with $H(S|Y)$ directly.** We note that $H(S|Y)$ measures the uncertainty of the true label after observing the reconstruction $y$, i.e., $H(S|Y) = \mathbb{E}_{Y\sim p_\phi}\Big[ - \sum_s p_\phi(s|y) \log p_\phi(s|y)\Big]$, which depends on the true posterior $p_\phi(s|y)$.

We now jointly train the classifier with the encoder/decoder. Rather than using a pretrained classifier, we alternate between (i) training the encoder-decoder with a fixed classifier and (ii) training the classifier using $H(S|Y)$ as loss function with a fixed encoder-decoder. Theoretically, the classifier should converge toward the true posterior $p_\phi(s|y)$ for the current reconstructions $y$. In the limit of an optimal classifier (infinite capacity and converged training), the cross-entropy becomes equal to $H(S|Y)$ (Wang et al., 2025).

The resulting tradeoff curves using the exact $H(S|Y)$ in Figure 18 for the super-resolution task on MNIST and Figure 19 for the denoising task on SVHN are consistent with those obtained using cross-entropy and confirm that decreasing $H(S|Y)$ corresponds to improved classification accuracy. These experiments further demonstrate the upper-bound relationship between $H(S|Y)$ and $\text{CE}(s,\hat{s})$ (Wang et al., 2025).

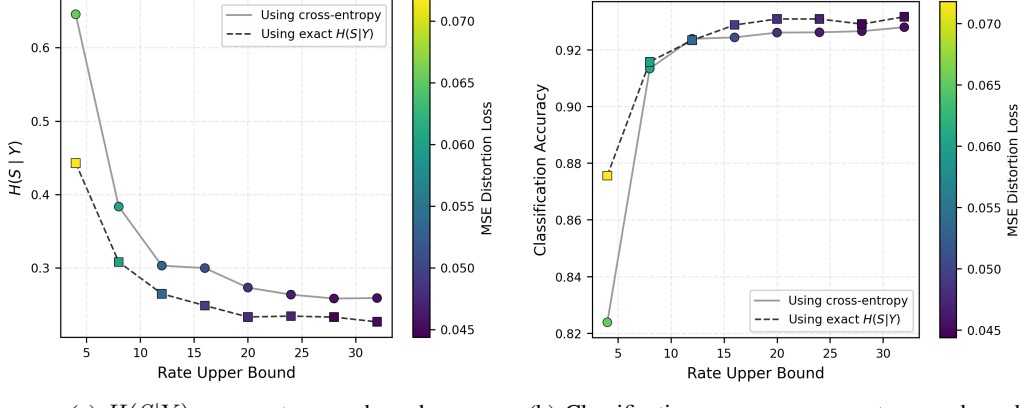

(a) $H(S|Y)$ versus rate upper bound.  (b) Classification accuracy versus rate upper bound.

Figure 18: Experimental results: $4\times$ super-resolution on MNIST. Minimizing the conditional entropy $H(S|Y)$ is equivalent to maximizing the classification accuracy.

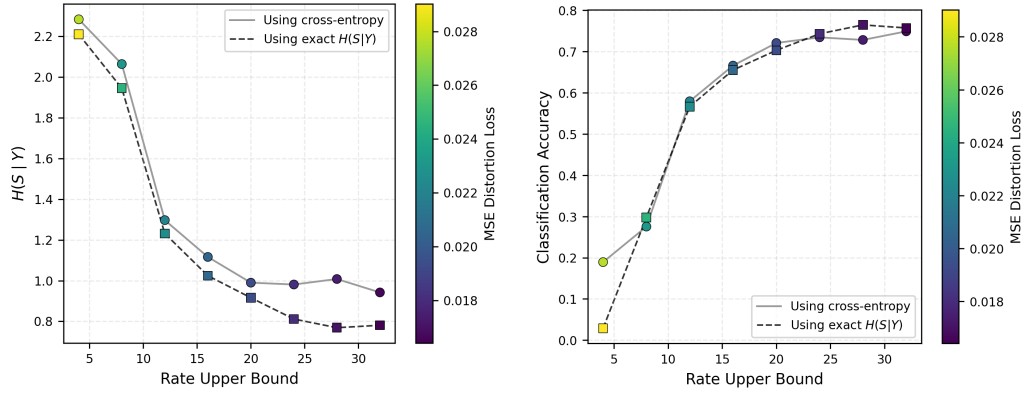

(a) $H(S|Y)$ versus rate upper bound.

(b) Classification accuracy versus rate upper bound.

Figure 19: Experimental results: denoising on SVHN with Gaussian noise with $\sigma = 20$. Minimizing the conditional entropy $H(S|Y)$ is equivalent to maximizing the classification accuracy.

## C.5 EMPIRICAL VALIDATION ON REAL-WORLD IMAGING AND PHOTOGRAPHIC NOISE.

**Fluorescence microscopy images.** In addition to standard benchmarks, we include experiments on real-world fluorescence microscopy images (Mouse Nuclei) (Buchholz et al., 2020), which exhibit statistics and structures very different from natural photographic images, under Gaussian noise $\mathcal{N}(0, \sigma^2)$. As shown in Table 13, our method achieves decent denoising performance under Gaussian noise across different noise levels, with consistently high peak signal-to-noise ratio (PSNR) and structural similarity index measure (SSIM) (Wang et al., 2004), as well as low perceptual distortions such as learned perceptual image patch similarity (LPIPS) (Zhang et al., 2018) and deep image structure and texture similarity (DISTS) (Ding et al., 2020), averaged over 67 test images.

| $\sigma$ | PSNR (dB) ↑ | SSIM ↑ | LPIPS ↓ | DISTS ↓ |
|---|---|---|---|---|
| 10 | 33.0340 | 0.8052 | 0.0443 | 0.1401 |
| 20 | 30.5880 | 0.8028 | 0.0734 | 0.1675 |

Table 13: Performance values for denoising on Mouse Nuclei dataset.

**Real-world smartphone images.** To further probe robustness under real-world photographic noise, we include experiments on the SIDD smartphone image dataset (Abdelhamed et al., 2018), which contains complex, signal-dependent noise patterns arising from real camera pipelines, averaged over 10 test images. As shown in Table 14, these additional experiments demonstrate that the proposed framework remains effective beyond controlled synthetic settings and applies to challenging real-world compression/restoration scenarios.

| PSNR (dB) ↑ | SSIM ↑ | LPIPS ↓ | DISTS ↓ |
|---|---|---|---|
| 33.605 | 0.9038 | 0.3233 | 0.2366 |

Table 14: Performance values on the SIDD dataset.

## C.6 COMPARISON WITH OTHER COMPRESSION AND DENOISING APPROACHES.

A quantitative comparison on the KODAK dataset corrupted by Gaussian noise with $\sigma = 25$ is shown in Table 15. We compare our method against the non-learning baselines JPEG-2K (Taubman et al., 2002) and BM3D (Dabov et al., 2007), as well as the recent unsupervised denoising approaches DeCompress (Zafari et al., 2025a) and OTDenoising (Wang et al., 2023b). While these methods differ in objectives and constraints, the comparison is informative regarding tradeoffs in distortion, perceptual quality, and rate. In particular, although BM3D achieves higher PSNR, which

is unsurprising, as it does not explicitly penalize the rate. On the other hand, our model performs favorably on perceptual metrics such as LPIPS, DISTS, and perceptual index (PI) (Ma et al., 2017), thanks to the WGAN-GP discriminator. For example, our PI score is significantly better than BM3D and DeCompress and is close to the best unsupervised method, OTDenoising, using a smaller representation rate. Overall, this comparison demonstrates that our framework preserves competitive restoration quality and perception.

| Method | PSNR (dB) ↑ | SSIM ↑ | LPIPS ↓ | DISTS ↓ | PI ↓ |
|---|---|---|---|---|---|
| JPEG-2K | 26.4408 | 0.7357 | 0.4018 | 0.2419 | 7.4794 |
| BM3D | **31.8757** | **0.8687** | 0.2235 | 0.1640 | 2.6503 |
| DeCompress | 27.8315 | 0.7519 | 0.2627 | 0.1967 | 2.7979 |
| OTDenoising | 31.2893 | 0.8677 | **0.1150** | **0.1032** | **2.0095** |
| Ours | 27.8961 | 0.8035 | 0.1987 | 0.1638 | 2.1670 |

Table 15: Comparison of denoising performance on the KODAK dataset with Gaussian noise $\mathcal{N}(0, \sigma^2)$, $\sigma = 25$. Best values are in **bold** and second-best values are underlined.

