# OpenReview forum: "Cross-Domain Lossy Compression via Rate- and Classification-Constrained Optimal Transport"
_ICLR.cc/2026/Conference — ICLR 2026 Oral_

### Official Review · Reviewer_AfGP · 2025-10-27

**Soundness:** 3
**Presentation:** 3
**Contribution:** 1
**Rating:** 2
**Confidence:** 4

**Summary:**

## Summary
* This paper studies the rate distortion characteristic of a problem named "cross-domain lossy compression". In this problem, the authors considers joint optimization of distortion loss, classification uncertainty and even perception divergence.
* The authors derive the closed formed solution to their function using constrained pptimal transport for Bernouli and Gaussian source.
* The authors validate their theory through a somewhat different experimental setup using CE loss instead of classification uncertainty.

**Strengths:**

## Strength
* This paper is well written and the study of the problem is comprehensisve. Both one-shot and asymptotic cases are considered.
* The authors provide detailed example for both discrete and continous source (Bernouli and Gaussian).
* The experimental design covers different degradation and different image dataset.

**Weaknesses:**

## Weakness
* First, the mathematical formulation of the proposed approach is deviated from its objective.
  * More precisely, the authors claim that the formulation considers the "Classification utility" and can be useful in practical scenario. However, the constraint imposed for the "classification" is $H(S|Y)$, the entropy of posterior distribution of class label conditioned on reconstruction. This $H(S|Y)$ has really nothing to do with classification utility as the classification in rate-distortion-classification (RDC) or rate-distortion-perception-classification
(RDPC) [Task-oriented lossy compression with data, perception, and classification constraints].  In general, the constraint $H(S|Y)$ seems to be close to a measure of "Inception Score" [Improved Techniques for Training GANs], which is only a measure of how the reconstruction is close to any label, instead of how the reconstruction is close to the label of the source.
  * For a MNIST example, we can let the decoder to constantly output random images with certainly recognizable digits. In that case, $\forall C, H(S|Y)=0\le C$. However, the reconstruction really has nothing todo with the input. And the $H(S|Y)$ really has no correlation with classification accuracy.
* Second, the mathematical formulation of the proposed approach is also deviated from experimental results. Despite the $H(S|Y)$ in their theory is upperbounded by the CE loss they use in experimental result, the nature of $H(S|Y)$ and CE is very different. As I have explained previously, $H(S|Y)$ only measures how certain the model is on any label, given the reconstruction. However, CE is really about the classification accuracy of given label. I doubt that the authors prefe CE loss instead of $H(S|Y)$ in their experimental results just to get a better result. I can not see why it is necessay to use CE over $H(S|Y)$, as $H(S|Y)$ is really not hard to implement.
* Overall, I do appreciate the effort the authors have spend in solving the formulated problem. However, the current problem formulation using $H(S|Y)$ is too different from the CE loss they use in experiments. Besides, the current problem formulation can not achieve the "joint restoration + classification" target that the authors have claimed. I recommend to reject this paper, as its theory deviates too much from their claim, and is too different from their empirical results.

**Questions:**

## Questions
* Is that possible to derive the theory with proper classification loss, such as CE?
* What is the experimental result like using the real $H(S|Y)$ loss instead of CE? This is really not difficult to implement and the authors should present the honest result using the real loss following their theory.

---

> ### Author Response · Authors · 2025-11-19
> **Author Response -- Part 1**
>
> We thank the reviewer for their feedback. We now address the points noted in the weaknesses section. In our response, we also address the reviewer’s questions regarding the use of the $H(S \mid Y)$ loss, as mentioned in the Questions.
>
> > **W1.** Use of $H(S\mid Y)$ for classification constraint.
>
> We believe that although the Inception Score is related to conditional entropy, it is primarily used in GAN-style generative models. On the other hand, we argue that the conditional entropy $H(S \mid Y)$ is fundamentally connected to classification performance, specifically
> the estimation error. Let $\hat S = g_\theta(Y)$ be any estimator of $S$ based on the observation $Y$. By Fano's inequality [1],
> $$
> \Pr(S \neq \hat S)
> \ge
> \frac{H(S \mid Y) - 1}{\log(M - 1)},
> $$
> where $M$ is the number of classes. In practice, a good estimator is often used and as such the Fano's bound is tight. Therefore, achieving a small conditional entropy $H(S \mid Y)$ directly
> enables the design of classification algorithms with correspondingly low
> error.
>
> Regarding the reviewer's MNIST example, if a decoder always outputs an image containing a clearly recognizable digit, then $H(S \mid Y) = 0$, because the digit can always be recovered perfectly from $Y$. This is true. This example, however, corresponds more closely to a generative model that freely selects a representative image for each class. In our setting, such freedom is not available. The representation $Y$ must remain close to the original data $X$ in some distortion sense (e.g., MSE), and the amount of information encoded in the latent variable $Z$ is limited. Our framework assumes a Markov chain
> $$
> S \rightarrow X \rightarrow Z \rightarrow Y,
> $$
> where $Z$ is a compressed representation of $X$, and $Y$ is a
> reconstruction constrained both by distortion limits (similarity to $X$)
> and by a bound on representational bitrate.
>
> For example, consider compressing an image of a cow: we aim to encode it using a limited number of bits (via $Z$), reconstruct it as $Y$ so that it remains visually close to the original $X$, and still retain enough information in $Y$ so that the object can be recognized as a cow with high accuracy. In this scenario, one cannot simply generate an arbitrary “cow-like” image in isolation; the reconstruction must remain faithful to the specific input while preserving the ability to classify it. These constraints make $H(S \mid Y)$ the appropriate quantity for characterizing the fundamental trade-off between compression, reconstruction accuracy, and classification performance.
>
> [1] Thomas M Cover and Joy A Thomas. Elements of information theory. John Wiley & Sons, 1999.

---

> ### Author Response · Authors · 2025-11-19
> **Author Response -- Part 2**
>
> > **W2.** Use of cross-entropy instead of $H(S\mid Y)$.
>
> Regarding the comment on using cross-entropy (CE) instead of $H(S \mid Y)$, there are several points that justify the use of CE in our framework.
>
> First, most state-of-the-art classification DNNs are trained using cross-entropy loss. If CE leads to the best-performing classifiers in practice, then using CE, even when $H(S \mid Y)$ is the theoretical quantity of interest, is reasonable. In this sense, CE acts as a practical surrogate for controlling $H(S \mid Y)$.
>
> Second, by relying on a well-pretrained classifier trained with CE, we can leverage a proven classification engine without the need to jointly train the classifier together with our encoder-decoder architecture. This increases stability and avoids unnecessary coupling between components during optimization.
>
> Finally, we can upper bound the conditional entropy constraint with CE as follows [2,3]. Let $p_\phi(s,y)$ be the joint distribution produced by our encoder and decoder, and let $p_\psi(s \mid y)$ be an approximate conditional distribution of $p_\phi(s \mid y)$ given by a pre-trained classifier. The reconstruction $y$ is fed to this classifier to produce a prediction $\hat{s}$, which is then used to compute the classification accuracy. Then,
> $$
> \begin{aligned}
>     H(S \mid Y)
>     &= \sum_s \sum_y p_\phi(s,y)
>        \log \frac{1}{p_\phi(s \mid y)} \\
>     &= \sum_s \sum_y p_\phi(s,y)
>        \log \frac{p_\psi(s \mid y)}{p_\phi(s \mid y)}
>        +
>        \sum_s \sum_y p_\phi(s,y)
>        \log \frac{1}{p_\psi(s \mid y)} \\
>     & = -\sum_y p_\phi(y)
>        D_{\mathrm{KL}} \bigl(
>          p_\phi(s \mid y)
>          \parallel
>          p_\psi(s \mid y)
>        \bigr)
>        +
>        \sum_s \sum_y p_\phi(s,y)
>        \log \frac{1}{p_\psi(s \mid y)}.
> \end{aligned}
> $$
> We denote the second term as the cross-entropy
> $$
> \mathrm{CE}(s,\hat s)
> =
> \sum_s \sum_y p_\phi(s,y)
> \log \frac{1}{p_\psi(s \mid y)}.
> $$
>
> Hence,
> $$
> H(S \mid Y)
> \\le
> \\mathrm{CE}(s,\\hat{s}) ,
> $$
> where the inequality follows from the non-negativity of the Kullback-Leibler (KL) divergence. When $p_\psi(s\mid y)$ is a good approximation of $p_\phi(s\mid y)$, the KL term becomes small and the cross-entropy $\mathrm{CE}(s,\hat s)$ provides a tight upper bound on $H(S \mid Y)$.
>
> [2] M. Boudiaf, Z. P. K. Millette, C. Ben Ayed, and I. B. Ayed. A unifying mutual information view of metric learning: Cross-entropy vs. pairwise losses. European Conference on Computer Vision, pages 548--564. Springer, 2020.
>
> [3] Y. Wang, Y. Wu, S. Ma and Y. -J. Angela Zhang. Task-Oriented Lossy Compression With Data, Perception, and Classification Constraints. IEEE Journal on Selected Areas in Communications, vol. 43, no. 7, pp. 2635-2650, July 2025.

---

> > ### Comment · Reviewer_AfGP · 2025-11-24
> >
> > * First, Fano is a pretty elementary and loose bound. For example, if we impose any constraint $H(S|Y) \le \epsilon < 1$, then Fano only tells us that $P(S\neq \hat{S})\ge 0$, which is trivial. I still do not think $H(S|Y)$ is related to classification accuracy. The example I give remains a valid counter example, if the distortion is not limited or if the distortion limitation is large.
> >
> > * Same as question 2. To show that H(S|Y) is actually related to classification performance, could you please provide experimental result by replacing CE with H(S|Y)? This is not hard to implement. Usually we use a upperbound when original loss is not easy to optimize. However, H(S|Y) is not hard to optimize and implement. Given a categorical distribution, it seems that torch.sum(o * torch.log o) is enough, where o is the softmax output.

---

> ### Author Response · Authors · 2025-11-26
> **Author Response -- Comments 1 and 2**
>
> > **C1.** Use of $H(S \mid Y)$ as the classification constraint in our distortion-limited and rate-limited problem setting.
>
> We agree that Fano’s inequality can indeed be loose, especially when using suboptimal classifiers. However, our analysis does not rely on Fano’s bound to assert classifier performance. Our reference to Fano was intended only to highlight that the central quantity of interest is $H(S|Y)$, which reflects the intrinsic statistical relationship between the reconstruction $Y$ and the label $S$, independent of any particular classifier. In contrast, Fano’s inequality characterizes the performance limits of any classifier, whose achievable error is fundamentally lower bounded by $H(S|Y)$.
>
> The key point is that reducing $H(S|Y)$, while still satisfying the rate constraint, is known to lower the minimum achievable classification error for any optimal or near-optimal classifier. This result is well-established in both statistical decision theory and information-theoretic learning. For example, consider $S, Y$ $\in$ \{$0,1$\}, with a joint distribution $p(s,y)$ defined so that $s = y$ with 90\% of the time and $s \neq y$ with 10\% of the time. The task is to predict $S$ from $Y$. The optimal predictor in this case is simply $\hat{S} = Y$, which yields a classification error of $0.1$. No other predictor can achieve a smaller error. This error is determined entirely by the joint distribution $p(s,y)$, specifically $H(S|Y)$, and is inherent to the statistical relationship between $S$ and $Y$, independent of any particular classifier.
>
> Regarding the comment that "the example remains a valid counterexample if the distortion is not limited or if the distortion limitation is large," we note the following. First, addressing the scenario with no distortion limitation corresponds to using no encoder and decoder at all, and instead relying solely on a classifier based on the original input as is routinely done. Second, our framework specifically addresses the second part of the comment where "large" is quantitatively defined. Specifically, our setting explicitly enforces constraints on distortion ($\mathbb{E}[d(X,Y)] \le D$), rate ($H(Z) \le R$), and classification performance ($H(S|Y) \le C$). We appreciate the reviewer’s insightful comments and hope this clarification makes clear why $H(S \mid Y)$ plays a central role in our analysis.
>
> > **C2.** Clarification of the conditional entropy $H(S \mid Y)$ and associated experiments.
>
> **Training with $H(S \mid Y)$ directly.** We note that $H(S\mid Y)$ measures the uncertainty of the true label after observing
> the reconstruction $y$, i.e.,
> $$
> \begin{aligned}
>     H(S \mid Y)
>     &= \sum_s \sum_y p_\phi(s,y)
>        \log \frac{1}{p_\phi(s \mid y)}
> \end{aligned}
> $$
> which depends on the true posterior $p_\phi(s\mid y)$.
>
> We now jointly train the classifier with the encoder/decoder. Rather than using a pretrained classifier, we alternate between (i) training the encoder-decoder with a fixed classifier and (ii) training the classifier using $H(S|Y)$ as loss function with a fixed encoder-decoder. Theoretically, the learned classifier should converge toward the true posterior $p_{\phi}(s \mid y)$ for the current reconstructions $y$. In the limit of an optimal classifier (infinite capacity and converged training), the cross-entropy becomes equal to $H(S \mid Y)$ as shown in the answer of **W2**.
>
> The resulting tradeoff curves using the exact $H(S|Y)$ in Figure 18 for the super-resolution task on MNIST and Figure 19 for the denoising task on SVHN (please see Appendix C.4 in the revised version) are consistent with those obtained using cross-entropy and confirm that decreasing $H(S \mid Y)$ corresponds to improved classification accuracy. These experiments further demonstrate the upper-bound relationship between $H(S|Y)$ and $\mathrm{CE}(s,\hat{s})$ given in the answer of **W2**.

---

> > ### Comment · Reviewer_AfGP · 2025-11-27
> >
> > Thanks for the additional results. My major concern that the experiment is disconnected with the theory is resolved. I do recommend remove the CE loss part and only present the result with $H(S|Y)$, as usually we only optimize the upperbound when the original loss is intractable. However, I still think my example is valid, as studying $R(D=\infty, P=0)$ seems to be an interesting corner case in RDP papers. Now I recommend to accept this papaer.

---

> ### Author Response · Authors · 2025-11-29
>
> We sincerely thank Reviewer AfGP for your detailed review, constructive discussions, and for revising the score (from 2 to 6) and recommendation. Your feedback prompted us to strengthen the connection between the theory and experiments, and we deeply appreciate the time and care you devoted to our paper.

---

### Official Review · Reviewer_tNsX · 2025-10-28

**Soundness:** 3
**Presentation:** 2
**Contribution:** 3
**Rating:** 6
**Confidence:** 3

**Summary:**

The paper models "cross-domain lossy compression" as a Constrained Optimal Transport problem. It derives closed-form expressions for the tradeoffs between Distortion-Rate-Classification (DRC) and Rate-Distortion-Classification (RDC).

**Strengths:**

1. The paper provides detailed and rigorous theoretical derivations.

2. The paper addresses "cross-domain lossy compression," which is more aligned with real-world application scenarios.

**Weaknesses:**

1. The core theoretical contribution of the paper lies in deriving closed-form DRC/DRPC tradeoffs for Bernoulli and Gaussian sources. However, the distribution of real-world natural images is far more complex than simple Bernoulli or Gaussian distributions. This may create a gap between the theory and practical application.

2. The paper does not provide a quantitative benchmark comparison against current state-of-the-art (SOTA) image restoration (e.g., denoising, super-resolution) or compression-de-compression (codec) methods. This makes it difficult for readers to judge whether the proposed method is superior in practical performance or merely more theoretically complete.

3. The paper's "one-shot" theory and experimental implementation both rely on shared common randomness ($U$). Although the paper mentions this can be implemented via a shared random seed, guaranteeing such randomness may be difficult in certain compression scenarios (e.g., write-once systems, broadcasting, or systems without two-way communication).

**Questions:**

see weakness.

---

> ### Author Response · Authors · 2025-11-19
> **Author Response -- Part 1**
>
> We thank the reviewer for their valuable feedback. We now address the points noted in the weaknesses section.
>
> > **W1.** Canonical Bernoulli/Gaussian sources versus real image distributions.
>
> The Bernoulli and Gaussian models in our analysis are not intended as literal models of natural images. They serve as canonical, analytically tractable cases that reveal how rate, distortion, and classification (and perception) interact in a principled manner. This follows the long-standing tradition in classical rate-distortion theory, where Gaussian sources and quadratic distortion play a foundational role in understanding the behavior of much broader classes of source distributions.
>
> On the empirical side, we deliberately tested our framework on more complex distributions: digits in natural backgrounds (SVHN), natural images (CIFAR-10, ImageNet, KODAK), and synthetic inpainting masks. Across these datasets and tasks, we consistently observe that the empirical $D(R,C)$ curves exhibit the same tradeoff behavior: distortion decreases monotonically with rate, and classification accuracy improves as rate increases. Therefore, the empirical findings capture the qualitative tradeoffs of the theoretically derived tradeoff among distortion, rate, and classification for different real-world datasets.
>
> > **W2.** Quantitative comparison with SOTA methods.
>
> Similar to our response to Reviewer F3r6, we add a quantitative comparison on the KODAK dataset corrupted by Gaussian noise with $\sigma = 25$, shown in Table 4. Our method is competitive with these methods, although the comparisons are not fair, as these methods have different objectives and constraints.  In particular, although BM3D achieves a higher PSNR, which is unsurprising, as it does not explicitly penalize the rate. On the other hand, our model performs favorably on perceptual metrics such as LPIPS, DISTS, and perceptual index (PI), thanks to the WGAN-GP discriminator. For example, our PI score is significantly better than BM3D and DeCompress and is close to the best unsupervised method OTDenoising, using a smaller representation rate. Overall, this comparison demonstrates that our framework preserves competitive restoration quality and perception. We expect that the performance of our framework can be further improved through more efficient architectural designs. The aim of this work, however, is to establish both empirically and theoretically the potential of compression-based restoration formulated via constrained optimal transport as a unified framework.
>
> **Table 4.** Comparison of denoising performance on KODAK dataset with Gaussian noise, $\mathcal{N}(0,\sigma^2)$ with $\sigma = 25$. Best values are in **bold** and second-best values are *italic*.
>
> | **Method**                     | **PSNR (dB) ↑** | **SSIM ↑** | **LPIPS ↓** | **DISTS ↓** | **PI ↓**   |
> | ------------------------------ | --------------- | ---------- | ----------- | ----------- | ---------- |
> | JPEG-2K (non-learning) [3]     | 26.4408         | 0.7357     | 0.4018      | 0.2419      | 7.4794     |
> | BM3D (non-learning) [4]        | **31.8757**     | **0.8687** | 0.2235      | 0.1640      | 2.6503     |
> | DeCompress (unsupervised) [5]  | 27.8315         | 0.7519     | 0.2627      | 0.1967      | 2.7979     |
> | OTDenoising (unsupervised) [6] | *31.2893*       | *0.8677*   | **0.1150**  | **0.1032**  | **2.0095** |
> | Ours (unsupervised)            | 27.8961         | 0.8035     | *0.1987*    | *0.1638*    | *2.1670*   |
>
> [3] David S. Taubman, Michael W. Marcellin, and Majid Rabbani. JPEG2000: Image compression fundamentals, standards and practice. Journal of Electronic Imaging, 11(2):286--287, 2002.
>
> [4] Kostadin Dabov, Alessandro Foi, Vladimir Katkovnik, and Karen Egiazarian. Image denoising by sparse 3-D transform-domain collaborative filtering. IEEE Transactions on Image Processing, 16(8):2080--2095, 2007.
>
> [5] DeCompress: Denoising via Neural Compression. Learn to Compress \& Compress to Learn Workshop, part of IEEE International Symposium on Information Theory, University of Michigan–Ann Arbor, 2025.
>
> [6] W. Wang, F. Wen, Z. Yan, and P. Liu. Optimal Transport for Unsupervised Denoising Learning. IEEE Transactions on Pattern Analysis and Machine Intelligence, 45(2):2104--2118, 2023.

---

> > ### Author Response · Authors · 2025-11-19
> > **Author Response -- Part 2**
> >
> > > **W3.** Practicality of shared common randomness ($U$).
> >
> > In Theorem 1, $U$ plays the standard role of common randomness between encoder and decoder, as is common in network information theory to model randomized encoders and test channels. Crucially, this does not require any two-way communication or per-block handshaking: $U$ can be fixed a priori as part of the codec specification. In practice, this is implemented by a pseudo-random number generator (PRNG) with a publicly known seed. All decoders (including in broadcast scenarios) run the same PRNG and therefore have access to the same realization of $U$. The seed can be fixed once and for all, or derived deterministically from metadata such as a block index or frame number. This is exactly how scramblers, interleavers, and dithering sequences are implemented in many existing standards (e.g., image/video codecs and wireless systems), including write-once or one-way broadcast settings.
> >
> > In our experiments, shared randomness appears through universal quantization: we inject uniform dither noise before quantization, which can be viewed as being generated from $U$. At deployment time, however, the encoder/decoder can either use a fixed pseudo-random sequence (public PRNG and fixed seed) or operate deterministically (e.g., by replacing the noisy quantization used during training with rounding at test time, as is standard in learned compression). In both cases, no additional bits are transmitted to synchronize $U$, and no feedback channel is required. Thus, the use of $U$ does not introduce a practical obstacle in write-once storage or broadcast-style deployments.

---

> > > ### Comment · Reviewer_tNsX · 2025-11-27
> > >
> > > Thanks the author for the detailed explanations. I have also read other reviewers' comments, and most of the responses have addressed my concerns. I will keep my current positive score.

---

> > > > ### Author Response · Authors · 2025-11-29
> > > >
> > > > We sincerely thank Reviewer tNsX for your thoughtful review and positive overall assessment. We appreciate the time you spent reading the paper, and your comments helped us refine and clarify several parts of the manuscript.

---

### Official Review · Reviewer_Av5a · 2025-10-28

**Soundness:** 3
**Presentation:** 2
**Contribution:** 3
**Rating:** 6
**Confidence:** 3

**Summary:**

The paper introduces a theoretical and algorithmic framework for cross-domain lossy compression, where the encoder observes degraded inputs while the decoder reconstructs samples from a distinct target distribution. The authors formalize this setting as a rate- and classification-constrained optimal transport (OT) problem, generalizing classical rate–distortion (RD) theory to cross-domain and task-aware scenarios. Closed-form solutions for Bernoulli and Gaussian models are derived, establishing distortion–rate–classification (DRC) and distortion–rate–perception–classification (DRPC) tradeoffs. The paper further bridges theory and practice through deep learning–based implementations, validated on tasks such as super-resolution, and denoising.

**Strengths:**

This paper makes a conceptually original and mathematically rigorous contribution by integrating optimal transport, rate-distortion theory, and classification constraints within a single unified framework.

The idea of cross-domain compression-where the reconstruction belongs to a different distribution than the source, is a clear and meaningful generalization beyond standard RD or perception-aware setups.

The theoretical development is of high quality: the paper provides single-letter characterizations, leverages shared randomness to achieve deterministic transport plans, and derives closed-form DRC expressions.

The mathematical derivations are supported by intuitive illustrations.

The experimental section, although concise, demonstrates consistency between theory and deep generative implementations, validating the theoretical tradeoffs with empirical data.

**Weaknesses:**

While they derived closed from solution of the RDC tradeoff, for Bernoulli and Gaussian sources, for practical distributions like images, their evaluation is short.

In terms of clarity, the derivations are mathematically thorough but dense, which may reduce accessibility for readers outside the information theory community.

The experimental validation, while diverse in datasets, remains relatively superficial compared to the theoretical derivations.

Discussion of related work could be more critical, especially regarding how prior task-aware generative compression frameworks differ in assumptions, experimental scope, or real-world feasibility.

**Questions:**

Could the authors create a heatmap version of Figure 6? The 2 constraints (rate and classification) could be the x and y axes, and distortion is the color of the pixel (or something similar to Figure 4a in [1]).

Could the authors create a 2D grid of generated samples from Figure 6 (similar to Figure 6 in [1])?

Could the authors provide quantitative comparisons between the theoretically predicted DRC curves and the empirical results on the Bernoulli or Gauss distribution?

Can the framework be extended to perception metrics (e.g., LPIPS or FID) rather than Wasserstein divergences?

[1] Blau, Yochai, and Tomer Michaeli. "Rethinking lossy compression: The rate-distortion-perception tradeoff." International Conference on Machine Learning. PMLR, 2019.

---

> ### Author Response · Authors · 2025-11-19
> **Author Response -- Part 1**
>
> We thank the reviewer for their valuable feedback. We first address the specific questions raised, followed by a response to the points noted in the weaknesses section.
>
> > **Q1.** Heatmap visualization of the empirical RDC function in Figure 6.
>
> We have added heatmap visualizations of the empirical RDC/CDR functions for the $4\times$ super-resolution task on MNIST and the denoising task on SVHN with Gaussian noise $\sigma = 20$. These are presented in Figure 14 and Figure 15 of the revised version in Appendix C.1, respectively. These results clearly illustrate the qualitative tradeoffs among distortion, rate, and classification accuracy: enforcing a tighter classification constraint (i.e., requiring higher accuracy) at a fixed rate consistently increases the achievable distortion. This behavior aligns with the theoretical structure of the RDC tradeoff.
>
> > **Q2.** Contour visualization of the empirical RDC function in Figure 6.
>
> We have added contour visualizations of the generated samples for the MNIST dataset on the super-resolution task and for the SVHN dataset on the denoising task, as shown in Figure 16 of the revised version in Appendix C.2. The equi-rate lines plotted on $R(D,C)$ highlight the inherent tradeoff between distortion and classification performance: for any fixed rate constraint, improving classification accuracy (i.e., achieving lower cross-entropy) requires accepting higher distortion, and conversely, reducing distortion necessitates a weaker classification constraint.
>
> > **Q3.** Quantitative comparison between theoretical and empirical DRC curves.
>
> We have added experiments to quantitatively compare the closed-form DRC curves with empirical estimates on synthetic data. The results are summarized in Figure 17 of the revised version in Appendix C.3. We will include these figures in the final manuscript.
>
> > **Q4.** Extending the framework to perceptual metrics such as LPIPS and FID.
>
> We understand this question as primarily concerning the empirical framework, i.e., the loss functions used when training our model. Our framework can be extended to handle different perception terms in the loss function; for example, one can use the Jensen-Shannon divergence in classical generative adversarial networks. In practice, we instantiate this term using standard feature-based perceptual losses on the reconstructed/denoised images, such as learned perceptual image patch similarity (LPIPS), deep image structure and texture similarity (DISTS), natural image quality evaluator (NIQE), perception-based image quality evaluator (PIQE), and perceptual index (PI) in our experiments (please see Tables 1, 2, and 3 in our response to Reviewer F3r6). Replacing (or augmenting) the current perception loss by LPIPS, fréchet inception distance (FID), or other learned-feature metrics is therefore straightforward. It does not require any change to the overall architecture or optimization procedure.
>
> > **W1.** Evaluation of the RDC tradeoff on practical image datasets.
>
> Our goal is to bridge the new information-theoretic characterization with realistic cross-domain restoration tasks. To this end, we have designed experiments spanning multiple datasets, including MNIST, SVHN, CIFAR-10, ImageNet, and KODAK as well as multiple tasks such as super-resolution, denoising, and inpainting, as detailed in Section 5 and Appendix B.1-B.2. Across these settings, we consistently observe that increasing rate improves both distortion and classification performance, while stronger task constraints (smaller $C$) restrict the achievable distortion. These results, therefore, capture the qualitative tradeoffs among distortion, rate, and classification for different real-world datasets.
>
> > **W2.** Improving clarity and reader accessibility in the theoretical section.
>
> Our primary audience includes both information theorists and practitioners in machine learning and computer vision, hence we agree that accessibility is important. In the revised version, we will improve clarity along several axes:
>
> * **Interpretive remarks.** After each main theorem, we currently include remarks describing the intuition and implications. We will expand these remarks and highlight them more explicitly so that readers can understand the qualitative meaning without following all algebraic steps.
>
> * **Figure-based intuition.** We will strengthen the connection between the theoretical formulas and the visualizations in Figures 2-4, which illustrate the shapes of $D(R,C)$. These figures provide accessible geometric intuition for readers.

---

> ### Author Response · Authors · 2025-11-19
> **Author Response -- Part 2**
>
> > **W3.** Experimental validation supporting the theoretical framework.
>
> Our primary contribution is a theoretical unified constrained optimal transport formulation together with closed-form DRC/DRPC characterizations. The experimental section is designed to validate and interpret these results across a range of representative tasks, rather than to exhaustively benchmark every architecture on every dataset.
>
> We believe that the empirical study is substantive and supports the theoretical development. It includes: (i) cross-domain super-resolution on MNIST; (ii) denoising on SVHN, CIFAR-10, ImageNet, and KODAK for multiple noise levels; and (iii) supervised and unsupervised inpainting on SVHN, all implemented with a single unified compression architecture. For each setting, we present rate-distortion and rate-accuracy curves (and now heatmaps and contour visualization), showing consistent agreement with our theory. Furthermore, as requested by Reviewer F3r6 and Reviewer tNsX, we add quantitative comparisons with denoising baselines on KODAK and additional real-world data experiments on fluorescence microscopy images and photographic noise smartphone images. We view this breadth of tasks and datasets as evidence that the proposed framework is both theoretically well-founded and practically impactful.
>
> > **W4.** Related work and comparison to prior task-aware generative compression frameworks.
>
> We will revise the related-work section to more explicitly differentiate assumptions and contributions:
> * **Rate-distortion-perception works** (e.g., Blau \& Michaeli, 2019 [1], and follow-ups) consider distortion-rate-perception tradeoffs but do not model downstream classification explicitly and typically operate in a single-domain setting (source and target distributions coincide). In contrast, we study cross-domain lossy compression with explicit classification constraints, leading to DRC and DRPC functions that simultaneously account for rate, distortion, and task loss.
>
> * **Cross-domain compression as entropy-constrained optimal transport (OT)** (e.g., Liu et al., 2022 [2]) treats rate-constrained transport between different domains but does not incorporate classification or perception constraints and does not derive closed-form DRC/DRPC expressions.
>
> * **Task-aware compression and RDC-style methods** (e.g., Zhang, 2023 [3]) analyze rate-distortion-classification tradeoffs but consider simpler single-domain settings and do not handle the cross-domain setting with shared randomness and perception divergences, nor do they provide the Bernoulli/Gaussian closed forms we derive.
>
> [1] Yochai Blau and Tomer Michaeli. Rethinking lossy compression: The rate–distortion–perception tradeoff. In International Conference on Machine Learning, pages 675–685. PMLR, 2019.
>
> [2] Huan Liu, George Zhang, Jun Chen, and Ashish Khisti. Cross-Domain Lossy Compression as Entropy Constrained Optimal Transport. IEEE Journal on Selected Areas in Information Theory, 3(3):513--527, 2022.
>
> [3] Yuefeng Zhang. A Rate-Distortion-Classification approach for lossy image compression. Digital Signal Processing, 141:104163, 2023.

---

> > ### Comment · Reviewer_Av5a · 2025-11-27
> >
> > The concerns and questions I raised in my initial review have been addressed, and I appreciate the clarifications provided.

---

> > > ### Author Response · Authors · 2025-11-29
> > >
> > > We sincerely thank Reviewer Av5a for carefully reading our paper and for your constructive and positive feedback. Your comments and suggestions were very helpful to improve the clarity and presentation of our work, and we truly appreciate your time and effort.

---

### Official Review · Reviewer_F3r6 · 2025-10-31

**Soundness:** 4
**Presentation:** 4
**Contribution:** 4
**Rating:** 10
**Confidence:** 5

**Summary:**

This paper presents a comprehensive theoretical and empirical framework for cross-domain lossy compression, where the encoder observes a degraded source while the decoder reconstructs samples from a distinct target distribution. The authors formulate this as a constrained optimal transport problem with dual constraints on compression rate and classification loss. The key theoretical contributions include closed-form characterizations of distortion-rate-classification (DRC) and rate-distortion-classification (RDC) tradeoffs for both Bernoulli sources under Hamming distortion and Gaussian models under mean-squared error. The framework is further extended to incorporate perception constraints using Kullback-Leibler and squared Wasserstein divergences. The theoretical findings are validated through extensive experiments on super-resolution (MNIST), denoising (SVHN, CIFAR-10, ImageNet, KODAK), and inpainting (SVHN) tasks, demonstrating strong consistency between theoretical predictions and empirical performance.

**Strengths:**

​Theoretical Novelty and Rigor:​​ The paper makes significant theoretical contributions by unifying optimal transport with rate-distortion theory and classification constraints. The derivation of closed-form solutions for both one-shot and asymptotic regimes represents substantial mathematical advancement in the field.
​​Comprehensive Framework:​​ The authors successfully integrate multiple constraints—compression rate, distortion, perception quality, and classification performance—into a unified framework. The extension to distortion-rate-perception-classification (DRPC) functions with explicit perception constraints is particularly noteworthy.
​​Strong Experimental Validation:​​ The paper provides extensive experimental validation across multiple datasets and tasks. The implementation of deep end-to-end compression models with quantization, entropy modeling, adversarial training, and classification components demonstrates practical applicability.
​​Practical Relevance:​​ The problem formulation addresses real-world scenarios where compressed representations must serve multiple purposes simultaneously—maintaining perceptual quality, supporting downstream tasks, and operating under rate constraints.

**Weaknesses:**

​​Limited Real-World Application Scope:​​ Although the experiments cover multiple datasets, they primarily focus on standard academic benchmarks. Validation on more practical, real-world compression scenarios would strengthen the paper's impact.
​​Computational Complexity:​​ The paper does not sufficiently address the computational requirements of the proposed approach, particularly for the adversarial training components and their scalability to very high-resolution images.
​​Comparison with State-of-the-Art:​​ While the paper includes comparisons with baseline methods, more extensive comparisons with recent state-of-the-art learned compression methods would provide better context for the proposed approach's advantages.

**Questions:**

​Scalability and Generalization:​​ The theoretical results are derived for Bernoulli and Gaussian sources. How well do these theoretical predictions hold for more complex, multi-modal distributions encountered in real-world data?
​​Optimization Tradeoffs:​​ The paper presents multiple constraints (rate, distortion, perception, classification). In practical applications, how should practitioners balance these competing objectives, and are there guidelines for setting the constraint parameters?

---

> ### Author Response · Authors · 2025-11-19
> **Author Response -- Part 1**
>
> We thank the reviewer for their valuable feedback. We first provide answers to the specific questions raised, followed by detailed responses to the points noted in the weaknesses section.
>
> > **Q1.** Scalability and generalization to complex multi-modal real-world data.
>
> We agree with the reviewer that natural images exhibit highly complex, multi-modal statistics that go beyond idealized Bernoulli or Gaussian sources. Our analytic DRC/DRPC characterizations for Bernoulli and Gaussian distributions are intended to play a role analogous to classical rate-distortion results: they provide closed-form tradeoffs that clarify, in principle, how rate, distortion, and classification (or perception) interact.
>
> We show empirical DRC curves for multiple image tasks. As detailed in Section 5 and Appendix B.1, our experiments on super-resolution (MNIST), denoising (SVHN, CIFAR-10, ImageNet, KODAK), and inpainting (SVHN) all exhibit the same tradeoff behavior: distortion decreases monotonically with rate increases, and classification accuracy improves as rate increases. Thus, these results do capture the qualitative tradeoffs among distortion, rate, and classification for different real-world datasets.
>
> > **Q2.** Practical guidelines for balancing rate, distortion, perception, and classification constraints.
>
> In practice, we implement the constrained problem via a Lagrangian objective of the form
> $$
> \mathcal{L} = \text{MSE} + \lambda_r R + \lambda_p \text{Perception} + \lambda_c \text{CE}(S,\\hat S)
> $$
> where $R$ is the rate estimate based on the entropy model, Perception denotes a divergence such as a Wasserstein distance, and CE is the classification loss. During training, we sweep $(\lambda_r,\lambda_p,\lambda_c)$ over a grid and then re-parameterize each trained model by its realized pair $(R,C)$ measured on a validation set, thereby empirically tracing out the DRC surface.
>
> For practitioners, a natural workflow is therefore: (i) choose an application-specific tolerance on classification accuracy, which implicitly specifies a range of admissible $C$; (ii) select a desired operating range of rates; and (iii) train a small grid of models with gradually increasing $\lambda_r$ (to move along the rate axis) and, if needed, $\lambda_c$ (to enforce stricter task fidelity). The resulting family of models can then be projected onto the empirical $D(R,C)$ surface, from which one can pick the operating point that best matches application constraints. When perception is critical, $\lambda_p$ can be turned on to move along the perception axis, trading off small increases in distortion for large improvements in perceptual quality, in accordance with the DRPC tradeoff. We will clarify this hyperparameter selection strategy in the revised manuscript to better guide practitioners.
>
> > **W1.** Empirical validation on real-world imaging and photographic noise.
>
> In addition to standard benchmarks, we include experiments on real-world fluorescence microscopy images (Mouse Nuclei) [1], which exhibit statistics and structures very different from natural photographic images, under Gaussian noise $\mathcal{N}(0,\sigma^2)$. As shown in Table 1, our method achieves decent denoising performance under Gaussian noise across different noise levels, with consistently high peak signal-to-noise ratio (PSNR) and structural similarity index measure (SSIM), as well as low perceptual distortions such as learned perceptual image patch similarity (LPIPS) and deep image structure and texture similarity (DISTS), averaged over 67 test images.
>
> **Table 1.** Performance values for denoising on the Mouse Nuclei dataset.
>
> | **σ** | **PSNR (dB) ↑** | **SSIM ↑** | **LPIPS ↓** | **DISTS ↓** |
> | ----- | --------------- | ---------- | ----------- | ----------- |
> | 10    | 33.0340         | 0.8052     | 0.0443      | 0.1401      |
> | 20    | 30.5880         | 0.8028     | 0.0734      | 0.1675      |
>
>
> To further probe robustness under real-world photographic noise, we include experiments on the SIDD smartphone image dataset [2], which contains complex, signal-dependent noise patterns arising from real camera pipelines, averaged over 10 test images. These additional experiments demonstrate that the proposed framework remains effective beyond controlled synthetic settings and applies to challenging real-world compression/restoration scenarios.
>
> **Table 2.** Performance values on the SIDD dataset.
>
> | **PSNR (dB) ↑** | **SSIM ↑** | **LPIPS ↓** | **DISTS ↓** |
> | --------------- | ---------- | ----------- | ----------- |
> | 33.605          | 0.9038     | 0.3233      | 0.2366      |
>
> [1] Tim-Oliver Buchholz, et.al. Denoiseg: Joint denoising and segmentation. In European Conference on Computer Vision, pages 324–337. Springer, 2020.
>
> [2] A. Abdelhamed, S. Lin, and M. S. Brown. A high-quality denoising dataset for smartphone cameras. In IEEE Conference on Computer Vision and Pattern Recognition (CVPR), pages 1692--1700, 2018.

---

> ### Author Response · Authors · 2025-11-19
> **Author Response -- Part 2**
>
> > **W2.** Computational complexity and scalability of the proposed framework.
>
> Our model uses standard components from learned compression and restoration: a convolutional autoencoder with an entropy model, a WGAN-GP discriminator, and a classifier (the detailed neural network architectures are provided in Appendix B.3.4 and B.4.4). Training was performed on two RTX 3090 GPUs under Ubuntu using the PyTorch framework. Both training and inference are dominated by a single forward/backward pass through the autoencoder. The discriminator and classifier reuse the same feature maps and therefore introduce only modest computational overhead, while rate estimation via the entropy model and universal quantization is computationally negligible.
>
> For very high-resolution images, we follow the standard patch/tile strategy used in modern learned codecs, so the overall complexity scales approximately linearly with the number of pixels and remains comparable to existing adversarial denoising and compression methods. We will clarify these computational aspects and report runtimes in the revised manuscript.
>
> > **W3.** Comparisons with other compression and denoising approaches.
>
> We add a quantitative comparison on the KODAK dataset corrupted by Gaussian noise with $\sigma = 25$, shown in Table 3. Our method is competitive with these methods, although the comparisons are not fair, as these methods have different objectives and constraints.  In particular, although BM3D achieves a higher PSNR, which is unsurprising, as it does not explicitly penalize the rate. On the other hand, our model performs favorably on perceptual metrics such as LPIPS, DISTS, and perceptual index (PI), thanks to the WGAN-GP discriminator. For example, our PI score is significantly better than BM3D and DeCompress and is close to the best unsupervised method OTDenoising, using a smaller representation rate. Overall, this comparison demonstrates that our framework preserves competitive restoration quality and perception. We expect that the performance of our framework can be further improved through more efficient architectural designs. The aim of this work, however, is to establish both empirically and theoretically the potential of compression-based restoration formulated via constrained optimal transport as a unified framework.
>
> **Table 3.** Comparison of denoising performance on KODAK dataset with Gaussian noise, $\mathcal{N}(0,\sigma^2)$ with $\sigma = 25$. Best values are in **bold** and second-best values are *italic*.
>
> | **Method**                     | **PSNR (dB) ↑** | **SSIM ↑** | **LPIPS ↓** | **DISTS ↓** | **PI ↓**   |
> | ------------------------------ | --------------- | ---------- | ----------- | ----------- | ---------- |
> | JPEG-2K (non-learning) [3]     | 26.4408         | 0.7357     | 0.4018      | 0.2419      | 7.4794     |
> | BM3D (non-learning) [4]        | **31.8757**     | **0.8687** | 0.2235      | 0.1640      | 2.6503     |
> | DeCompress (unsupervised) [5]  | 27.8315         | 0.7519     | 0.2627      | 0.1967      | 2.7979     |
> | OTDenoising (unsupervised) [6] | *31.2893*       | *0.8677*   | **0.1150**  | **0.1032**  | **2.0095** |
> | Ours (unsupervised)            | 27.8961         | 0.8035     | *0.1987*    | *0.1638*    | *2.1670*   |
>
> [3] David S. Taubman, Michael W. Marcellin, and Majid Rabbani. JPEG2000: Image compression fundamentals, standards, and practice. Journal of Electronic Imaging, 11(2):286--287, 2002.
>
> [4] Kostadin Dabov, Alessandro Foi, Vladimir Katkovnik, and Karen Egiazarian. Image denoising by sparse 3-D transform-domain collaborative filtering. IEEE Transactions on Image Processing, 16(8):2080--2095, 2007.
>
> [5] DeCompress: Denoising via Neural Compression. Learn to Compress \& Compress to Learn Workshop, part of IEEE International Symposium on Information Theory, University of Michigan–Ann Arbor, 2025.
>
> [6] W. Wang, F. Wen, Z. Yan, and P. Liu. Optimal Transport for Unsupervised Denoising Learning. IEEE Transactions on Pattern Analysis and Machine Intelligence, 45(2):2104--2118, 2023.

---

> > ### Comment · Reviewer_F3r6 · 2025-11-27
> > **Response to authors' rebuttal**
> >
> > Thank the authors for the rebuttal. I recommend accepting this paper as a highlight.

---

> > > ### Author Response · Authors · 2025-11-29
> > >
> > > We sincerely thank Reviewer F3r6 for the very thoughtful and positive review, and for recommending our work for acceptance as a highlight. We really appreciate the time and care you put into evaluating our paper and your encouraging assessment of both the theory and experiments.

---

### Author Response · Authors · 2025-11-29
**Summary of Author Response**

We would like to thank all reviewers for their time and constructive comments. We believe that the comments and questions have helped to greatly improve the manuscript. Based on the reviewers' comments, we have revised our manuscript. The new changes are **marked in blue**.

A summary of our responses and revisions is as follows:

* **Reviewer F3r6:** The comments emphasized real-world applicability, scalability, and guidance for choosing operating points on the DRC/DRPC tradeoff. In response, we added additional experiments on real-world microscopy (Mouse Nuclei) and smartphone photographic noise (SIDD), expanded quantitative comparisons on KODAK with standard baselines, and clarified how practitioners can navigate the rate-distortion-classification-perception tradeoffs in practice.

* **Reviewer Av5a:** The comments focused on better visualizing the empirical RDC surfaces and strengthening the link between theory and experiments. We added heatmap and contour visualizations of the empirical DRC/RDC surfaces, included synthetic experiments comparing the closed-form DRC curves with empirical estimates, and clarified how our framework extends to perception divergences (e.g., Wasserstein-based and learned perceptual metrics).

* **Reviewer tNsX:**    The comments highlighted the gap between Bernoulli/Gaussian models and real image distributions, the need for quantitative benchmarks against existing methods, and the practicality of shared common randomness. In response, we added quantitative comparisons with classical and learned denoising/compression baselines on KODAK and real-world datasets, and we clarified how the shared randomness is implemented via a public PRNG/seed in a way that is compatible with codecs, broadcasting, and write-once storage.

* **Reviewer AfGP:**     The comments raised concerns about the formulation of the classification constraint via $H(S \mid Y)$ and its relationship to the cross-entropy loss used in experiments, and about the alignment between the theoretical objective and the empirical setup. In response, we implemented experiments that directly optimize and report the exact conditional entropy $H(S \mid Y)$ by jointly training the classifier with the encoder-decoder, and showed that the resulting tradeoff curves are consistent with the cross-entropy-based curves and track classification accuracy. We also clarified the interpretation of $H(S\mid Y)$ as the fundamental quantity in our theory.

Please kindly check our responses and revisions below and in the revised manuscript.

---

### Meta-Review · Area_Chair_hy82 · 2026-01-01

**Summary:**

All reviewers give positive ratings during initial assessment or rebuttal, and there are no outstanding remaining concerns.

The paper makes significant theoretical contributions by unifying optimal transport with rate-distortion theory and classification constraints. The derivation of closed-form solutions for both one-shot and asymptotic regimes represents substantial mathematical advancement in the field.​​

The experiment demonstrates consistency between theory and deep generative implementations, validating the theoretical tradeoffs with empirical data.



Though there are still some concerns related to practical applications and SOTA method comparisons, which I think is not fully solved during the rebuttal, I recommend to accpet this manuscript as it is more theory-oriented and provide novel and enlightening results.

**Reviewer Concerns:**

Concerns addressed:
Practical guidelines for balancing rate, distortion, perception, and classification constraints.
Empirical validation on real-world imaging and photographic noise.
Computational complexity and scalability of the proposed framework.
Comparisons with other denoising approaches.
Related work and comparison to prior task-aware generative compression frameworks.
Better visualizing the empirical RDC surfaces and strengthening the link between theory and experiments.
The gap between Bernoulli/Gaussian models and real image distribution.
The practicality of shared common randomness.
The formulation of the classification constraint and its relationship to the cross-entropy loss used in experiments.


Concerns not fully addressed:
Scalability and generalization to complex multi-modal real-world data. （F3r6）
Lack comparisons with recent state-of-the-art learned compression method. (F3r6, tNsX)

**Reviewer Scores:**

F3r6 might keep rating 10 and F3r6 recommended accepting this paper as a highlight.
Av5a might keep rating 6 or increse the score.
tNsX kept rating 6.
AfGP said the major concern that the experiment is disconnected with the theory is resolved, and recommend to accept this papaer. So AfGP might rate 6.

---

### Decision · Program_Chairs · 2026-01-26

Accept (Oral)